# Single-cell atlas of the developing Down syndrome brain cortex

Michael Lattke [1]✉, Wee Leng Tan [2], Salil Kalarikkal Sukumaran[2], Kagistia Hana Utami[2], Marcos Sintes[1,2], Srinivasan Sakthivel[2], Jonathan Tan [2], Auriel Lim[2], Vibhavari Aysha Bansal [2], Katerina Rekopoulou[3], Nik Matthews [3], Ivan Alić [4,5], Željka Krsnik[6], Dean Nižetić [4], Boaz P. Levi [7] & Vincenzo De Paola [1,2]✉

Down syndrome (DS), caused by trisomy of chromosome 21, is the leading genetic cause of intellectual disability, yet the mechanisms disrupting fetal brain development remain unclear. We performed single-cell transcriptomic and chromatin accessibility profiling of approximately 250,000 cells from 15 DS and 15 control human fetal cortices (10–20 weeks postconception). Our analysis revealed a subtype-specific reduction in RORB- and FOXP1-expressing excitatory neurons and widespread disruption of neurodevelopmental transcriptional programs. Chromosome 21 transcription factors BACH1, PKNOX1 and GABPA emerged as dosage-sensitive hubs regulating genes linked to intellectual disability. Antisense oligonucleotide-mediated normalization of these transcription factors in human neural progenitors in vitro partially rescued target gene expression. Benchmarking a humanized in vivo model captured additional molecular and cellular signatures of DS, complementing the in vitro model. Together, we present a resource defining the gene-regulatory landscape underlying cortical development in DS and highlight molecular pathways for further investigation.

Trisomy of chromosome 21 (Ts21) is the most common chromosomal abnormality in humans and a major cause of intellectual disability[1–3]. Despite the severe impact on quality of life, there are no effective treatments available for the intellectual disability and other neurological manifestations of DS, such as early-onset Alzheimer-like dementia, behavioral impairments and infant seizure susceptibility[1–3]. Imaging and postmortem studies have revealed reduced fetal brain volume starting from gestational week (GW) 23 (that is, postconceptional week (PCW) 21), linked to decreased neural progenitor proliferation and excitatory neuron production from GW18, increased astrogliogenesis, and impaired dendrite and synapse formation in postnatal stages[2]. Traditional candidate-based genetic approaches in mice and stem

cell models have identified various genes that may contribute to these phenotypes, such as *DYRK1A*, *DSCAM*, *OLIG2*, *APP* and *IFNAR1/2*[2,4,5]. Previous efforts to comprehensively understand global gene expression dysregulation in the DS brain using stem cells, mouse models and postmortem human samples have provided valuable insights into altered molecular pathways in Ts21[6–10]. However, postmortem human studies have either focused on adult stages[11,12] or lacked cellular resolution[13]. As a result, despite important insights from previous work, it is still unclear how the subtle (approximately 1.5-fold) increase in gene dosage of ~200 chromosome (Chr.) 21 genes due to Ts21 impacts the development and function of various cell populations in the human cortex, the region central to cognitive functions. Specifically, we lack

[1]Department of Brain Sciences, Imperial College London, London, UK. [2]Duke-NUS Medical School, Singapore, Singapore. [3]NIHR Imperial BRC Genomics Facility, Imperial College London, London, UK. [4]The Blizard Institute, Queen Mary University of London, London, UK. [5]Department of Anatomy, Histology and Embryology, University of Zagreb, Zagreb, Croatia. [6]Croatian Institute for Brain Research, School of Medicine, University of Zagreb, Zagreb, Croatia. [7]Allen Institute for Brain Science, Seattle, WA, USA. ✉e-mail: m.lattke@imperial.ac.uk; videpa@nus.edu.sg

insights into how these changes affect the genomic programs that govern human cortical development and function during the critical period (approximately PCW10 to PCW20) when most neurons are generated[14]. This period lays the foundation for cortical organization and connectivity, likely contributing to the neurological features observed in DS. By integrating single-cell genomics (single-nucleus RNA sequencing (snRNA-seq) and single-cell ATAC sequencing) across in vitro and in vivo human neuron models, we constructed a publicly available atlas of the developing DS cortex (PCW10–20). This resource provides an unprecedented cellular-resolution view of gene expression and regulatory architecture, revealing some of the earliest cellular and molecular perturbations in the condition.

## Results

### A single-cell gene expression and chromatin accessibility atlas of the human fetal cortex in DS

To identify early molecular and cellular changes in the developing DS cortex, we performed single-cell transcriptional and chromatin accessibility profiling (10X Genomics Multiome) on 15 DS and 15 diploid control fetal brain samples spanning PCW10–20, the period from early cortical neurogenesis to early gliogenesis (Fig. 1a,b). After quality control and mapping to a reference atlas[15], we retained 248,998 high-quality cells from samples including mostly cortical tissue (Methods, Extended Data Fig. 1 and Supplementary Table 1). Dimensionality reduction and clustering identified 21 cell populations, which we characterized using established markers and the reference atlas[15] (Fig. 1c,d and Extended Data Figs. 1c,d and 2a,b). Most cells expressed high levels of neural lineage markers, including four populations expressing the radial glia and astrocyte markers *SLC1A3* (also known as GLAST), *SOX9*, *PAX6*, *NES* and *HOPX* (RG_c0/c11, RG_prol_c8, AST_c13). One of these populations (RG_prol_c8) coexpressed proliferation markers *MKI67*, *TOP2A* and *CDK1*, and one (AST_c13) coexpressed astrocyte-specific markers *ALDH1L1*, *GFAP*, *GJA1* (CX43), *AQP4* and *S100B*. Three populations expressed excitatory intermediate progenitor cell markers *EOMES*, *GADD45G* and *ASCL1*, including one coexpressing proliferation markers (IPC_c5/c12, IPC_prol_c9). Seven populations expressed excitatory cortical pyramidal neuron markers *NEUROD2* and *NEUROD6*, including one (NEU_TLE4_c3) expressing the L5–L6 neuron markers *TLE4*, *TBR1* and *BCL11B* (CTIP2), one expressing the L4 markers *RORB* and *FOXP1* (NEU_RORB_c4), three expressing the L2–L3 marker *CUX2* (NEU_CUX2_c0/c2/c10) and two minor populations only lowly expressing subtype markers (NEU_low_c17/c20). Consistent with their putative identity as cortical excitatory lineage cells, these clusters expressed the dorsal forebrain marker *FOXG1*. Four populations expressed *GAD1*, *GAD2* and markers of ventrally, ganglionic eminences-derived cells (LHX6, DLX2, ADARB2), consistent with GABAergic interneuron identity. These also expressed different interneuron subtype markers, including *SST*, *CALB2* (calretinin) and *RELN* (NEU_SST_c6, NEU_CALB_c7, NEU_RELN_c14/c15). We also found three minor populations expressing the oligodendrocyte lineage markers *PDGFRA*, *CSPG4* (NG2), *MBP* and *MOG*, putative oligodendrocyte precursor cells (OPC_c16), microglia markers (CX3CR1, ITGAM; MIC_c19), or endothelial cell and pericyte markers *PECAM1*, *CLDN5*, *MCAM* and *PDGFRB*, putative vascular cells (VASC_c18).

Most populations were present in all samples (Fig. 1e), although their abundance varied strongly between samples. Nevertheless, compositional changes broadly followed expected developmental patterns, with later stages including more late-developing *RORB*- and *FOXP1* (*RORB/FOXP1*)-expressing neurons and *SST*-interneurons (NEU_RORB, NEU_SST) and fewer progenitors (Extended Data Fig. 2c,d).

Notably, L4-like neurons expressing *RORB/FOXP1* were dramatically reduced in DS samples, particularly at later stages (Fig. 1e and Extended Data Fig. 2d), a defect previously reported only in adult DS[11] and in people with Alzheimer disease[16]. We confirmed this phenotype with an alternative cluster-free analysis (MiloR[17]), and FOXP1 immunostainings of tissue from PCW16–20, including in two additional

pairs of well-preserved paraffin-embedded brains and sections from cryopreserved brains used for the transcriptomic analyses (Methods, Fig. 1f and Extended Data Fig. 2e–g). Contrary to previous reports of reduced proliferating progenitors and increased interneuron or astrocyte numbers in DS at later stages[2], we found no changes in progenitor, interneuron or astrocyte numbers at PCW10–20 (Fig. 1e).

Overall, our mid-gestation dataset primarily encompasses neural cells, including the entire excitatory lineage, multiple interneuron populations and early glial cells. It reveals a marked reduction in putative L4 pyramidal excitatory neurons expressing *RORB/FOXP1* as the earliest cellular phenotype, while other previously reported compositional changes could not be detected, suggesting they may arise at a later stage.

### Gene expression changes mainly affect excitatory neurons and are linked to neural development and function

Next, we investigated how Ts21 affects global gene expression, to identify cell types and genetic programs that may contribute to the biological features associated with DS. We compared gene expression between DS and controls (CON) for each cell cluster using a pseudobulk-based approach with a low differential expression threshold (1.2-fold; Methods) to allow detection of subtly deregulated genes, including Chr. 21 genes, expected to be upregulated 1.5-fold on average because of the presence of the additional copy of Chr. 21.

As expected[18], the 87 differentially expressed Chr. 21 genes were exclusively upregulated in a wide range of cells (Extended Data Fig. 3a and Supplementary Table 2). Of the remaining 732 differentially expressed genes (DEGs), the majority were identified in *RORB/FOXP1*-expressing neurons (NEU_RORB_c4)−whose abundance is reduced−as well as in *TLE4*-expressing neurons (NEU_TLE4_c3), and in two smaller populations (NEU_RELN_c14, NEU_low_c17). To identify biological processes likely affected by the observed transcriptomic changes, we performed a Gene Ontology (GO) analysis (Extended Data Fig. 3b and Supplementary Table 2). Most prominent among the 114 enriched GO terms were those related to neurodevelopmental processes, whose deregulation could contribute to cognitive impairment in DS, such as 'forebrain development', 'neural precursor cell proliferation', 'regulation of neuron differentiation', 'axonogenesis' or 'dendrite development'.

Because most inhibitory neurons and microglia−cell types previously reported to be affected in DS[11]−showed only limited transcriptional changes in our dataset, we focused our analyses on excitatory neurons, in which we detected substantially more widespread transcriptional alterations. To investigate these changes in more detail, we subsetted and re-clustered our dataset, retaining only excitatory neurons and their progenitors, including putative immature astrocytes, which derive from the same progenitors and are not unambiguously distinguishable from radial glia (Methods, Fig. 2a and Extended Data Fig. 4a,b). In all cell clusters in this subset together, we detected 672 DEGs (Fig. 2b,c and Supplementary Table 3), including many non-Chr. 21 genes in *RORB/FOXP1*- and *TLE4*-expressing neurons (NEU_RORB_s4, NEU_TLE4_s3) (Fig. 2b), which are also reduced in abundance (Extended Data Fig. 4c). Differential genes were enriched for GO terms linked to processes impaired in DS, including 'cognition' and neurodevelopmental terms, such as 'forebrain development', 'sensory organ morphogenesis', 'axonogenesis', 'dendrite development', 'gliogenesis' and 'neural precursor cell proliferation'[2] (Fig. 2c and Supplementary Table 3). Genes showed generally only subtle changes and included Chr. 21 genes such as *APP*, *GART* and *C21ORF91*, as well as key transcription factors (TFs), receptors and ligands involved in cortical neuron specification and differentiation located on other chromosomes, such as *NEUROG2*, *FEZF2*, *FOXP1*, *NTRK2*, *FGFR2*, *NOTCH1* and *WNT4* (Fig. 2d). Of note, *FOXP1*, whose mutations can cause intellectual disability, and which has been implicated in impaired generation of L4 to L6 excitatory neurons expressing RORB or TLE4, respectively[19],

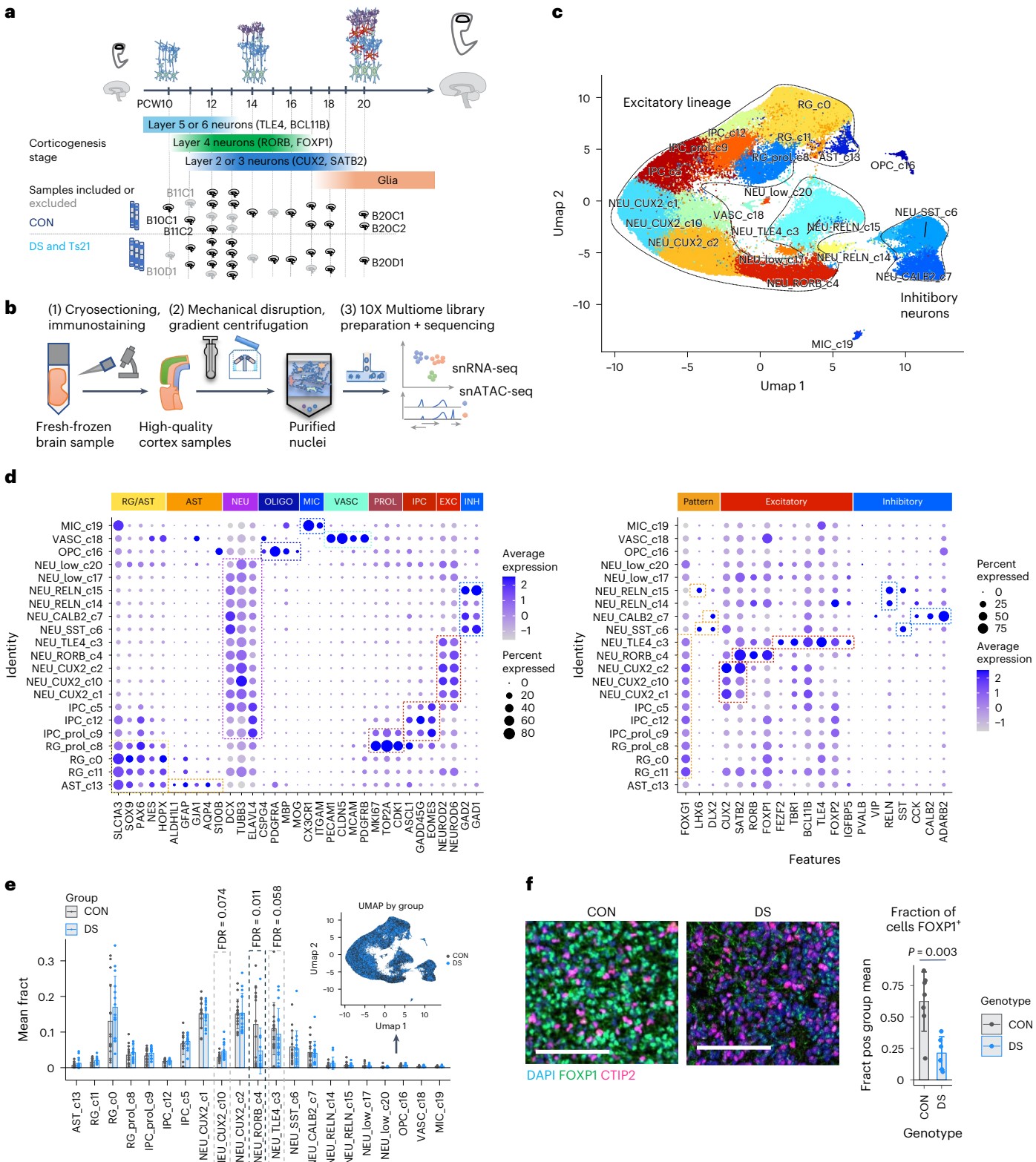

**Fig. 1 | A single-cell gene expression and chromatin accessibility atlas of the human fetal cortex in DS. a**, Stages of cortical development covered by fetal tissue samples used for this dataset. **b**, Experimental pipeline for the processing of brain samples for control stainings, nuclei extraction and combined single-cell transcriptome and chromatin accessibility analyses. **c**, Cell type assignment of identified cell clusters (Uniform Manifold Approximation and Projection (UMAP) plot). **d**, Expression of marker genes used to assign clusters to cell types (left) and subtypes (right). **e**, Abundance of cell populations in CON and DS samples. Barplot showing individual samples (n = 15 CON and n = 15 DS) and

mean ± s.d. with the false discovery rate (FDR) for DS versus CON from sccomp compositional analysis[71] (other clusters FDR > 0.05). Inset: combined UMAP plot (arrow indicates NEU_RORB_c4 cluster reduced in DS). **f**, FOXP1 immunostaining in CON and DS brains from PCW16–20. Sections from cryopreserved brains used for sequencing analyses (representative CTIP2 positive cortical plate areas from images analyzed for quantification). Scale bar: 100 μm. Barplot showing individual samples (n = 7 CON and n = 6 DS) and mean ± s.d. Statistical analysis: two-tailed t-test. See also Extended Data Figs. 1 and 2, and Supplementary Table 1. scATAC-seq, single-cell ATAC sequencing.

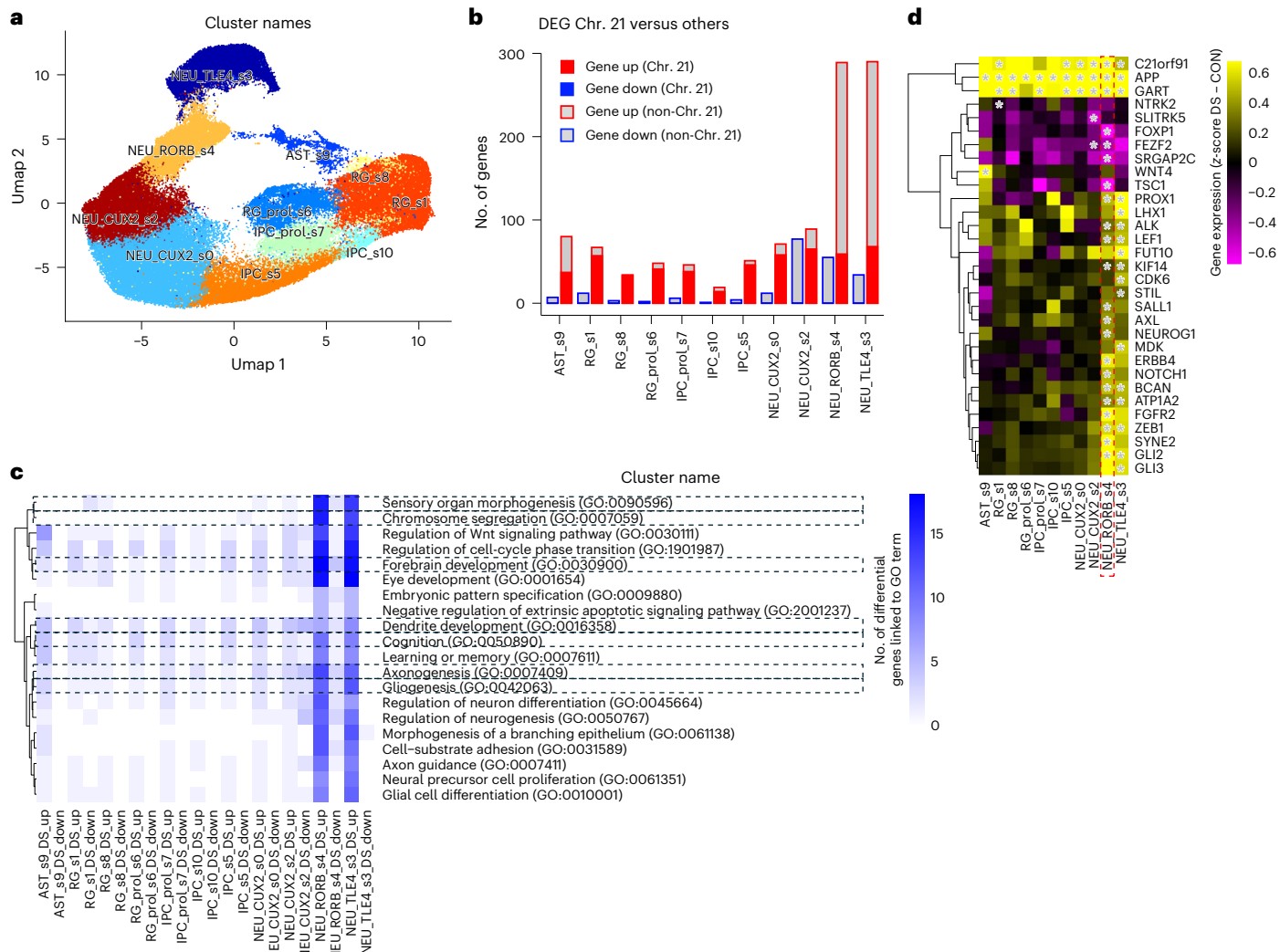

**Fig. 2 | Gene expression changes mainly affect excitatory neurons and are linked to neural development and function. a**, Cell populations of the subsetted and re-clustered excitatory lineage PCW10–20 dataset (UMAP plot). Marker expression for cluster assignment is shown in Extended Data Fig. 4a,b. **b**, Number of genes differentially expressed between DS and CON samples by cluster; DESeq2 pseudobulk analysis with Wald test, threshold for adjusted $P$ value ($P_{adj}$) < 0.10, |$\log_2$(FoldChange)| > $\log_2$(1.2). **c**, Biological processes linked to DEGs: heatmap showing number of differentially expressed genes in selected enriched GO terms by cell cluster. GO terms referred to in the main text are highlighted by grey dotted outlines. **d**, DEGs linked to the GO term 'forebrain development' by cluster. The heatmap is colored by relative expression in DS versus CON (difference vst-normalized gene expression $z$-score DS versus CON samples). The red dotted box indicates RORB/FOXP1-expressing neurons reduced in DS, and gray asterisks indicate $P_{adj}$ < 0.10. See also Extended Data Figs. 3–6 and Supplementary Tables 2 and 3.

also showed reduced nuclear immunoreactivity in tissue sections (Fig. 1f and Extended Data Fig. 2f,g), validating downregulation on the protein level.

Altered expression of the majority of these genes was confirmed with Nebula, an alternative cell-level differential expression analysis approach[20], and recapitulated in published bulk RNA sequencing (RNA-seq) data from adult cortex and induced pluripotent stem cell (iPSC)-derived neurons[12] (Extended Data Fig. 5 and Supplementary Table 3).

To pinpoint the timing of these changes, we separately subsetted excitatory cells from early (PCW11–13) and late-stage samples (PCW16–20) (Extended Data Figs. 4 and 6 and Supplementary Table 3). Although a trend toward reduced *RORB/FOXP1*-expressing neurons was present already at PCW11–13, significant reductions of *RORB/FOXP1*- and *TLE4*-expressing neurons were observed only at PCW16–20 (Extended Data Fig. 4c). Some 472 genes were already differentially expressed at PCW11–13 (Extended Data Fig. 6a,b and Supplementary Table 3), including 75 Chr. 21 genes and genes linked

to neurodevelopmental programs (GO terms 'axonogenesis', 'dendrite development') and excitatory synaptic signaling (GO terms 'regulation of trans-synaptic signaling', 'ionotropic glutamate receptor signaling pathway').

To further validate and characterize these early dysregulated programs, we performed bulk RNA-seq on a representative subset of samples from PCW11–14 (Extended Data Fig. 6c–f and Supplementary Table 3). This analysis confirmed that an average of 27.5% of genes significantly altered in the snRNA-seq analysis show concordant changes in bulk RNA, consistent with previous work[21]. This increases to an average of 46.8% for upregulated genes, reaching more than 70% in two clusters. Importantly, the analysis also revealed additional DEGs not previously detected (Extended Data Fig. 6c–f and Supplementary Table 3). These included Chr. 21 genes implicated in DS-related phenotypes—such as *DSCAM* and *SOD1*[8,22]—as well as several largely uncharacterized noncoding RNAs, including *CEROX1*, which has been reported to regulate mitochondrial activity[23], a pathway known to be altered in DS[24].

In the PCW16–20 samples, 307 differential genes were detected (including 54 on Chr. 21), predominantly in the major *RORB/FOXP1*-expressing neuronal population (NEU_RORB_s1), consistent with greater heterogeneity at later developmental stages. Genes were enriched for GO terms such as 'forebrain development', 'regulation of neuron projection development' and 'gliogenesis' (Extended Data Fig. 6g,h and Supplementary Table 3).

Overall, this indicates that before PCW11–13, Ts21 perturbs transcriptional programs regulating excitatory neuron development and function. This leads to a selective deficit of *RORB/FOXP1*-expressing subtypes that may stem from impaired generation, maturation or increased vulnerability[16], and contribute to later neurological phenotypes in DS.

### Integrated gene-regulatory network analysis predicts key mediators contributing to the deregulation of transcriptional programs downstream of Chr. 21 genes

We next asked how the increased gene dosage of Chr. 21 genes might cause the observed transcriptional alterations. Because cell type-specific chromatin accessibility shapes TF-mediated gene regulation and is altered in DS by Chr. 21 chromatin remodelers such as *BRWD1* and *HMGN1*[25,26], we integrated single-cell ATAC sequencing and RNA-seq data using scMEGA[27] to predict deregulated *cis*-regulatory elements and TFs (Methods and Fig. 3a). Predicted TF–*cis*-regulatory element interactions were supported by chromatin immunoprecipitation sequencing (ChIP-seq) data, and TF interactions with Chr. 21 genes were inferred from experimentally validated protein–protein interaction (PPI) datasets (Methods and Fig. 3a).

Harnessing the main excitatory populations from the whole dataset (PCW10–20), scMEGA revealed a strong correlation of gene expression with chromatin accessibility at gene loci, including putative *cis*-regulatory elements (Fig. 3b,c). scMEGA predicted 6,299 interactions of 30 TFs with putative *cis*-regulatory elements of 353 DEGs with dynamic expression and accessibility along the excitatory lineage trajectory (Supplementary Table 4). Of these, 3,722 interactions were consistent with roles in determining the differences between DS and CON; that is, for predicted positive interactions, the regulator and its target gene were both upregulated in DS or both downregulated, whereas for negative interactions, the regulator was upregulated in DS when the target was downregulated, or vice versa (Fig. 3a). Many of these interactions were predicted to regulate genes associated with enriched GO terms related to neural development (Fig. 3d) and included TFs that are well-characterized regulators of neuronal subtype specification and maturation, such as *FEZF2*, a key regulator of lower layer cortical excitatory neuron specification[28,29], *TCF7L2*[30–32], *RORA*, which is closely related to the cortical L4 excitatory neuron marker *RORB* and required for dendritic maturation of these cells[33], and *FOXP1*, both enriched in the neuronal population reduced in DS (Fig. 1). Importantly, the network also included three Chr. 21 TFs, which have been implicated in mitochondrial function and stress responses, but whose roles in DS are not well understood: *BACH1*[34], *GABPA*[35] and *PKNOX1*[36,37]. The Chr. 21 TFs were predicted to directly regulate several key network nodes implicated in neural development, including *FEZF2*, *FOXP1*, *RORA* and *TCF7L2* (Fig. 3e). Remarkably, the predicted Chr. 21

TF target genes were significantly enriched for genes with known mutations causing intellectual disability syndromes, as cataloged in the Genomics England PanelApp database[38] (84 of 312 targets; odds ratio 2.0, $P = 2.1 \times 10^{-7}$, Fisher's exact test), including TFs such as *FOXP1*, *TCF7L2*, *SOX9* and *MEF2C*, and effector genes including *FGFR2*, *NRXN3*, *NOTCH2* and the potassium channels *KCNH5* and *KCNQ3* (Fig. 3f).

Because not all TFs bind to all accessible predicted binding sites, we compared our predictions with experimentally validated TF binding sites from ChIP-seq data from various human cell types in the ChIP-Atlas database[39]. This showed a significant enrichment of binding of 27 predicted upstream TFs to regulatory elements of their targets (Methods, Extended Data Fig. 7a and Supplementary Table 4), providing an experimental validation of 1,419 predicted interactions (20%–80% of all predicted interactions for most TFs), including several potentially important examples such as *PKNOX1* targeting *FEZF2*, *BACH1* targeting *SMAD3* and *SOX2*, and *GABPA* targeting *SOX9*.

Separate analyses of early- and late-stage samples suggest that the TF networks are highly dynamic and context dependent (Extended Data Fig. 7b–e). Although *PKNOX1* and *GABPA* emerged as key network nodes at both stages, their predicted targets differed; for instance, *PKNOX1* was linked to *FEZF2* in PCW11–13 samples and to *FOXP1* in PCW16–20 samples.

Many Chr. 21 genes implicated in DS are not TFs but may modulate the TF network through PPIs. Using experimentally validated BioGRID PPIs, we assessed whether expression of differentially expressed Chr. 21 proteins correlated with TF activity along the excitatory lineage trajectory (PCW10–20) (Methods, Extended Data Fig. 8 and Supplementary Table 4). We identified 783 significant Chr. 21 protein-TF pairs, with 186 interactions, spanning 35 Chr. 21 proteins and 22 TFs, consistent with altered TF activity in DS. Notably, this analysis identified DYRK1A, one of the best-characterized mediators of DS phenotypes, as a central regulator of the TF network, influencing FEZF2, FOXP1, ESR2 and PAX3, with additional modulation by APP and the chromatin remodeler BRWD1 affecting TFs including FEZF2, GLI3, ESR2 and FOXP1. USP25, a deubiquitinase recently implicated in DS-related intellectual disability[40] may also regulate FOXP1, EBF3, BNC2 and IRF2. Only a few of the identified interactions were direct (for example, APP–FEZF2), while many were mediated by transcriptional or epigenetic co-regulators such as CREBBP (CBP) or TAF1 (Extended Data Fig. 8b and Supplementary Table 4).

Together, these analyses implicate the Chr. 21 TFs BACH1, PKNOX1 and GABPA as key regulators of cortical transcriptional programs, acting directly on intellectual disability-associated genes and neurodevelopmental TFs, including *FEZF2*, *FOXP1*, *TCF7L2* and *RORA*, with additional modulation by Chr. 21 genes such as *DYRK1A*, *APP*, *BRWD1* and *USP25* via PPIs.

### Altered transcriptional programs and predicted Chr. 21 TF targets in the developing DS cortex are partially recapitulated in vitro and rescued by TF modulation

To further validate the mechanisms identified here, we initially assessed to what extent experimentally accessible iPSC-based models across different differentiation stages and genetic backgrounds recapitulate human fetal cortex development and DS-associated changes.

**Fig. 3 | Integrated gene-regulatory network analysis predicts key mediators contributing to the deregulation of transcriptional programs downstream of Chr. 21 genes. a**, Approach to identify key regulators of transcriptional programs altered in DS. **b**, Excitatory lineage trajectory defined by scMEGA for network modeling. **c**, Chromatin accessibility versus gene expression along the scMEGA trajectory, identifying dynamically accessible putative *cis*-regulatory elements indicating TF activity and determining gene expression. **d**, TFs predicted to regulate DEGs linked to altered neural functions. Heatmap showing number of interactions between TFs and differential genes linked to selected enriched GO terms. The total number of targets per TF is shown in parentheses. GO terms

and predicted key regulators discussed in the main text are highlighted by grey dotted outlines. **e**, Network plot showing predicted interactions between TFs regulating DEGs (the number of TF–target interactions is given in parentheses). Node size indicates the relative expression in CON samples (vst-normalized) and node color indicates the relative expression (*z*-score) in DS versus CON (each mean of all cell clusters). **f**, Predicted direct targets of Chr. 21 TFs *BACH1*, *PKNOX1* and *GABPA* with known mutations causing intellectual disability syndromes (from Genomics England PanelApp[38]). See also Extended Data Figs. 7 and 8 and Supplementary Table 4.

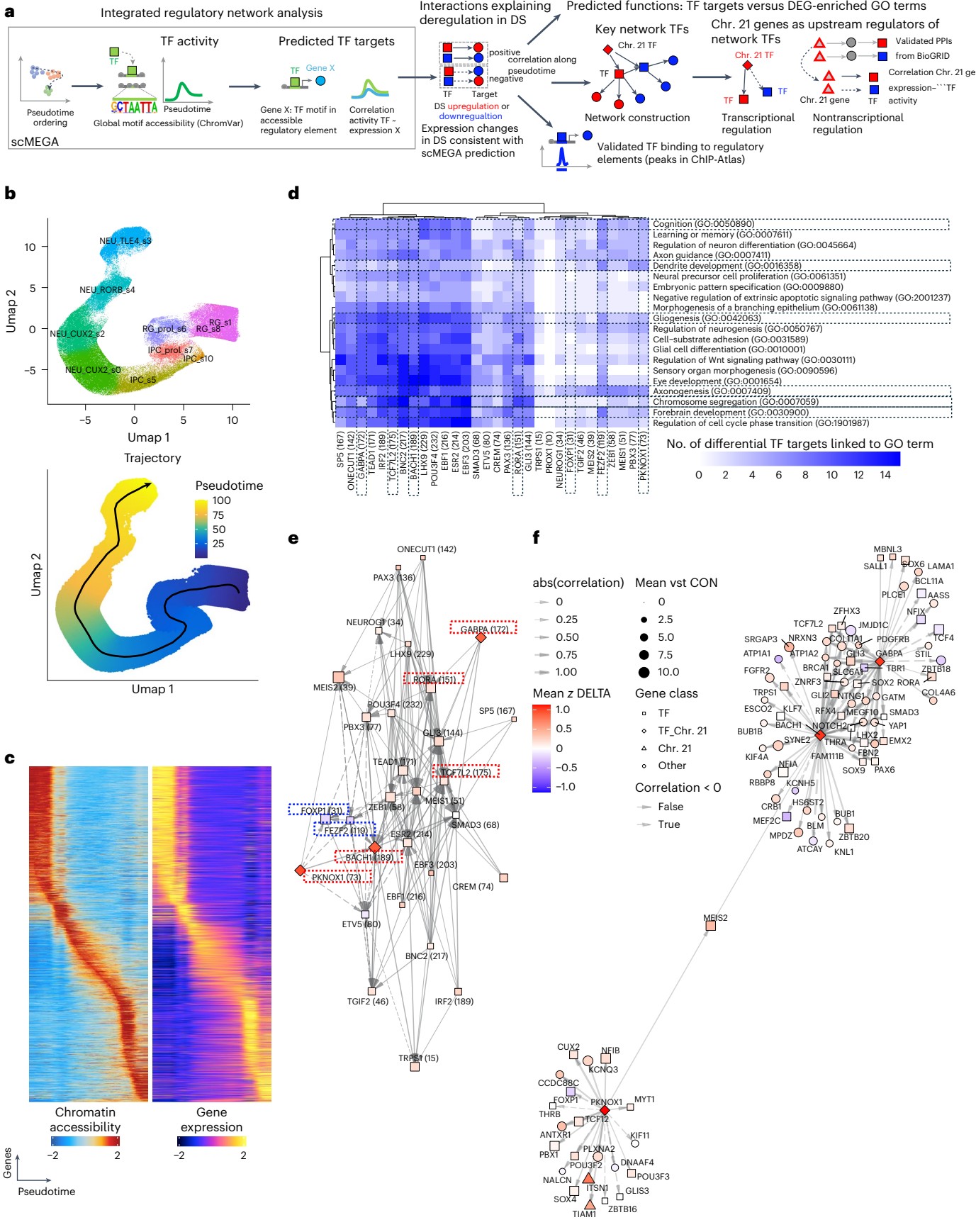

We differentiated multiple batches of neural progenitors and neurons from two pairs of trisomic iPSC lines (named DS1, C13) from individuals with DS and corresponding isogenic disomic control lines (DS2U and C9, respectively)[41–43], and performed bulk RNA-seq (Fig. 4a and Methods). Gene expression in cultures of in vitro neural progenitors (iNPCs) and neurons (iNEUs) strongly correlated with neural progenitor cells (NPCs; radial glia (RG), intermediate progenitor cell (IPC) populations) and neurons in the fetal cortex (Extended Data Fig. 9a), confirming successful differentiation. We detected each ~2,000–4,000 upregulated and downregulated genes in both iNPCs and iNEUs from both pairs of iPSC lines, including ~80–100 mostly upregulated Chr. 21 genes (Extended Data Fig. 9b and Supplementary Table 5), indicating, as expected, lower variability of the side-by-side differentiated isogenic DS and CON NPCs compared to fetal tissue. Up to ~50%–90% of the DEGs (downregulated and upregulated) detected in fetal tissue populations were concordantly altered in NPCs in vitro, and up to ~40%–80% showed concordant changes in neurons, including many genes implicated in forebrain development (Fig. 4b,c). Importantly, these included also *PKNOX1*, *BACH1* and *GABPA*, the Chr. 21 TFs predicted to be critical regulators of neurodevelopmental alterations in DS, as well as many of their putative targets (Supplementary Table 5).

We therefore hypothesized that reducing the increased expression of these TFs in DS NPCs may rescue the dysregulated expression of their target genes and DS-associated molecular phenotypes in vitro. To normalize the elevated expression of Chr. 21 TFs and identify which predicted genes are bona fide downstream targets, we developed an antisense oligonucleotide (ASO)-based approach[44]. We designed ASOs to downregulate each TF by targeting their messenger RNAs and established an efficient transfection protocol for iPSC-derived NPCs using fluorescently labeled nontargeting control ASOs (Extended Data Fig. 9c,d). Using a quantitative reverse transcriptase polymerase chain reaction (RT–qPCR), we confirmed the effectiveness of several ASO designs at different concentrations, robustly reducing the elevated TF mRNA levels in DS NPCs close to control levels (Extended Data Fig. 9e). Western blotting for PKNOX1 demonstrated that ASO-mediated treatment also normalized TF protein levels (Extended Data Fig. 9f). Finally, we selected effective ASOs to test their ability to modulate expression of the predicted TF target genes using bulk RNA-seq. In an initial experiment, we confirmed that the exposure to nontargeting ASOs had only minor effects on global gene expression compared to targeting ASOs (Extended Data Fig. 9g). As expected from inherent differences between in vitro NPCs and neural cells in tissue, and the complexity of gene-regulatory network dynamics, only a subset of the predicted targets showed differential expression in DS versus CON NPCs, and a proportion of these exhibited partial normalization following ASO-mediated TF modulation (Fig. 4d). Importantly, the ASO approach reverted or showed trends toward reverting the deregulation of several predicted targets linked to intellectual disability syndromes, including the PKNOX1 targets *MYT1*[45], *SOX4*[46] and *ETV5*[47], eight BACH1 targets, including *HS6ST2*[48], *LIFR*[49],

*SYNE2*[50], *SRGAP3*[51] and *ATP1A1*[52], and eight GABPA targets including *EGR1*[53], *DOCK1*[54], *BCL11A*[45], *MEGF10*[55] and *SLC6A1*[56], as well as previously established regulators of neuronal differentiation, such as the BACH1 and GABPA target *NEUROD1* and the GABPA target *NEUROG2*[57].

Together, these results suggest that iPSC-derived neural cells cultured in vitro partially recapitulate molecular DS phenotypes in the fetal human cortex, and that Chr. 21 TFs *PKNOX1*, *BACH1* and *GABPA* partially drive these phenotypes.

## Transplanted human neural cells reveal DS molecular and cellular phenotypes not recapitulated in vitro and emerging at later stages of fetal development

Our in vitro isogenic stem cell model captured key DS neural phenotypes, although some fetal tissue gene expression patterns differed (for example, *FOXP1* upregulated in DS1 iPSC-derived NPCs (Fig. 4c); many predicted Chr. 21 TF targets unchanged), highlighting both its utility for mechanistic studies and the value of fetal benchmarking. However, understanding the function of the identified regulators will require models that more accurately capture the in vivo environment of the human brain. We therefore tested to what extent our recently established DS human xenograft system[58], which avoids some of the drawbacks of animal models (for example, lack of Chr. 21) and of in vitro conditions (for example, limited neuronal maturation), recapitulates human fetal development and the observed changes in DS. As in our previous work, we transplanted iPSC-derived neural cells into adult mice to minimize their integration with host networks[58], which could otherwise complicate interpretation.

We differentiated a DS iPSC line and its isogenic control into mixed progenitor and neuron cultures and transplanted them into adult mouse brains to mature in vivo for 12–24 weeks. snRNA-seq of these grafts (eight CON, four DS) yielded 98,545 high-quality human nuclei (Fig. 5, Methods and Supplementary Table 6). Mapping these to our fetal tissue dataset showed that grafts largely resembled excitatory fetal neurons (Fig. 5b and Extended Data Fig. 10a,b), with fewer progenitors and more mature RORB-expressing or TLE4-expressing neurons (Fig. 5c), suggesting similarity to later developmental stages.

Notably in DS grafts, we observed more astrocyte-like cells and fewer proliferating progenitors, with a trend toward reduced *CUX2*-expressing neurons (Fig. 5c), consistent with previous reports at later developmental stages[2]. Some 3,290 genes were differentially expressed between DS and CON grafts, mostly in neuronal populations as in fetal tissue (Extended Data Fig. 10c,d). Up to >70% of genes upregulated in fetal cells were concordantly upregulated in the corresponding graft populations, while fewer genes were discordantly regulated than in vitro (Fig. 5d). Many genes dysregulated in fetal DS tissue involved in forebrain development, including *ERBB4*, *ALK* and *FOXP1*, showed consistent changes in grafts but not in vitro (Fig. 5e).

Overall, transplanted cells recapitulate multiple DS molecular and cellular phenotypes, including those emerging at later developmental stages, for which human tissue samples are scarce.

**Fig. 4 | Altered transcriptional programs and predicted Chr. 21 TF targets in the developing DS cortex are partially recapitulated in vitro and rescued by TF modulation. a**, Experimental approach for modeling DS neurodevelopment and normalizing Chr. 21 TF expression in vitro. **b**, Fraction of differential genes per tissue population also detected in vitro. Red and blue boxes indicate genes upregulated or downregulated both in fetal tissue and in vitro. **c**, Expression changes in DS versus CON for genes differentially expressed in fetal tissue linked to the GO term 'forebrain development'. DESeq2 analysis with the likelihood ratio test (LRT) was used to assess group effect across paired DS versus CON technical and biological replicates (between three and ten RNA samples from wells of paired side-by-side differentiated DS or CON cells per condition from *n* = 6, 2, 3 and 1 independent differentiation experiments for iNPC_C9_C13, iNPC_DS2U_DS1, iNEU_C9_C13 and iNEU_DS2U_DS1, respectively; Methods and Supplementary Table 5). The heatmap shows the difference in mean *z*-scores

between DS and CON samples for each cell line and differentiation stage, and for each tissue population pseudobulk (excitatory lineage, PCW10–20). Red dotted boxes indicate regulators discussed in the main text. *Benjamini–Hochberg adjusted *P* < 0.10. **d**, Expression changes of predicted Chr. 21 TF targets deregulated in DS-derived NPCs upon Chr. 21 TF 100 nM ASO treatment. DESeq2 analysis for cultures with LRT was used to assess the ASO effect across technical and biological replicates (7–16 RNA samples from ASO-treated and untreated wells of paired side-by-side differentiated C13 (DS) cells per condition from each of *n* = 5 independent differentiation experiments; Methods and Supplementary Table 5). Red and magenta dotted boxes denote examples of dysregulated targets rescued by ASO treatment, including ID-linked genes (red) or other key neurodevelopmental regulators (magenta). *Benjamini–Hochberg adjusted across predicted TF targets *P* < 0.10. #Trend with nominal *P* < 0.1.

## Discussion

Our study provides insight into early alterations in fetal cortical development that may contribute to intellectual disability in DS, one of the most common congenital causes of lifelong disability. We discovered

a reduction in RORB-expressing or FOXP1-expressing L4-like excitatory neurons, mirroring adult DS[11], suggesting impaired generation or maturation rather than neurodegeneration as observed in Alzheimer disease[16]. Many of the approximately 700 genes altered in the excitatory

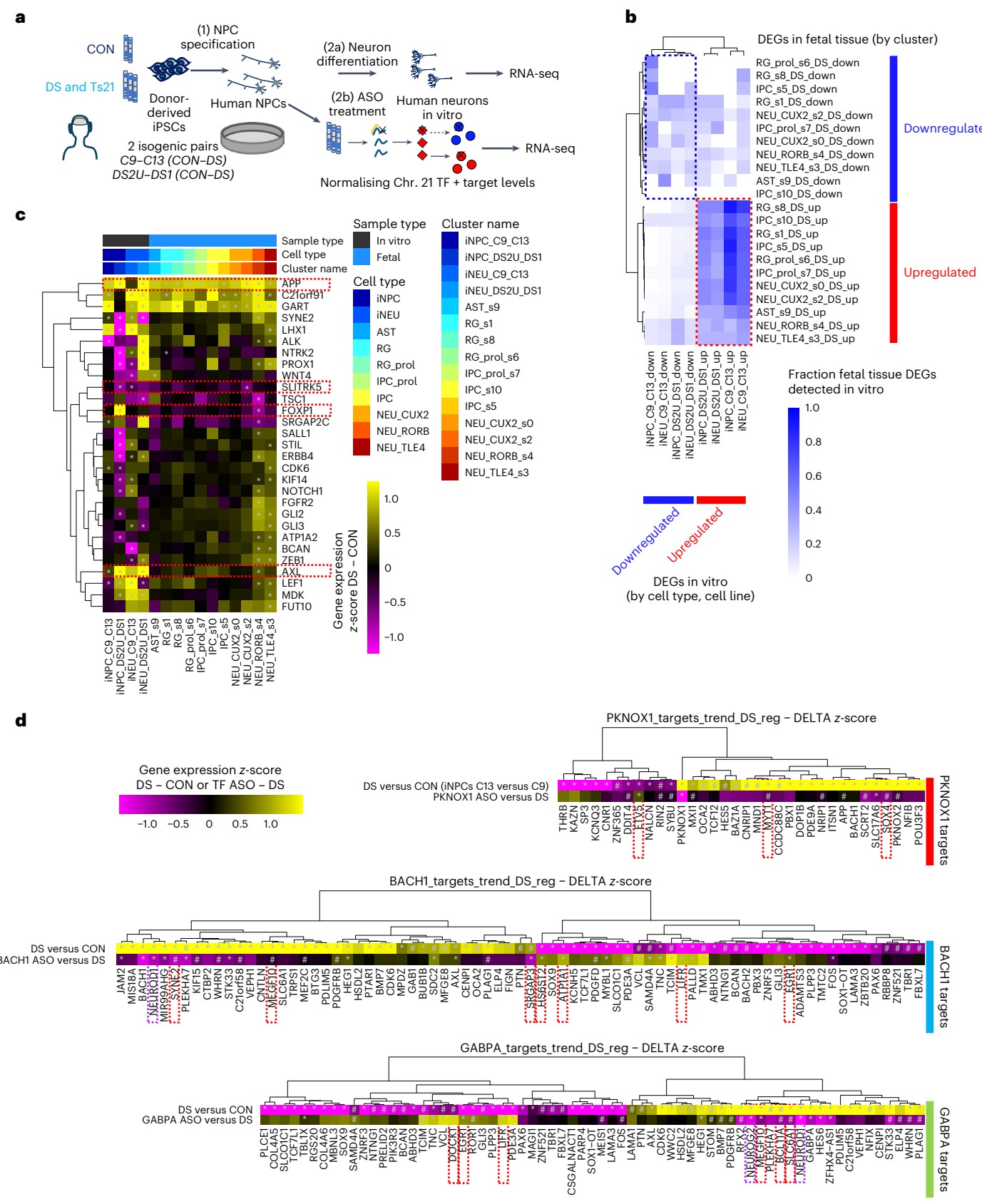

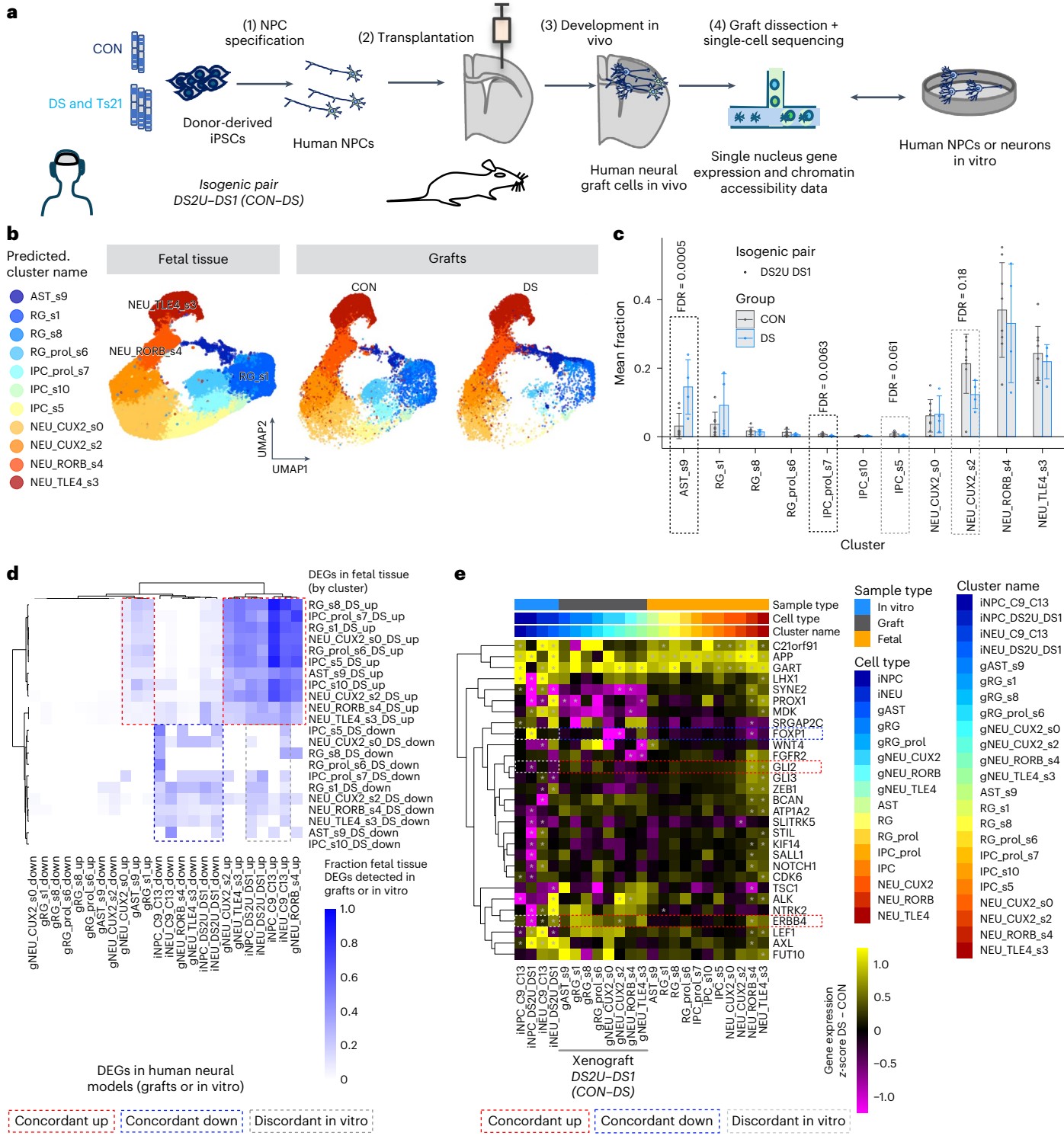

**Fig. 5 | Transplanted human neural cells reveal DS molecular and cellular phenotypes not recapitulated in vitro and emerging at later stages of fetal development. a**, Experimental approach for modeling DS neurodevelopment in vivo. **b**, Mapping of CON and DS excitatory lineage transplanted cells to fetal tissue populations (UMAP plot). **c**, Cell abundance in CON and DS transplants. Barplot showing individual samples (*n* = 8 for CON and *n* = 4 for DS), mean ± s.d. and FDR for sccomp compositional analyses[71] (other clusters FDR > 0.05). Dotted boxes highlight selected populations: FDR < 0.05 (black) or FDR > 0.05 (gray). **d**, Fraction of differential genes per cluster in fetal tissue also detected in grafts and in vitro. Comparisons showing concordant changes in fetal tissue versus models, and fraction of discordantly regulated genes in vitro versus fetal tissue are highlighted. DESeq2 analysis for grafts with LRT test by cluster

with correction for sequencing technology, threshold $P_{adj}$ < 0.10. **e**, Expression changes in DS versus CON for genes differentially expressed in fetal tissue and linked to the GO term 'forebrain development'. The heatmap shows the difference in mean *z*-scores between DS and CON samples (corrected for sequencing technology) for merged bulk data from in vitro cultures (three to ten RNA samples from wells of paired side-by-side differentiated DS and CON cells per condition from *n* = 6, 2, 3 and 1 independent differentiation experiments for iNPC_C9_C13, iNPC_DS2U_DS1, iNEU_C9_C13 and iNEU_DS2U_DS1, respectively; Methods and Supplementary Table 5) and for each cluster pseudobulk from the graft and fetal tissue analyses (excitatory lineage, PCW10–20). Gray asterisks indicate $P_{adj}$ < 0.10. See also Extended Data Fig. 10 and Supplementary Table 6.

lineage are involved in forebrain development, neuronal subtype specification and intellectual disability, representing some of the earliest phenotypes associated with DS. Many overlap with genes deregulated in adult cortex[12], suggesting that these changes persist into adulthood. Our resource complements previous single-cell analyses of typical brain development[15,59–63] from this critical period.

Despite the increased dosage of more than 200 Chr. 21 genes, only subtle changes were observed in cortical cells, indicating that multiple small-effect alterations act synergistically on shared pathways rather than converging on obvious individual gene targets. Modulating upstream TFs may offer a coordinated approach to target such pathway-level alterations[64]. Notably, three Chr. 21 TFs, *PKNOX1*, *BACH1* and *GABPA*, emerged as potential master regulators, targeting TFs involved in excitatory neuron specification (for example, *FEZF2*[28,29], *FOXP1*[19] and *RORA*[33,65]), suggesting a mechanism underlying the selective deficit in *RORB*/*FOXP1*-coexpressing excitatory neurons, and more than 80 genes mutated in intellectual disability syndromes. Our analysis aligns with recent studies implicating these TFs and their targets in altered neurodevelopment in DS[66,67] and independently validates known Chr. 21 regulators, including *DYRK1A*[68], *APP*[69], *BRWD1*[25] and *USP25*[40].

Stem cell-derived DS neurons in vitro recapitulated many of these changes, and ASO-mediated normalization of Chr. 21 TFs partially rescued expression of several predicted target genes involved in neuronal differentiation and intellectual disability, suggesting a mechanistic link between Chr. 21 gene dosage and the neurodevelopmental pathology of DS.

Finally, we demonstrated that our previously established xenograft model[58] complements these in vitro findings, despite the heterochronic design. Although not fully recapitulating the reduction in *RORB*/*FOXP1*+ neurons observed in primary tissue, likely because of challenges in generating upper-layer diversity and mapping across maturational stages, it captured key DS cellular and molecular phenotypes, including late developmental features, making it a valuable model for preclinical studies[58,70] and in vivo testing of human-specific ASO tools.

Although we provide a valuable publicly available resource identifying cell type-specific changes and associated gene-regulatory networks, a proof-of-concept ASO-mediated molecular rescue and a benchmarking of human in vitro and in vivo models, several technical and biological limitations should be acknowledged.

Surgical terminations disrupt tissue architecture, limiting resolution of regional organization and increasing variability in cell type composition, which may obscure abundance changes beyond *RORB*/*FOXP1*-expressing neurons. Although we validated this phenotype using FOXP1, including in well-preserved paraffin-embedded human brain sections, unreliable RORB antibodies underscore the need for future validation with additional markers. Variable cell numbers also limit sensitivity for detecting differential expression, especially in bulk RNA-seq and rare single-nucleus populations.

Xenograft benchmarking was limited to a single isogenic pair with imperfectly balanced groups, requiring further validation across additional genetic backgrounds. However, comparison with matched in vitro cultures suggests that the xenograft model more faithfully captures some DS-associated changes.

The predicted regulatory networks are based on correlative and cross-cell-type data and require further functional validation, as we have demonstrated here for targets of the Chr. 21 TFs *PKNOX1*, *BACH1* and *GABPA*, whose dysregulation was partially rescued by ASO-mediated normalization of TF expression levels.

Future work should functionally test whether correcting the deregulation of these TFs can rescue core DS cellular phenotypes, including reduced neural network activity, in vitro and in humanized mice in vivo[58,70]. This includes establishing the efficacy of combinatorial modulation, and whether rescue is possible at postnatal or adult stages, when the three key Chr. 21 TFs remain expressed[11].

In conclusion, this study generates a foundational molecular map of the DS cortex during a critical developmental window. This resource, combined with the in vitro and in vivo platforms we benchmarked, enables the identification and future preclinical validation of candidate regulators, contributing to improved understanding of the neurological symptoms of DS.

## Online content

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

## Methods

### Fetal tissue samples and ethics

Human fetal brain samples were collected from 19 fetuses with Ts21 and 20 euploid control fetuses aged PCW10–20, the latter obtained following elective terminations of pregnancy and likely free of developmental defects, all confirmed by karyotyping through the Human Developmental Biology Resource (HDBR), and with maternal informed written consent. Human fresh-frozen brain tissue was provided by the Joint MRC/Wellcome Trust HDBR (Project Number 200585, supported by Joint MRC/Wellcome Trust grant nos. 099175/Z/12/Z and MR/006237/1; http://www.hdbr.org), in compliance with ethical approval from the National Health Service (NHS) Research Health Authority (HDBR; London/Newcastle; REC approval 18/LO/0822 and 18/NE/0290) and stored at −80 °C. The HDBR is overseen by the UK Human Tissue Authority and operates in compliance with the applicable Human Tissue Authority Codes of Practice. Sample sizes for human fetal tissue were determined and constrained by the availability of this precious donated material. Because the study primarily relies on systematic global computational analyses rather than strong initial hypotheses, systematic blinding was not necessary. We matched sex and developmental stage between CON and DS groups, as far as sample availability allowed (Supplementary Table 1).

Paraffin-embedded, immersion-fixed (4% paraformaldehyde (PFA) in phosphate-buffered saline (PBS), pH 7.4) postmortem human prenatal brain tissue (PCW16–20; Extended Data Fig. 2g) was obtained from the Zagreb Neuroembryological Collection with ethical approval from the Internal Review Board of the Ethical Committee of the University of Zagreb School of Medicine. Procedures followed the Declaration of Helsinki (2000).

Remaining material from tissue samples (available from some samples) will be shared upon reasonable request and subject to conditions of the HDBR.

### Tissue sectioning and immunostaining

Fresh-frozen brain tissue was cut on a cryostat (Leica) in 20-µm sections for immunostaining, mounted on slides, or as 80-µm sections for nuclei extraction, collected in RNase-free low-binding tubes (LoBind, Eppendorf), which were stored at −80 °C for further processing. For immunostaining to identify cortical tissue, sections were briefly thawed and dried (~30 min), before fixation in 4% PFA in PBS for 15 min at 4 °C. Sections were washed with PBS + 0.1% Triton X-100 (PBST), incubated in blocking buffer (PBST + 10% normal goat serum) for 1 h at room temperature and then with primary antibodies overnight in blocking buffer in a humidified chamber at 4 °C. After three washes (PBST), sections were incubated in the dark at room temperature for 2 h with secondary antibodies in blocking buffer, and again washed three times before incubating with DAPI (1 µg ml⁻¹ in PBS) for 45 min. After one wash in PBS, the sections were embedded with ProLong Gold Antifade Mountant (Thermo Fisher, cat. no. P36930), and stored in the dark at 4 °C. Details of primary antibodies used are included in the reporting summary. Secondary antibodies used were anti-mouse Alexa Fluor 488, anti-rabbit Alexa Fluor 555 or anti-rat Alexa Fluor 647 (all raised in goat; Thermo Fisher). Sections were imaged with a Leica SP8 confocal microscope (×10 or ×20 objectives), creating a single Z-plane scan of the whole tissue section using the Leica Application Suite X.

Paraffin-embedded tissue (Extended Data Fig. 1g) was sectioned coronally at 10 µm using a Leica SM2000R microtome. Slides were deparaffinized in xylene (2 × 10 min), rehydrated through graded ethanol (100%, 96%, 70%) and rinsed in PBS. Sections were blocked (1% BSA, 0.5% Triton X-100 in PBS) for 2 h at room temperature. Primary antibodies (details are included in the reporting summary) were applied overnight at 4 °C. After PBS washes, secondary antibodies (anti-rabbit Alexa Fluor 488, anti-rat Alexa Fluor 555; Thermo Fisher) were incubated for 2 h at room temperature in the dark. Sections were treated with TrueBlack to reduce autofluorescence, washed, then

mounted with DAPI-containing Vectashield. High-resolution images were acquired using the Hamamatsu NanoZoomer 2.0 RS scanner with a ×40 (numerical aperture 0.75) objective at 455 nm per pixel. Fluorescence images were captured using the Hamamatsu LX2000 Lightning exciter and an Olympus FV3000 confocal microscope with a ×20 (numerical aperture 0.75) objective, using FV31S-SW Fluoview software at 1,024 × 1,024 resolution.

### Quantification of FOXP1 immunostaining

Images of DAPI, SATB2, FOXP1 and CTIP2 immunostainings were taken with a Leica SP8 (×20 objective) with the same settings for all stained cryosections, creating a single Z-plane scan of the whole tissue section. To quantify nuclear FOXP1 intensity with high throughput and avoid manual counting bias, we developed an automated FIJI–R analysis pipeline (https://github.com/lattkem1/Nuc_fluor_Fiji_R_025). Regions of interest (ROI) of well-preserved cortical plate tissue were manually defined, unaware of genotype and FOXP1 staining, as regions with nuclear staining in the CTIP2 channel (six ROI per section or sample). For each cortical plate ROI, nucleus ROIs were identified in the DAPI channel using automated threshold selection and a FIJI watershed algorithm to separate close nuclei, followed by measuring the mean intensity for each nucleus ROI channel. FOXP1 fluorescence intensity per nucleus was further analyzed using R scripts. Fluorescence intensity per nucleus by experimental group was summarized and plotted as histograms. A threshold of 10,000 AU for positive or negative classification was determined based on these histograms and manual inspection of images, and nuclei from all ROIs of each sample of both groups (DS and CON) were compared using a two-sided t-test. The group difference was confirmed to be robust to different positive or negative thresholds (not shown).

### Human iPSC culture and neural induction

Two pairs of human iPSCs lines from individuals with DS and two corresponding isogenic iPSC lines were used. From WiCell, we acquired the trisomic DS1 line (UWWC1-DS1) and the corresponding isogenic disomic line DS2U (UWWC1-DS2U)[43]. The trisomic line C13 (DS) and isogenic control C9 (CON) were previously generated and described[41,42].

iPSCs were maintained on six-well culture plates (coated with Matrigel (Corning)) in mTeSR Plus medium supplemented with 0.5 µM Thiazovivin (Tocris). Media changes were performed the next day with complete mTeSR Plus medium (STEMCELL Technologies) and then refreshed every other day.

For most in vitro experiments, adherent cultures of NPCs were derived from iPSCs using the Gibco protocol (Thermo Fisher Scientific, cat. no. MAN0008031) and used between passages six and ten (adapted by D. Nizetic and colleagues, protocol 'DN' in Supplementary Table 5). NPCs were expanded in Geltrex-coated six-well culture plates prepared by diluting a 60-µl Geltrex aliquot in 6 ml of cold DMEM–F12, incubated at 37 °C for at least 60 min. Cells were thawed, centrifuged at 300g for 5 min and resuspended in Neural Expansion Medium (NEM) with 5 µM ROCK inhibitor Y27632 (STEMCELL Technologies) at a density of 1–2 million cells per well. Media changes were performed the next day with complete NEM and then changed every other day thereafter.

For most ASO experiments, including validation of ASO efficacy using RT–qPCR (Extended Data Fig. 9e and Supplementary Table 5), NPCs were differentiated from human iPSCs following a previously published protocol[72] (protocol LI in Supplementary Table 5). Briefly, human iPSCs were dissociated into single cells and plated at a density of 30,000 cells per cm² in neural induction medium (NIM), composed of DMEM–F12 and NeuroBasal (1:1) supplemented with 1% N2, 2% B27, 1% penicillin–streptomycin, 1% GlutaMax, 10 ng ml⁻¹ human leukemia inhibitory factor and 5 µg ml⁻¹ bovine serum albumin. The medium was further supplemented with 4 µM CHIR99021 (Tocris), 3 µM SB431542 (Sigma) and 0.1 µM Compound E (Millipore) for 7 days. Cultures were

subsequently passaged at a 1:3 ratio for five passages using Accutase, and maintained in NIM without Compound E on Matrigel-coated plates.

For xenotransplantation experiments, and the in vitro bulk RNA-seq analysis using the DS2U and DS1 lines (Supplementary Table 5), NPCs were generated using a neurosphere-based protocol developed previously[73] (SCZ in Supplementary Table 5). iPSCs were transitioned to neural differentiation medium (NDM), consisting of a 1:1 mixture of DMEM–F12 (Thermo Fisher Scientific) and Neurobasal medium (Thermo Fisher Scientific) supplemented with 1× GlutaMAX (Thermo Fisher Scientific), 0.5× N2 (STEMCELL Technologies), 0.5× B27 (STEMCELL Technologies) and 100 µM ascorbic acid (Sigma-Aldrich). On the next day, dual SMAD inhibition was initiated by supplementing NDM with 10 µM SB431542 (Tocris) and 2 µM DMH1 (Tocris). Cells were cultured for 7 days with daily medium changes. Cells were then dissociated with Versene and transferred to low-attachment flasks in NDM supplemented with 10 ng ml$^{-1}$ basic fibroblast growth factor (bFGF; STEMCELL Technologies) and 0.5 µM Thiazovivin to promote neurosphere formation. Neurospheres were maintained with medium changes every 3 days until day 25–29, after which they were prepared for transplantation and simultaneously plated for an additional 30 days in culture to allow comparison of the same cells in vivo and in vitro by RNA-seq.

## Cortical neuron differentiation

Twenty-four-well culture plates, with or without coverslips, were coated with 300 µl poly-L-ornithine (Sigma-Aldrich) per well and incubated overnight at 37 °C, followed by coating with 20 µg ml$^{-1}$ laminin (Thermo Fisher Scientific) for at least 2 h at 37 °C.

For differentiation from adherent iPSC-derived NPCs, NPCs were dissociated with Accutase (Thermo Fisher Scientific) and seeded onto poly-L-ornithine and laminin-coated 24-well culture plates at a density of 70,000 cells per well in NEM with 5 µM Y27532. Media change was performed the next day with complete BrainPhys Neuronal Medium (STEMCELL Technologies) and then changed every 4 days for 2 weeks. After which, BrainPhys Neuronal Medium was supplemented with 2 µg ml$^{-1}$ Laminin for media changes every 4 days until 30 days of differentiation.

For differentiation from day 25–29 neurospheres[73] (iPSC lines DS2U, DS1; Supplementary Table 5), neurospheres were dissociated using TrypLE (Thermo Fisher Scientific) and seeded onto poly-L-ornithine and laminin-coated 24-well culture plates at a density of 70,000 cells per well in neuron medium, composed of neurobasal medium supplemented with 1× GlutaMAX, 1× B27 with vitamin A (STEMCELL Technologies), 10 ng ml$^{-1}$ brain-derived neurotrophic factor (Thermo Fisher Scientific), 10 ng ml$^{-1}$ glial cell line-derived neurotrophic factor (Thermo Fisher Scientific), 1 µM dibutyryl cyclic AMP (STEMCELL Technologies) and 200 µM ascorbic acid. Compound E (0.1 µM; Merck Millipore) was added on the first day of plating and withdrawn during subsequent medium changes. Neuronal cultures were maintained with medium changes every 3–4 days in neuron medium for the first 2 weeks and partial BrainPhys medium changes until 30 days of differentiation.

## Human iPSC-derived cortical neuron transplantation and graft extraction

All mouse experimental procedures were approved by the Animal Care and Use Committee (IACUC) at Duke-NUS, Singapore (ref. no. 2022/SHS/1766), and ethics approved by the Institutional Review Board (NUS-IRB-2022-149), as well as by the Ministry of Health (MOH ref. RR-2023/01) under the Human Biomedical Research Act guidance. Immunodeficient mice (NOD.Cg-PrkdcScid;Il2rgtm1Wjl/SzJ, JAX NSG) ($n$ = 17, 4 females and 13 males) aged between 3 and 5 months were kept to a 12-h light/dark cycle, a temperature of approximately 22 °C and a relative air humidity of approximately 50%. Mice were given 5% isoflurane mixed with oxygen as induction anesthesia.

Craniotomies were performed over the right somatosensory cortex, as previously described[74].

Seven days before injection, DS1 and DS2U neurospheres were dissociated into small spheres or cell clumps, and ~2 × 10$^6$ cells were seeded into an upright T25 flask with NDM plus 10 ng ml$^{-1}$ bFGF, then transduced with 5 µl of lentiviral vector expressing GFP under the human Synapsin-1 promoter. On the next day, medium was replaced with fresh bFGF-containing NDM. On the transplantation day, neurospheres were dissociated with TrypLE, washed with Cortex buffer, resuspended at 1 × 10$^5$ cells per µl, and 1 µl was injected with a glass needle using a microsyringe pump (UMP-3, World Precision Instruments), at the following stereotactic coordinates: anterior–posterior = −1.8 mm, medial–lateral = +2.8 mm, dorsal–ventral = −0.6 mm from bregma. A 5-mm diameter glass coverslip was placed over the craniotomy and sealed with cyanoacrylate tissue adhesive. The exposed skull was covered in dental cement and a metal plate placed on the left side of the skull, for positioning and monitoring the human cell transplant at the two-photon microscope. Grafts were analyzed after 12–24 weeks of maturation in vivo.

For extraction of grafts for single nuclei sequencing, mice were killed by cervical dislocation. The whole brain was extracted and immediately placed in ice-cold Cortex buffer (125 mM NaCl, 5 mM KCl, 10 mM glucose, 10 mM HEPES, 2 mM CaCl$_2$, 2 mM MgSO$_4$). All steps from here on were conducted on ice. The hemispheres were dissected, the right-side cortex was removed from the midbrain, and the hippocampus was removed to reveal the underside of the graft. Directed by the fluorescence of the GFP-labeled human cells, a small (2–5 mm in diameter) square containing the graft was dissected using a scalpel before the mouse tissue was removed by carefully tearing along the edge of the graft using fine forceps. The extracted graft was placed in a low-adhesion Eppendorf tube before being flash frozen in liquid nitrogen. Samples were stored at −80 °C until further processing.

## Nuclei isolation for single-nucleus Multiome analysis (RNA-seq and ATAC-seq)

In total, 10–50 mg of fetal or graft tissue were processed using a protocol based on ref. 75. All steps were performed on ice or at 4 °C with prechilled RNase-free buffers and tools, and up to four samples were processed in parallel.

After removal from dry ice, tissue was immediately suspended in homogenization buffer (10 mM Tris–HCl pH 7.4, 320 mM sucrose, 3 mM CaCl$_2$, 3 mM MgCl$_2$), supplemented freshly with 0.1% NP-40, 1 mM DTT and 1 U µl$^{-1}$ RNAse inhibitor ((Protector; Sigma cat. no. 3335402001), RiboLock (Thermo Fisher, cat. no. PN-EO0382), or RNaseOUT (Thermo Fisher, cat. no. 10777019); Supplementary Table 1). The tissue was then immediately homogenized using a 1-ml dounce homogenizer and after exactly 5 min diluted with 1 volume of homogenization buffer (without NP-40), filtered (30 µm mesh size) to remove large debris, and centrifuged (500$g$, 5 min, 4 °C) to collect raw nuclei. Raw nuclei were resuspended in homogenization buffer (without NP-40) and mixed with 1 volume of 50% iodixanol buffer (10 mM Tris–HCl pH 7.4, 3 mM CaCl$_2$, 3 mM MgCl$_2$, 1 mM DTT, 0.5 U µl$^{-1}$ RNAse inhibitor). This suspension was carefully overlaid on a 29% iodixanol buffer (as above +160 mM sucrose), and centrifuged (6,000$g$, 30 min, 4 °C). The nuclei pellet was resuspended in a lysis buffer (10 mM Tris–HCl pH 7.4, 10 mM NaCl, 3 mM MgCl$_2$, 1% BSA, 1 mM DTT, 0.5 U µl$^{-1}$ RNAse inhibitor, 0.1% Tween-20, 0.1% NP-40, 0.001% digitonin), and after exactly 2 min nuclei were diluted in 10 volumes of wash buffer (lysis buffer without NP-40 and digitonin) and centrifuged (500$g$, 5 min, 4 °C) to collect the nuclei. The nuclei were then resuspended in 1× Nuclei Buffer (from the Chromium Next GEM Single Cell Multiome ATAC + Gene Expression Reagent Bundle; 10X Genomics PN-1000283/PN-1000285; supplemented with 1 mM DTT, 0.5 U µl$^{-1}$ RNAse inhibitor).

## Multiome library preparation and sequencing

snRNA-seq and snATAC-seq libraries were prepared from isolated nuclei by the NIHR Imperial BRC Genomics Facility using a 10X Genomics Chromium X and the Chromium Next GEM Single Cell Multiome ATAC + Gene Expression Reagent Bundle (10X Genomics PN-1000283/PN-1000285) according to manufacturer's instructions. Libraries were sequenced using Illumina NextSeq 2000 or NovaSeq 6000 sequencers.

## Basic processing and quality control of Multiome data from fetal tissue

Raw demultiplexed sequencing data (fastq files) were mapped to the human genome (GRCh38) and quantified using cellranger-arc (v.2.0.2; 10X Genomics) and loaded into an R environment (R v.4.3.3), using the Seurat single-cell analysis package v.5.1.0[76,77], with the Signac extension (v.1.13.0) for analyzing single-nucleus ATAC data[78]. To retain only high-quality datasets and cells, low-quality nuclei and potential nuclei clumps were removed; that is, nuclei with low or extremely high transcript counts (<500 or >30,000 Unique Molecular Identifiers (UMIs) per cell), high counts of mitochondrial genes (>2%), low or extremely high numbers of mapped chromatin fragments (<100 or 25,000 ATAC counts per cell), or poor or unspecific chromatin fragmentation (nucleosome_signal >2 or transcriptional start site enrichment <1.1). Datasets with a high fraction of low-quality cells (>50%) or fewer than 500 retained cells were considered as low-quality datasets and completely removed. In cases of tissue samples with adequate tissue quality, that resulted in low-quality datasets, Multiome sequencing was repeated with a second tissue aliquot. Overview of all removed and retained datasets (Supplementary Table 1).

## Data integration, dimensionality reduction and mapping to the reference atlas

To account for technical variability such as sequencing depth and batch effects, RNA counts per cell were normalized using the function (with parameters) SCTransform(ncells = 3000, variable.features.n = 2000, conserve.memory = TRUE). Principal components of the normalized RNA counts were calculated with RunPCA() and used to integrate the individual sample datasets using IntegrateLayers(method = HarmonyIntegration, assay = "SCT", orig.reduction = "pca"). As dimensionality reduction for visualization of cell populations based on transcriptome similarity, UMAP was performed using RunUMAP(reduction = "harmony", dims = 1:30, return.model = TRUE) and transcriptome similarity neighborhoods were detected using FindNeighbors(reduction = "harmony", dims = 1:30).

The dataset was then mapped to the reference atlas from ref. 15. The processed count matrix and cell-level metadata for the complete dataset from ref. 15, were downloaded from https://cells.ucsc.edu/dev-brain-regions/wholebrain/ (files meta.tsv, exprMatrix.tsv.gz; accessed 17 November 2023) and imported into Seurat. Cells with <750 UMI or >10% mitochondrial reads were removed, and the dataset was split by samples and processed as described above. Transfer anchors were generated using FindTransferAnchors(reference = seur_ref, query = seur, dims = 1:30, reference.reduction = "pca"), followed by mapping and label transfer with MapQuery(anchorset = anchors, reference = seur_ref, query = seur, refdata = list(cell_cluster = "cell_cluster", cell_type = "cell_type", area = "area"), reference.reduction = "pca", reduction.model = "umap").

Predicted cluster and cell type and area assignment were projected on UMAP dimensionality reductions (dataset randomly subsampled to 100,000 cells for plotting to avoid excessive plot sizes). The fraction of cells of each sample mapping to each of the brain areas of the reference dataset was plotted as heatmap, which identified three samples with high mapping to noncortical regions (Extended Data Fig. 1e), which were removed from all following analyses.

The remaining samples were reintegrated, followed by dimensionality reduction and neighborhood detection as above.

## Identification of cell populations and differential abundance testing in fetal Multiome dataset

To identify transcriptionally similar cell populations at different resolutions, clustering was performed FindClusters(algorithm = 1) with different resolution parameter values (range 0.3–1.5). Cluster assignment at different resolutions, sample metadata and expression of selected cell type markers were projected on UMAP dimensionality reductions (dataset randomly subsampled to 100,000 or 10,000 cells for plotting to avoid excessive plot sizes). Based on the mapping to the reference atlas and a curated set of cell type markers, cell clusters at a final resolution of 0.5 were assigned to cell types and labeled accordingly (Fig. 1c,d and Extended Data Fig. 2a).

Changes in cellular composition were assessed with the sccomp package (v.1.7.15)[71], using the functions sccomp_estimate(formula_composition = ~group,.sample = sample,.cell_group = cluster_name, bimodal_mean_variability_association = TRUE, cores = 8) and sccomp_test(). As an alternative approach for cluster-free differential abundance analysis, the MiloR package (v.1.10.0)[17] was used. For this, the Seurat object was converted to a SingleCellExperiment object (as.SingleCellExperiment()), followed by neighborhood detection with buildGraph(k = 50, d = 30) and makeNhoods(prop = 0.05, k = 50, d = 30, refined = TRUE) and neighborhood abundance quantification and testing with countCells(samples = "sample"), calcNhoodDistance(d = 30) and testNhoods(design = ~group). For each cell, membership in any differential neighborhood was identified, and the maximum |log$_2$(FoldChange)| of any associated neighborhood mapped onto the UMAP dimensionality reduction plot.

For analyses of the excitatory lineage, the Seurat object of the full dataset was subsetted to include only cells of samples from the relevant stages, and cell clusters of astrocytes (AST), progenitors (RG, IPC) and excitatory lineage neurons (NEU_CUX2, NEU_RORB and NEU_TLE4). This subsetted dataset was reintegrated and re-clustered as above (final clustering with resolution = 0.3 for whole dataset PCW10–20 and PCW11–13 and 0.5 for PCW16–20).

Throughout the analyses, as TFs we defined the 1,672 genes from a curated list from the Fantom5 consortium (https://fantom.gsc.riken.jp/5/sstar/Browse_Transcription_Factors_hg19, retrieved 21 December 2021). As Chr. 21 genes we defined the 210 protein-coding genes annotated with their HUGO Gene Nomenclature Committee (HGNC) symbol in the Ensembl database (Homo_sapiens.GRCh38.105.chromosome.21_220405.gff3, accessed 5 April 2022).

## Differential gene expression analysis

For differential gene expression analysis, for all cells of each sample and cell cluster, transcript (UMI) counts for each gene were aggregated into pseudobulk samples. To avoid spurious results due to low numbers of UMI counts due to low cell numbers, pseudobulks with fewer than ten cells, and very low expressed genes with a less than average 0.1 UMI count per cell in all clusters were removed from further analysis. Subsequently, differential gene expression analysis was performed for cell clusters with at least two pseudobulks per condition (CON and DS) using DESeq2 v.1.42.1[79], comparing DS versus CON pseudobulks for each cluster (Wald test, design ~cluster_group). Genes with $P_{adj}$(FDR) < 0.10 and |log$_2$(FoldChange)| > log$_2$(1.2) were considered as differentially expressed.

Overrepresentation analyses for GO terms for biological processes were performed on the union of all DEGs, using the R clusterProfiler package v.4.10.1 (ref. 80) with annotations from the DOSE (v.3.28.2) and org.Hs.eg.db (v.3.18.0) packages. Enriched genes per term or gene set were overlapped with the DEGs per cluster to identify which genes related to the respective gene set were deregulated in which cluster. For heatmap representations, the package pheatmap (v.1.0.12) was used. Gene z-scores were calculated over all analyzed pseudobulks, and for each cluster the mean of the z-scores of all CON and all DS samples was calculated, as well as the difference in mean z-scores (DS − CON)

as a measure to visualize the magnitude of the expression difference between both groups.

As alternative single-cell based differential gene expression analysis approach, the Nebula package, was used[20] (v.1.5.3). The Seurat object was subsetted for each cluster to retain only cluster cells, converted to a Nebula object, using scToNeb(assay = "RNA", id = "sample", pred = c("group"), offset = "nCount_RNA"), a model matrix generated using model.matrix(~group, data=seuratdata$pred), and the differential expression results calculated using nebula(seuratdata$count, seuratdata$id, pred=df, offset=seuratdata$offset, ncore = 16).

## Chromatin accessibility mapping and gene-regulatory network analysis

For accurate identification of accessible regions, peaks of ATAC reads were called for each cluster using the Signac function CallPeaks(group. by = "cluster_name"), using annotation packages BSgenome.Hsapiens.UCSC.hg38 (v.1.4.5) and EnsDb.Hsapiens.v86 (v.2.99.0), and refined by removing nonstandard chromosome annotations (keepStandardChromosomes(pruning.mode = "coarse")) and "blacklisted" regions that are generally excluded from ATAC-seq analyses (subsetByOverlaps(ranges = blacklist_hg38_unified, invert = TRUE)).

Peaks were then classified with Ensembl annotations using the EnsDb.Hsapiens.v86 package into peaks overlapping with exons, introns, promoter regions and other peaks (intergenic), using the functions intronicParts(), exonicParts() and promoters() to retrieve Ensembl annotations and the findOverlaps() function to identify overlapping ATAC peaks. To link peaks as putative active cis-regulatory elements to likely target genes, peaks were mapped to the closest gene promoter, using the distanceToNearest() function.

Gene-regulatory network analysis was performed using the R package scMEGA v.1.0.2[27], based on the scMEGA GitHub analysis workflow for 10X Multiome data. Cells of the main excitatory lineage populations (excluding AST and NEU_low populations) were ordered along the excitatory lineage trajectory using the manually ordered subset clusters with AddTrajectory(trajectory = cluster_names, group.by = "cluster_name", reduction = "umap", dims = 1:2, use.all = TRUE).

All steps of scMEGA were based on the SCT-normalized RNA data and the ATAC peak data mapped by cluster, and the ChromVar TF activity data calculated from these (parameters tf.assay = "chromvar", rna. assay = "SCT", atac.assay = "peaks_by_cluster"). TF motifs from the JASPAR database (JASPAR2024 package, v.0.99.6) were retrieved using getMatrixSet(x = JASPAR2024@db, opts = list(collection = "CORE", tax_group = 'vertebrates', all_versions = FALSE)) and mapped to the ATAC peaks, using AddMotifs(genome = BSgenome.Hsapiens.UCSC. hg38). TF activity was calculated using RunChromVAR(). Selection of TFs was not restricted for the initial network analysis (except excluding TFs with activity or expression of 0 over the whole trajectory), to prevent exclusion of TFs with repressive activity and without prominent alterations along the differentiation trajectory. Peak–gene links were identified with SelectGenes(), which also generated the matched chromatin accessibility–gene expression heatmap shown in Fig. 3c. The TF activity–gene expression correlation was calculated with GetTFGeneCorrelation(), limited to genes differentially expressed in DS. The predicted TF–target interactions were extracted with GetGRN(), and interactions between differentially expressed TFs and target genes with FDR ≤ 0.05 and |correlation| ≥ 0.3 were used for the construction of the 'unfiltered' network based on correlation of chromatin accessibility and/or TF activity with gene expression along the scMEGA-defined excitatory lineage trajectory.

We then filtered for interactions that are consistent with the hypothesis that the change of expression of the TF determines the change of target gene expression in DS versus CON. For this, we calculated a scaled relative expression in DS versus CON for each gene over all cells (difference mean SCT-normalized expression $z$-score DS cells versus CON cells; that is, all genes with a mean $z$-score >0 are increased in DS), retaining only interactions with $z$(TF) × $z$(target) × correlation > 0 (that is, for a positive regulation or correlation along the trajectory, both TF and target are either upregulated ($z$(TF) > 0, $z$(target) > 0) or downregulated ($z$(TF) < 0, $z$(target) < 0) in DS, for negative regulation, either the TF is upregulated and the target downregulated ($z$(TF) > 0, $z$(target) < 0), or vice versa).

The number of TF–target interactions per factor was quantified and the genes in the network were classified as TFs, Chr. 21 genes or Chr. 21 TFs (TF_Chr. 21). As a (coarse) measure for the average relative gene expression of each gene, the mean of the vst-normalized expression (DEseq2 vst() function) of all CON pseudobulks was calculated. Network plots were generated using the ggraph package (v.2.2.1), representing genes as nodes with node sizes corresponding to the mean CON gene expression (vst-normalized) and the color indicating the relative expression in DS versus CON (difference of mean expression $z$-score of DS samples and CON samples).

To validate scMEGA predictions of TF binding to putative cis-regulatory elements with ChIP-seq data publicly available in the ChIP-Atlas database, we downloaded merged bed files containing ChIP-seq peaks from all human cell datasets (files for individual TFs from https://chip-atlas.dbcls.jp/data/hg38/assembled/Oth.ALL.05. [TF].AllCell.bed, with [TF] representing the individual TF symbols). Overlaps of ChIP-Atlas ChIP peaks with the ATAC peaks of the DS dataset were identified using findOverlaps(). ATAC peaks with ChIP-validated TF binding were overlapped with peaks predicted by scMEGA to bind the corresponding TF, to identify high-confidence TF binding regulatory elements, and their target genes predicted by scMEGA. Enrichment of TF binding in predicted target genes was statistically assessed by quantifying the number of predicted targets with or without high-confidence TF binding regulatory elements versus the background of predicted nontargets, followed by a two-sided Fisher's exact test for each TF and Benjamini–Hochberg correction for multiple testing.

To identify putative PPIs, including the network TFs from the gene-regulatory network analysis with Chr. 21 genes, we extracted experimentally validated PPIs from the BioGRID database (v.4.4.233), using the BioGRID API via the R packages jsonlite (v.1.8.8) and httr (v.1.4.7). We extracted all interactions including network TFs or differentially expressed Chr. 21 genes using https://webservice.thebiogrid. org/interactions/, then filtering for interactions of Chr. 21 genes with network TFs directly or via one common interacting protein.

Inspired by the scMEGA approach, for this space of potential Chr. 21 gene–TF interactions, we then calculated the correlation of TF activity along the excitatory lineage trajectory with the expression of the linked Chr. 21 gene. For this we assigned cells to 100 trajectory bins as determined by scMEGA and calculated the mean expression or TF activity per bin. The correlation for each Chr. 21–TF pair over all bins was determined with the R cor.test() function, and $P$ values were adjusted by Benjamini–Hochberg multiple testing correction to identify statistically significant correlations. To identify interactions of Chr. 21 genes with TFs that might determine the changes in TF activity in DS, we filtered the interactions as described for the scMEGA analysis. We calculated mean $z$-scores of Chr. 21 expression and TF activity of all DS and all CON cells for all Chr. 21 genes and TFs in the analysis, respectively, and retained only interactions with $P_{adj} \le 0.05$, abs(correlation) ≥ 0.2, abs($z$(TF activity)) ≥ 0.01, abs($z$(Chr. 21 gene expression)) ≥ 0.1, and consistent directions of the predicted regulation and changes in DS ($z$(TF activity) × $z$(Chr. 21 gene expression) × correlation > 0, as described for the scMEGA analysis).

Network plots were generated using the ggraph package (v.2.2.1), representing genes as nodes with node sizes corresponding to the CON gene expression (vst-normalized, as described for the scMEGA analysis), border color indicating the relative expression in DS versus CON (difference of mean expression $z$-scores of DS samples and CON

samples), and fill color the relative TF activity in DS versus CON (difference of mean activity z-scores of DS samples and CON samples).

## Bulk RNA-seq and analyses

For bulk RNA-seq of fetal tissue samples (Supplementary Table 3), total RNA was extracted using the RNeasy Plus Micro kit (Qiagen) according to manufacturer's instructions. RNA-seq was then performed by the NIHR Imperial BRC Genomics Facility using the NEBNext rRNA Depletion kit v.2 (Human/Mouse/Rat) and NEBNext Ultra II Directional RNA Library Prep Kit from Illumina. Libraries were sequenced in PE75 mode.

For in vitro differentiated neuronal cultures, cells were allowed to mature for 30 days before being dissociated with Accutase, pelleted and flash frozen. Samples were stored in −80 °C until further processing. Library preparation and bulk RNA-seq were performed using AccuraCode RNAseq Kit (Singleron Biotechnologies).

For ASO experiments, total RNA was extracted from NPCs using FastPure Cell/Tissue Total RNA Isolation Kit (Vazyme) according to manufacturer's protocol. RNA samples were stored in −80 °C before processing by DNBSEQ eukaryotic strand-specific transcriptome resequencing (BGI Genomics).

Gene-level count matrices were generated by mapping to the human genome (GRCh38) using the pipelines nf-core/rnaseq (v.3.18.0; https://doi.org/10.5281/zenodo.1400710) or AccuraCode (v.1.2.0; for Singleron data).

Differential gene expression analyses for the tissue bulk analysis was performed using DESeq2 comparing DS and CON using Wald's test.

For analyses of in vitro experiments, groups were compared using DESeq2 with a LRT, using a multifactorial design comparing group effects (DS versus CON or DS ASO-treated versus untreated) between one or more paired technical replicates across experimental batches (biological replicates). The analysis was performed using the commands DESeqDataSetFromMatrix(counts_comp, colData = meta_comp, design = ~ batch + group) and DESeq(test = "LRT", reduced = ~ batch), applying a significance cutoff of $P_{adj} < 0.1$. For the ASO experiments, samples treated with different ASO designs targeting the same Chr. 21 TF were considered as technical replicates.

For visualizing expression z-scores, the count matrices were normalized using vst(), followed by batch correction with removeBatchEffect(batch = meta$batch") from the limma package[81] (v.3.58.1). For each group comparison, the difference of the mean of the z-scores by group were visualized as measure of the magnitude of the expression difference between both groups.

## Antisense oligonucleotide in vitro treatment

NPCs derived from DS (C13) and isogenic control (C9) iPSC lines were seeded on Geltrex (Gibco)-coated 24-well culture plates, with or without coverslips, at a density of 70,000 cells per well in NEM.

To assess and optimize transfection efficiency, C13 NPCs were seeded on poly-L-ornithine-coated coverslips in NEM. The following day, the medium was replaced with BrainPhys Neuronal Medium, and cells were transfected with 100 nM Alexa Fluor 488-labeled HPRT control ASO (Integrated DNA Technologies) using Lipofectamine 3000 (Thermo Fisher Scientific) according to the manufacturer's protocol. Transfection was carried out at 37 °C for 96 h, after which cells were harvested for downstream applications. Following treatment, cells were fixed in 4% PFA in 1× PBS for 15 min at room temperature, washed three times with 1× PBS for 10 min each, and permeabilized with 0.1% Triton X-100 in 1× PBS. Cells were then counterstained with DAPI (1:1,000, Thermo Fisher Scientific), followed by 1× PBS washes. Coverslips were mounted using ProLong Glass Antifade Mountant (Thermo Fisher Scientific), stored in the dark at 4 °C overnight, and then imaged using a LSM980 confocal fluorescence microscope (Zeiss).

To assess knockdown efficacy of ASOs by RT–qPCR and RNA-seq experiments (Extended Data Fig. 9e and Supplementary Table 5),

NPCs derived from DS (C13) and isogenic control (C9) iPSC lines were seeded on Matrigel-coated plates at a density of 100,000 cells per cm² in NIM without penicillin or streptomycin. We transfected cells with 2′-O-methoxyethyl gapmer ASOs targeting Chr. 21 TFs (Integrated DNA Technologies) at concentrations of 100 nM or 1,000 nM using Lipofectamine Stem (Thermo Fisher Scientific), following the manufacturer's protocol. Briefly, Lipofectamine Stem was mixed with Opti-MEM and incubated for 5 min, followed by a 15-min incubation after addition of ASOs. The transfection mixture was added drop-wise to freshly seeded cells, and the medium was replaced the following day. Cells were harvested 4 days post-transfection for downstream analyses. ASO sequences are listed in Supplementary Table 5.

## Quantitative reverse transcription PCR

Total RNA was extracted from NPCs using the FastPure Cell/Tissue Total RNA Isolation Kit, following the manufacturer's instructions. RNA concentration and purity were assessed using a NanoDrop N2000 spectrophotometer (Thermo Fisher Scientific). For each sample, 500 ng of total RNA was reverse transcribed into complementary DNA using SuperScript IV VILO Master Mix (Thermo Fisher Scientific).

Quantitative PCR was performed using 10-μl reactions containing 5 ng of cDNA template, 0.5 μl of 10 μM forward and reverse primers (Integrated DNA Technologies), and 5.5 μl of SupRealQ Ultra Hunter SYBR qPCR Master Mix (Vazyme). Reactions were run on a QuantStudio 5 Real-Time PCR System (Thermo Fisher Scientific) using the following cycling conditions: 95 °C for 30 s, followed by 40 cycles of 95 °C for 1 s and 60 °C for 20 s. Melt curve analysis was performed to confirm amplification specificity. Relative gene expression was calculated using the $\Delta\Delta C_t$ method, normalized to GAPDH as the housekeeping gene. All reactions were performed in technical triplicates. Primer sequences are provided in Supplementary Table 5.

## Western blot

NPCs were lysed in RIPA buffer (Thermo Fisher Scientific) supplemented with protease and phosphatase inhibitors on ice for 30 min. Lysates were centrifuged at 15,000g for 15 min at 4 °C, and the supernatant was collected for protein quantification using the BCA Protein Quantification Kit (Vazyme) following manufacturer's protocol.

Equal amounts of protein (50 μg) were mixed with 1× NuPAGE LDS sample buffer (Thermo Fisher Scientific) and 1× NuPAGE Sample Reducing Agent (Thermo Fisher Scientific), then denatured at 70 °C for 10 min. Samples were resolved by sodium dodecyl sulfate–polyacrylamide gel electrophoresis on 4%–20% Mini-PROTEAN TGX Precast Protein Gels (Bio-Rad) in 1× Tris glycine–sodium dodecyl sulfate running buffer at 100 V for 1 h. A 250 kDa Plus Prestained Protein Marker (Vazyme) was used as the molecular weight reference. Proteins were transferred to nitrocellulose membranes (Bio-Rad) using a wet transfer system at 120 V for 1.5 h on ice.

Membranes were blocked in 5% nonfat milk in TBST for 1 h at room temperature on shaker, followed by overnight incubation at 4 °C with primary antibodies diluted in blocking buffer. After three washes with TBST, each of 5 min, at room temperature, membranes were incubated with appropriate horseradish peroxidase-conjugated secondary antibodies for 1 h at room temperature on a shaker. After three additional washes, the signal was visualized using a SuperPico ECL Chemiluminescence Kit (Vazyme) and imaged on the ChemiDoc MP Imaging System (Bio-Rad). Band intensities were quantified using ImageJ and normalized to actin beta (ACTB) as the loading control. Details of antibodies used are included in the reporting summary.

## snRNA-seq and analysis of human graft tissue

Frozen human grafts were processed and sequenced as described above for fetal tissue (10X Multiome technology), or were processed by Singleron and sequenced using CeleScope scope v.3.0.1 (kit V2) technology (Supplementary Table 6). For samples processed by Singleron,

nuclei were isolated from frozen human graft tissue, and single nuclei RNA-seq libraries were constructed using GEXSCOPE Single Nuclei RNAseq Library Kit (Singleron Biotechnologies) according to the manufacturer's instructions. Briefly, for each library the nuclei suspension of specified concentration was loaded onto a microfluidic chip for capture. The single-nuclei partitioning, lysis and mRNA capture steps were automated using Singleron Matrix NEOTM system. The final single-nuclei RNA sequencing libraries were sequenced on an Illumina NovaSeq6000 flow cell with paired-end 150 bp. Count matrices were generated as described for fetal tissue (cellranger-arc; v.2.0.2; 10X Genomics), or for Singleron-sequenced samples using the CeleScope tools (v.1.14.0; www.github.com/singleron-RD/CeleScope, assay RNA, Singleron Biotechnologies), to generate gene expression matrix files using default parameters. Briefly, cellular barcodes in Read 1 were used to demultiplex and identify reads of the same cell origin. The mapping was done using STARSOLO (https://github.com/alexdobin/STAR/blob/master/docs/STARsolo.md) against the human genome build GRCh38 with ENSEMBL Gene Annotation (v.99). The reads were assigned to genes using the featureCount tool and the cell calling was performed by fitting a negative bimodal distribution and determining the threshold between empty wells and cell-associated wells. The gene count matrix was then generated, providing the number of unique molecular identifiers (UMIs) for each gene and cell.

Quality control was performed as for fetal tissue with following modifications: because ATAC data were not analyzed here, only RNA-based filtering was performed to remove low-quality cells. Cells with UMI counts <500 or >30,000, or with mitochondrial gene content >2%, were excluded from the analysis. Datasets with <500 cells were removed, and large datasets were randomly subsampled to 2 × median number of cells of the remaining samples, to reduce bias toward overrepresented samples. RNA data were then normalized by SCTransform and mapped onto the complete fetal dataset as a reference (from Fig. 1; excluding noncortical samples) using FindTransferAnchors(reference = seur_ref, query = seur, dims = 1:30, reference.reduction = "pca") and MapQuery(anchorset = anchors, reference = seur_ref, query = seur, refdata = list(cluster_name = "cluster_name", …), reference.reduction = "pca", reduction.model = "umap"), to transfer metadata related to UMAP coordinates, cluster names, cell types and developmental stage to the graft dataset. Cells mapping to the fetal excitatory lineage were re-mapped as above to the fetal excitatory lineage clusters (from Fig. 2a).

Differential cell abundance and gene expression analyses were performed with sccomp and DESeq2 using the transferred cluster labels, as described for fetal tissue. To correct for the use of a different sequencing technology (Singleron CeleScope) for some of the graft samples, DESeq2 analysis was performed with a LRT including the sequencing technology as covariate, separately for each cluster, using the commands DESeqDataSetFromMatrix(counts_cluster, colData = meta_cluster, design = ~seq_tech + group) and DESeq(test = "LRT", reduced = ~seq_tech), with a cutoff of $P_{adj}$ < 0.1.

For visualizing expression z-scores including fetal and in vitro data, the combined bulk and pseudobulk count matrices were normalized using vst(), followed by batch correction with removeBatchEffect(batch = meta[["seq_tech"]], group = meta$group) from the limma package[81] (v.3.58.1).

## Statistics and reproducibility
Data shown for representative experiments were repeated, with similar results, in at least two independent biological replicates and at least three technical replicates, unless otherwise noted. No statistical method was used to predetermine sample size. Low-quality tissue samples were excluded as outlined in the relevant Methods sections and Supplementary Tables 1 and 6. To avoid biases, analyses were performed using automated computational approaches. Therefore, no blinding was performed.

## Reporting summary
Further information on research design is available in the Nature Portfolio Reporting Summary linked to this article.

## Data availability
Raw and processed sequencing data generated for this study will be made available under GEO accessions GSE305153. The fetal tissue dataset will also be available as an interactive online resource on the CELLxGENE platform (https://cellxgene.cziscience.com/collections/0e9fd1d3-ef4c-47c6-a2e4-ef4bfadf7c79). Other data (for example, immunostainings) will be provided by the authors upon reasonable request. Source data are provided with this paper.

## Code availability
The pipelines used for the analyses in this paper will be made available on GitHub (https://github.com/lattkem1/Down_Syndrome_Multiome).

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

## Acknowledgements
We thank the MRC-Wellcome Trust Human Developmental Biology Resource (HDBR), S. Lisgo and N. Solanski, the donors and their families for the support, without which this study could not have been possible. A. Murray and E. Brockman for help with cell culture. P. J. Kang and members of S.-C. Zhang's laboratory for help setting up the human iPSC-derived neuron culture protocols. S. H. Li and M. Varela for help with the mouse colony maintenance. S. Pasculli for sharing analysis protocols for the qPCR experiments. A. Tan for help with the RNA extraction and qPCR/WB. X. Roca and H. Hua for assistance with the design and interpretation of the ASO experiments. J. J. Gooley and X. Roca, for comments on the manuscript. This work was supported by the U.K. Medical Research Council MR/V034529/1 and Singapore Ministry of Health (MOH) (V.D.P.), the Wellcome Trust Collaborative Award in Science 217199/Z/19/Z (D.N.) and the Croatian Science Foundation grants HRZZ-UIP-2025-02-5828 (I.A.) and HRZZ-IP-2022-10-5975 (Z.K.). The funders had no role in study design, data collection and analysis, decision to publish or preparation of the manuscript.

## Author contributions

M.L. conceptualized and performed experiments with fetal tissue (stainings, sequencing), conceptualized and performed all sequencing data analyses, interpreted results and wrote the manuscript. K.R. performed library preparations and sequencing, N.M. contributed to sequencing experiment design, Ž.K. sourced and processed fetal tissue, performed stainings and imaging. M.S., S.S., W.L.T. and S.K.S. generated the neurons for the in vitro and in vivo experiments. K.H.U. and W.L.T. performed the qPCR, WB and ASO experiments, and V.D.P., J.T., A.L. and V.A.B. the transplantation and graft extraction experiments. I.A. and D.N. provided some of the human iPSC models and derived neural progenitors used for cell culture experiments. B.P.L. contributed to study conceptualization and results interpretation. V.D.P. conceived and directed the study, acquired funding, interpreted results and wrote the manuscript.

## Competing interests

A patent application has been filed related to antisense oligonucleotide targets identified in this study (V.D.P. and M.L.). The other authors declare no competing interests.

## Additional information

**Extended data** is available for this paper at https://doi.org/10.1038/s41591-026-04211-1.

**Correspondence and requests for materials** should be addressed to Michael Lattke or Vincenzo De Paola.

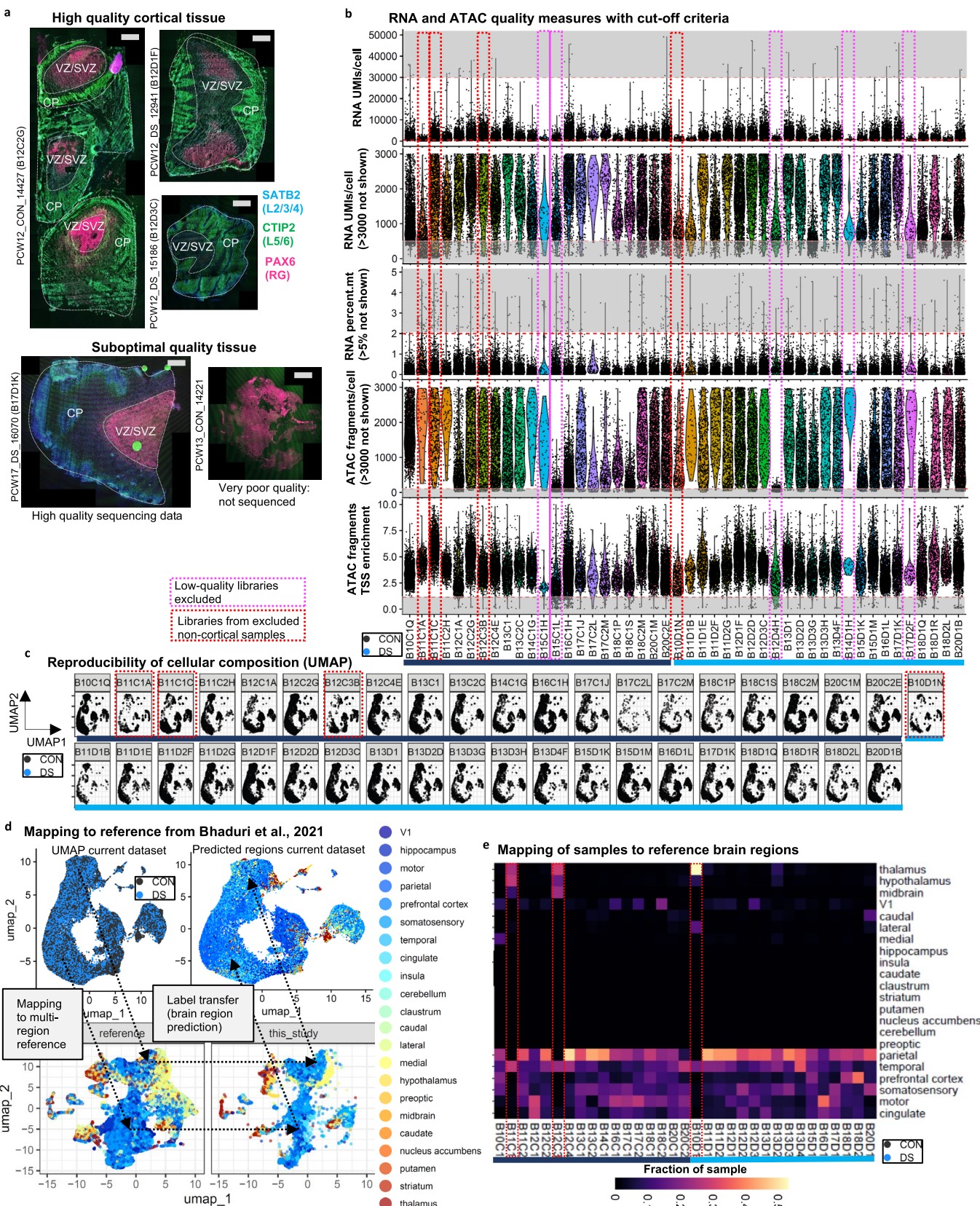

**Extended Data Fig. 1 | Quality control and identification of non-cortical samples in the fetal brain single-cell multiomic atlas. a**, Immunostaining for markers of the cortical progenitor zone (PAX6) and cortical plate neurons (SATB2, CTIP2) to identify samples suitable for single-cell multiome analysis. Representative confocal images of all 20 CON and 19 DS samples, scale bar: 1 mm. **b**, RNA- and ATAC-seq quality measures per cell by library; grey shadows delimited by the red dotted cut off lines: excluded low-quality cells; purple dotted boxes: libraries excluded due to high fraction of low-quality cells; red dotted boxes: libraries from excluded non cortical samples. **c**, UMAP plot split by libraries included in the dataset. Note reproducible contribution of libraries/ samples to cell populations. **d-e**, Regional identity predicted by mapping of cells / samples to multi-region reference dataset[15]; highlighted by dotted green boxes (e): Excluded samples with predicted non-cortical cells.

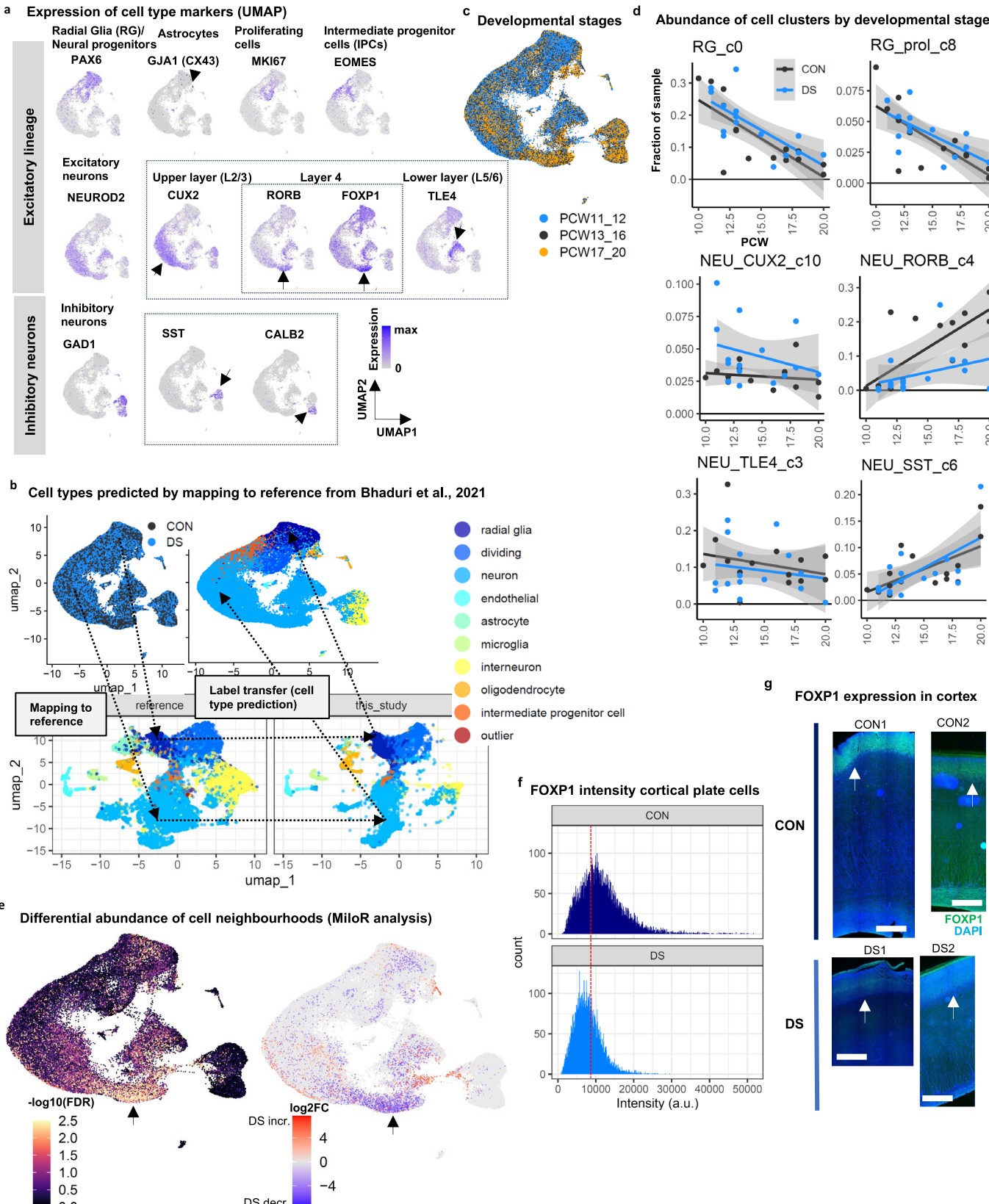

**Extended Data Fig. 2 | See next page for caption.**

**Extended Data Fig. 2 | Characterization of cell populations in the fetal brain single-cell multiomic atlas. a**, Cell type marker gene expression projected on UMAP dimensionality reduction plot. Arrows: small populations with high marker expression; rectangles: subtype markers. **b**, Cell-type predictions from mapping to reference dataset[15]. **c**, Contribution of samples from different stages to cell populations (UMAP plot). **d**, Changes of cell proportions in samples over developmental time. Shown individual samples and linear regression prediction with 95% confidence interval. **e**, Differential abundance of cells in local neighborhoods quantified with MiloR[17]. Negative log10-transformed adjusted p-value (left) and log2-transformed fold change by neighborhood (right); Arrows: population of RORB/FOXP1-expressing neurons. **f-g**, FOXP1 immunostaining in CON and DS brains from PCW16-20; Intensity distribution of FOXP1 staining of cryosections (n = 7 CON and 6 DS brains) (f) with cut-off for quantification (red line); Sections from two paraffin-embedded CON and two DS brains. Representative images of each at least 2 sections (g, arrow: cortical plate); scale bars: 1 mm.

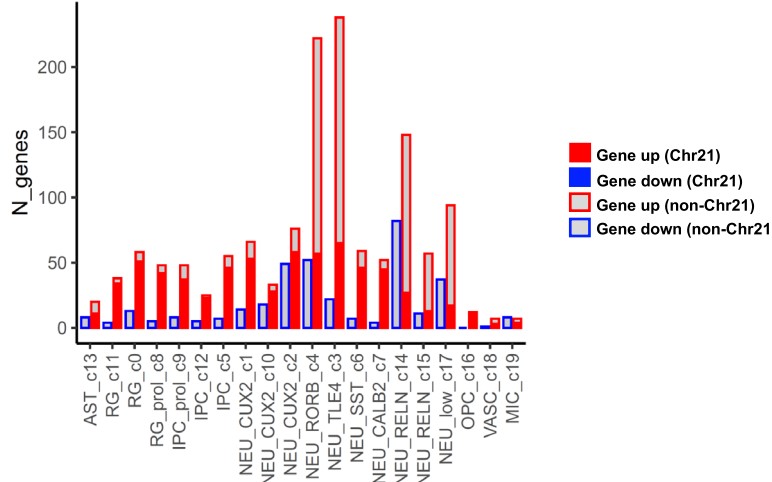

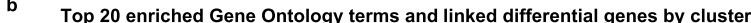

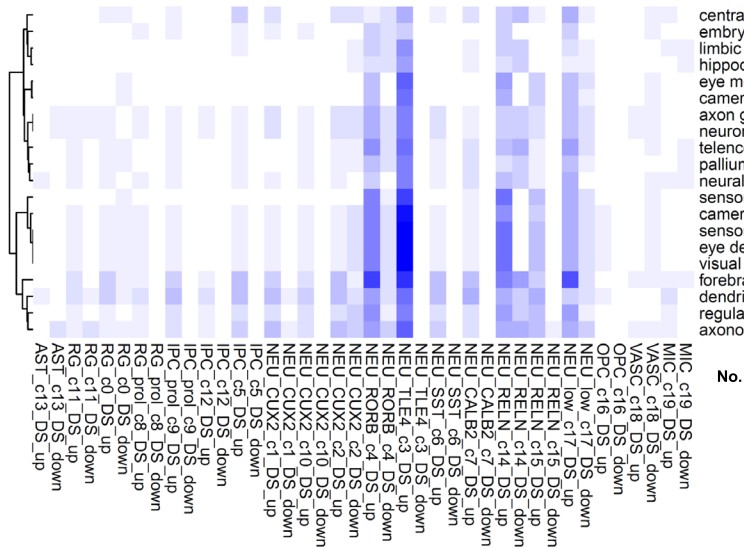

**Extended Data Fig. 3 | Expression changes in the complete dataset comprising all cell populations and gene ontology (GO) terms enriched for differential genes. a**, Number of genes differentially expressed between DS and CON samples by cluster; DESeq2 pseudobulk analysis with Wald test, threshold padj<0.10, |log2FoldChange| > log2(1.2). **b**, Biological processes linked to differentially expressed genes: heatmap showing number of differentially expressed genes in Top20 enriched gene ontology (GO) terms by cell cluster.

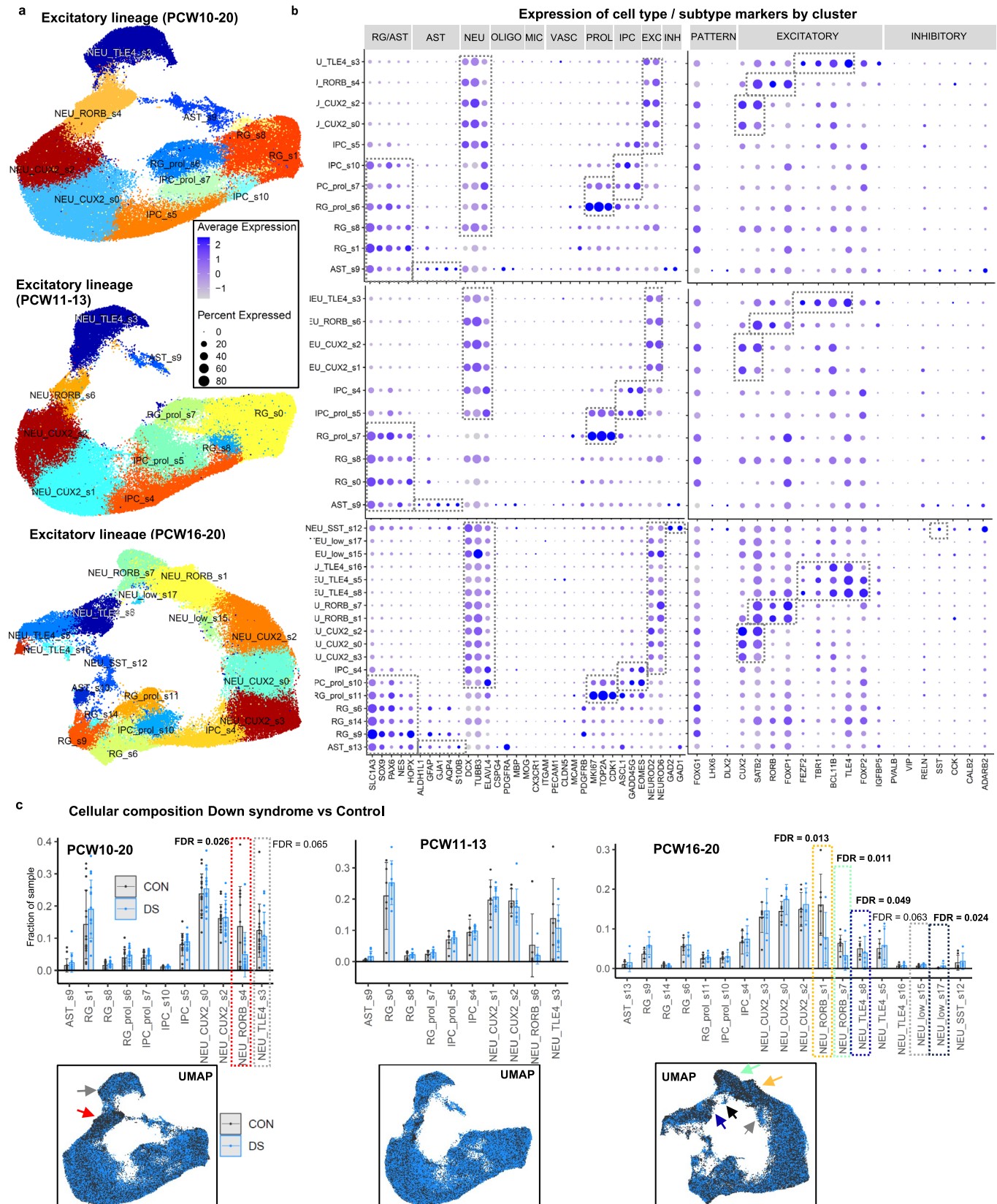

**Extended Data Fig. 4 | Subsetting of excitatory lineage cells from complete dataset (PCW10-20) or samples from early (PCW11 to PCW13) or late stages (PCW16 to PCW20) separately. a**, Dimensionality reduction, re-clustering and cell type assignment of identified excitatory lineage cell clusters (UMAP plot). **b**, Expression of marker genes used to assign clusters to cell types/subtypes. **c**, Abundance of cell populations in control (CON) and Down syndrome (DS) samples. Barplot showing individual samples with False Discovery Rate (FDR) for DS vs CON from sccomp compositional analysis[18] (other clusters FDR > 0.10) and combined UMAP plot (dotted boxes / arrows: clusters altered in DS, sccomp FDR < 0.10). n = 15/15 for CON/DS in left panel; 6/10 in middle panel; 7/5 in right panel. Error bars: mean +/- SD.

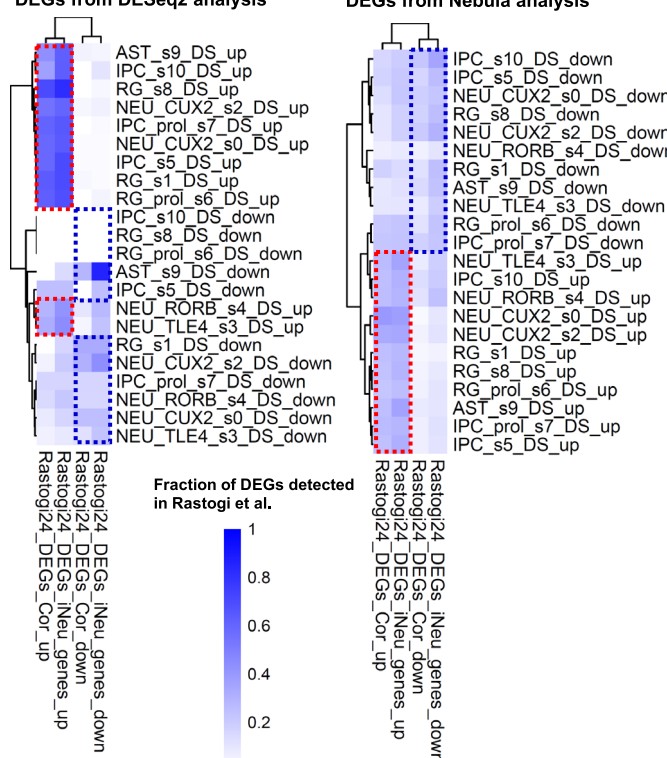

**a** Differentially expressed genes excitatory lineage (Nebula)

**b** Fraction of DESeq2 DEGs also detected by Nebula analysis

**c** Fraction of DEGs also identified in adult cortex and iPSC-derived neurons (from Rastogi et al., 2024)

DEGs from DESeq2 analysis　　DEGs from Nebula analysis

**Extended Data Fig. 5 | Validation of differentially expressed genes with alternative analysis approach and published data. a**, Number of differentially expressed genes between DS and CON samples by cluster detected by Nebula analysis[21]; Threshold padj<0.05, |log2FoldChange| > log2(1.2). **b**, Fraction of differential genes per cluster identified with DESeq2 also detected by Nebula.

**c**, Fraction of differential genes (identified with DESEq2 or Nebula) also detected in adult cortical tissue or iPSC-derived neurons (bulk-RNA-seq from[12]); note that Nebula identified more differential genes (a, see also Fig. 2b), but showed lower concordance with the published data (c). Boxes (b, c): Genes consistently upregulated (red) / downregulated (blue) in both analyses.

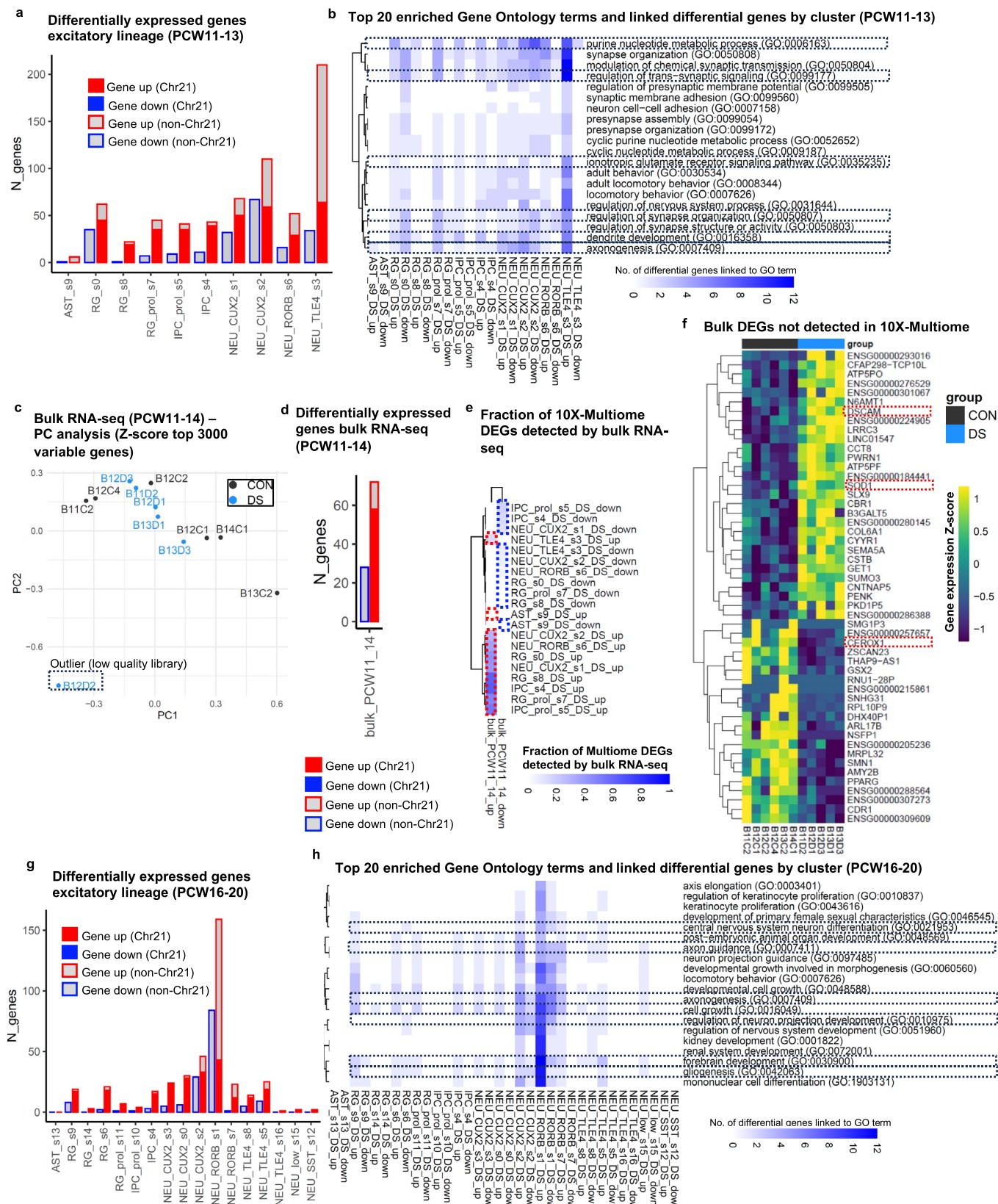

**a** Differentially expressed genes excitatory lineage (PCW11-13)

**b** Top 20 enriched Gene Ontology terms and linked differential genes by cluster (PCW11-13)

**c** Bulk RNA-seq (PCW11-14) – PC analysis (Z-score top 3000 variable genes)

**d** Differentially expressed genes bulk RNA-seq (PCW11-14)

**e** Fraction of 10X-Multiome DEGs detected by bulk RNA-seq

**f** Bulk DEGs not detected in 10X-Multiome

**g** Differentially expressed genes excitatory lineage (PCW16-20)

**h** Top 20 enriched Gene Ontology terms and linked differential genes by cluster (PCW16-20)

**Extended Data Fig. 6 | See next page for caption.**

**Extended Data Fig. 6 | Expression changes in early and late excitatory lineage (PCW11-13 / PCW16-20). a**, Number of genes differentially expressed between DS and CON samples PCW11-13 by cluster; DESeq2 pseudobulk analysis with Wald test, threshold padj<0.10, |log2FoldChange | > log2(1.2). **b**, Biological processes linked to differentially expressed genes PCW11-13: heatmap showing number of differentially expressed genes in Top20 enriched gene ontology (GO) terms by cell cluster. **c-f**, Complementary bulk-RNA-seq analysis from samples PCW11-14; Principal component analysis showing outlier low-quality library omitted from following analyses (c); Number of differential genes (d, DESeq2 pseudobulk analysis with Wald test, threshold padj<0.10, |log2FoldChange | > log2(1.2)); Fraction of differential genes in PCW11-13 populations also detected by bulk RNA-seq (e). Heatmap of expression of differential genes detected by bulk-RNA-seq, but not in single-cell analyses (f); Red dotted boxes: genes discussed in main text. **g, h**, Number of differential genes and enriched GO terms by cluster for PCW16-20 (as in a-b).

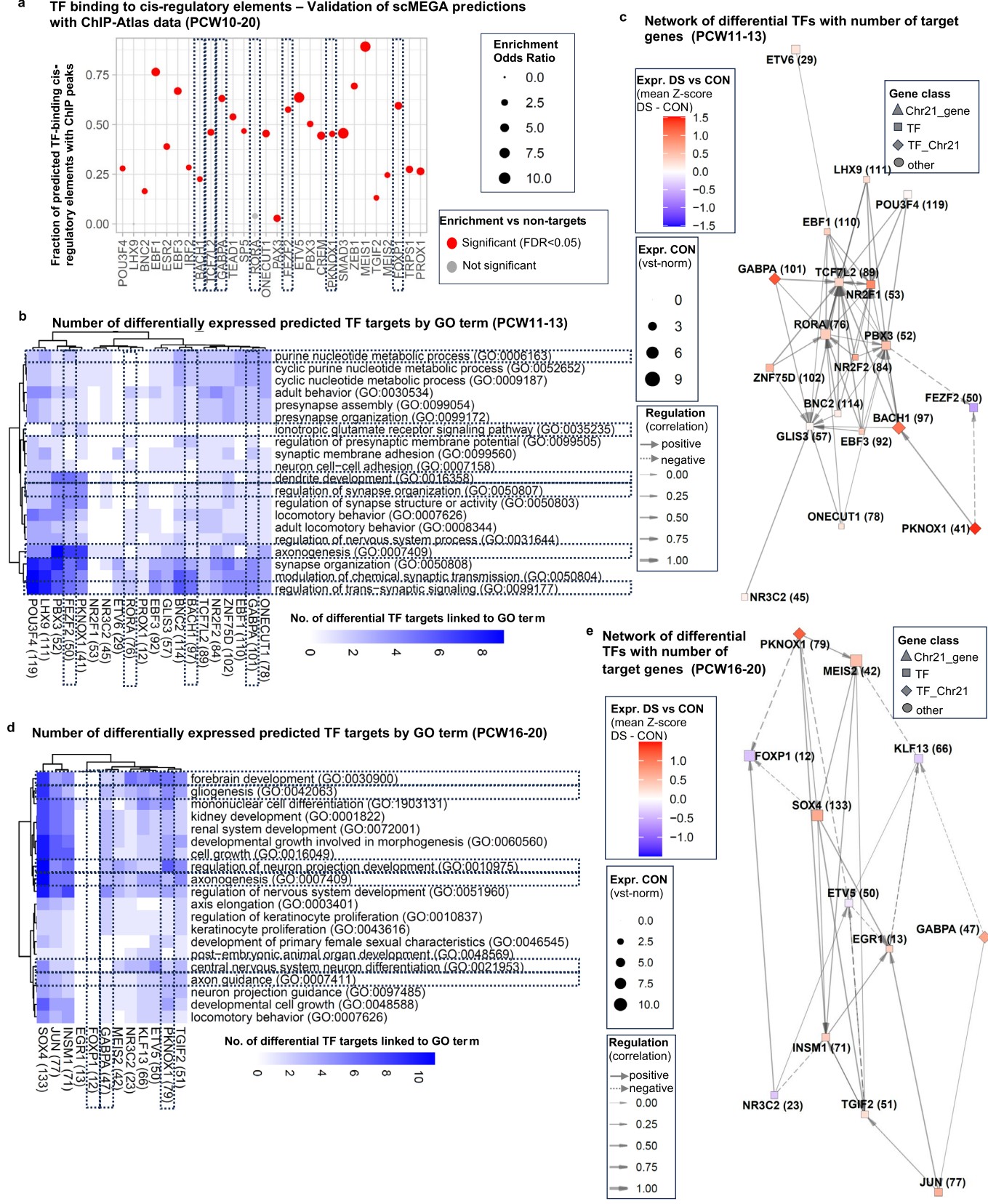

**Extended Data Fig. 7 | See next page for caption.**

**Extended Data Fig. 7 | Validation of predicted TF binding to cis-regulatory elements and prediction of early and late-stage TF networks. a**, Fraction of predicted TF-cis-regulatory-element interactions validated by published ChIP-seq data from ChIP-Atlas, with enrichment over non-targets. Note that low validation rate may be due to ChIP data being low quality or coming from non-neural cells; Boxes: selected key network TFs. **b**, TFs predicted to regulate differentially expressed genes linked to altered neural functions at PCW11-13. Heatmap showing number of interactions between TFs and differential genes linked to Top20 enriched gene ontology (GO) terms. In brackets: total number of targets per TF. Highlighted: GO terms and putative key regulators discussed in main text. **c**, Network plot showing predicted interactions of TFs regulating differentially expressed genes for PCW11-13 (in brackets: number of TF-target-interactions). Node size: relative expression in control (CON) samples (vst-normalized); node color: relative expression (z-score) in DS vs CON (each mean of all cell clusters). **d, e**, TFs predicted to regulate differentially expressed genes linked to altered neural functions, and network plot showing predicted interactions of TFs regulating differentially expressed genes at PCW16-20 (as b, c).

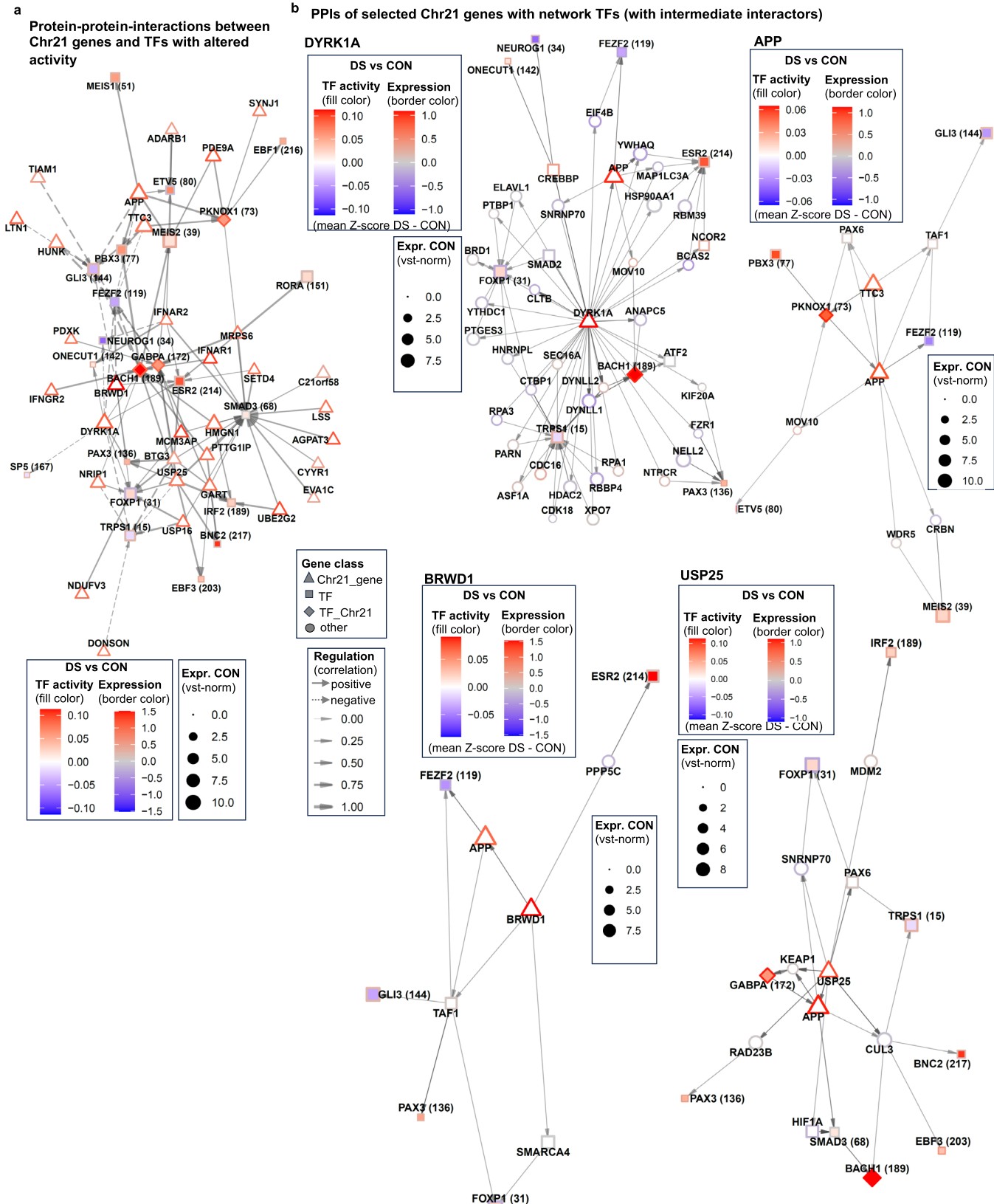

**Extended Data Fig. 8 | Prediction of Chr21 genes regulating network TFs via protein-protein-interactions from BioGRID database. a**, Interactions of Chr21 proteins with network TFs predicted to regulate TF activity via experimentally validated protein-protein-interactions (PPIs) from BioGRID database. **b**, Network plot with proteins predicted to mediate interactions of Chr21 genes DYRK1A, APP, BRWD1 and USP25 with network TFs responsible for modulating TF activity changes in DS.

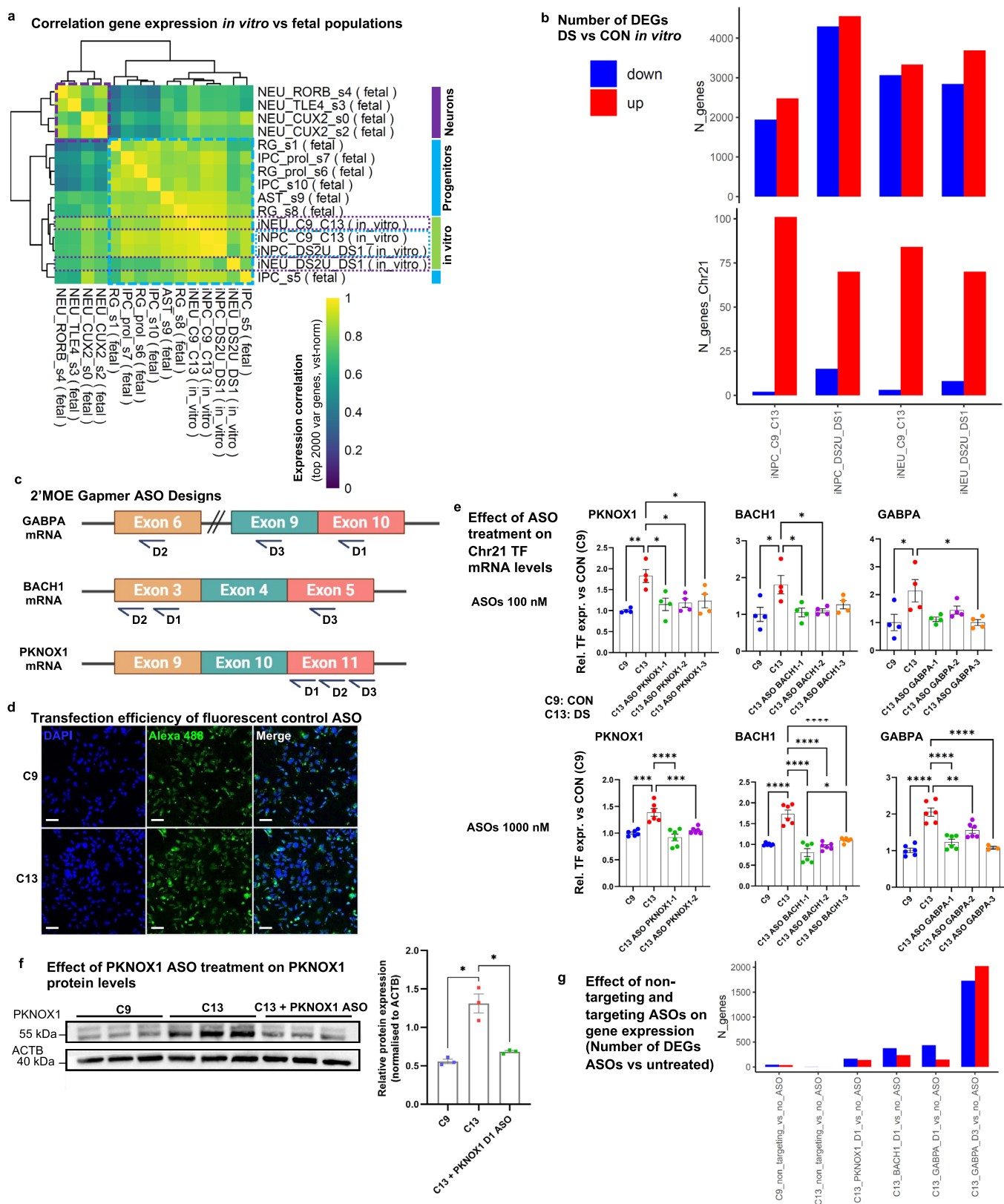

**a** Correlation gene expression *in vitro* vs fetal populations

**b** Number of DEGs DS vs CON *in vitro*

**c** 2'MOE Gapmer ASO Designs

**d** Transfection efficiency of fluorescent control ASO

**e** Effect of ASO treatment on Chr21 TF mRNA levels

ASOs 100 nM

C9: CON
C13: DS

ASOs 1000 nM

**f** Effect of PKNOX1 ASO treatment on PKNOX1 protein levels

**g** Effect of non-targeting and targeting ASOs on gene expression (Number of DEGs ASOs vs untreated)

**Extended Data Fig. 9 | See next page for caption.**

**Extended Data Fig. 9 | Characterization of iPSC-derived neural cells in vitro and validation of antisense oligonucleotide (ASO) mediated reduction of Chr21 TF expression. a**, Correlation of gene expression from in vitro-differentiated neural progenitor cell (iNPC) and neuron (iNEU) cultures and in fetal tissue; Heatmap showing the correlation of mean vst-normalised expression between merged bulk data from in vitro cultures (3-10 RNA samples from wells of paired side-by-side differentiated DS/CON cells per condition from n = 6/2/3/1 independent differentiation experiments for iNPC_C9_C13/iNPC_DS2U_DS1/ iNEU_C9_C13/iNEU_DS2U_DS1; see Methods and Supplementary Table 5) and each tissue cluster pseudobulk (excitatory lineage, PCW10–20). **b**, Number of differentially expressed genes between DS vs CON *in vitro*; Benjamini-Hochberg-adjusted across predicted TF targets p < 0.10, determined by DESeq2 analysis with Likelihood-Ratio-Test (LRT) to assess group effect across paired DS vs CON technical and biological replicates (3-10 RNA samples from wells of paired side-by-side differentiated DS/CON cells per condition from n = 6/2/3/1 independent differentiation experiments for iNPC_C9_C13/iNPC_DS2U_DS1/ iNEU_C9_C13/iNEU_DS2U_DS1; see Methods and Supplementary Table 5). **c**, Design of 2'-O-methoxyethyl (2'MOE) ASOs targeting Chr21 TFs. **d**, Validation of transfection of neural progenitors (NPCs) with non-targeting fluorescent ASOs (Alexa488-labeled Hs HPRT control ASO (100 nM)); Repeated at least three times, two-three technical replicates each, with the same result. Scale bar: 50 μm. **e**, qRT-PCR analysis of Chr.21 transcription factor (TF) mRNA levels in hiPSC-derived NPCs from DS hiPSC line (C13) and isogenic control (C9) following treatment with 100 nM (repeated at least three times, three-four technical replicates each, with the same result) or 1000 nM (repeated two times, three technical replicates each, with the same result) of TF-targeting antisense oligonucleotides (ASOs). Expression levels were normalized to GAPDH as the reference gene. Statistical analysis was performed using one-way ANOVA followed by Tukey's multiple comparison test (*p < 0.05, **p < 0.01, ***p < 0.001). Error bars represent mean ± SEM. **f**, Validation of PKNOX1 protein downregulation after 100 nM ASO treatment (repeated at least two times, three technical replicates each, with the same result); Western blot with ACTB as housekeeping gene/loading control; Protein band intensities were quantified using ImageJ. Statistical analysis was performed using two-sided Welch's t test; *p < 0.05, **p < 0.01, ***p < 0.001. Error bars represent mean ± SEM. **g**, Non-targeting ASOs exert minimal influence on gene expression compared with chromosome 21 TF-targeting ASOs. Number of differential genes determined by DESeq2 analysis with LRT to assess ASO effect across each 2-3 technical replicates from one differentiation experiment; DESeq2 analysis with LRT test, Benjamini-Hochberg-adjusted across all genes p < 0.10.

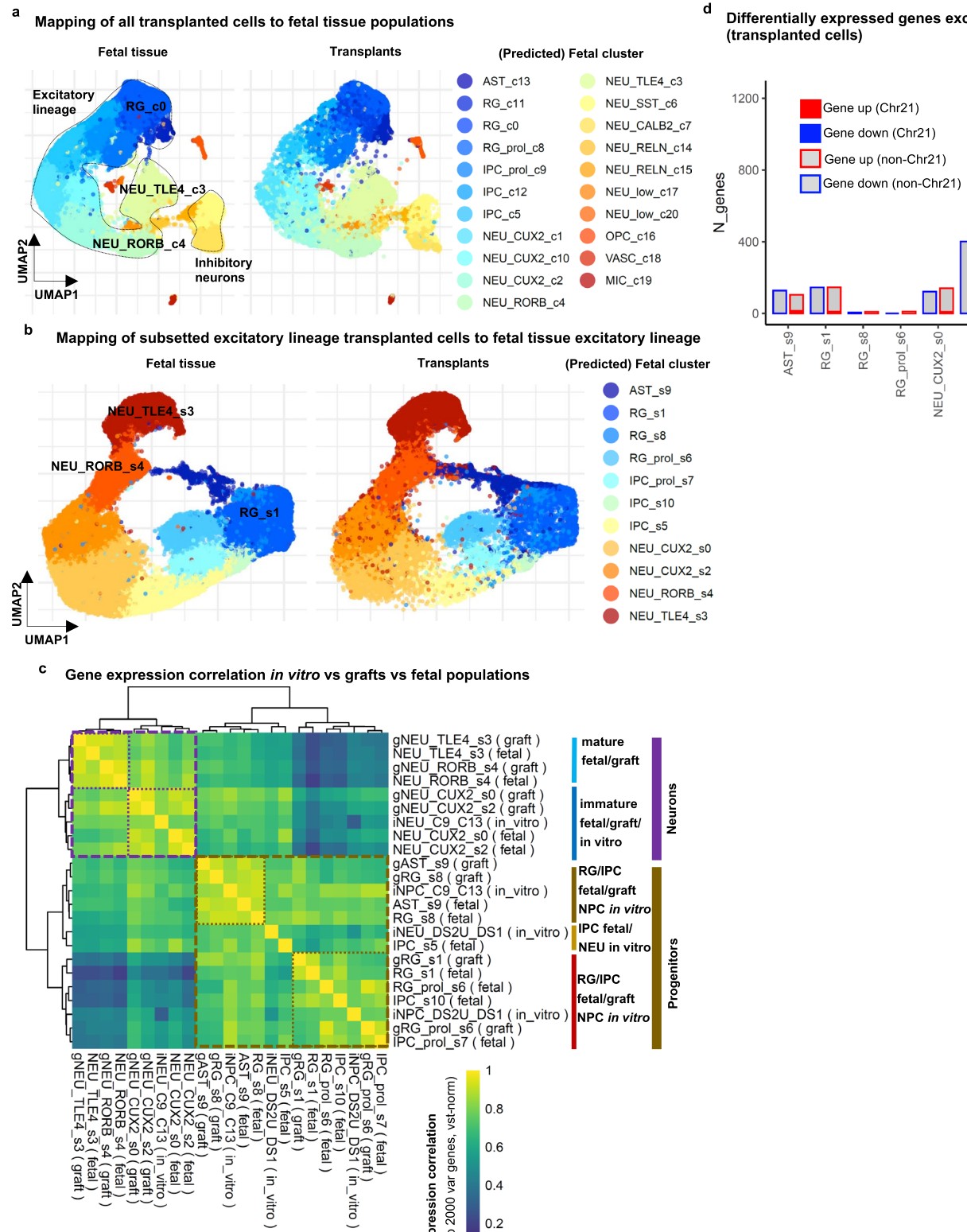

**a** Mapping of all transplanted cells to fetal tissue populations

**b** Mapping of subsetted excitatory lineage transplanted cells to fetal tissue excitatory lineage

**c** Gene expression correlation *in vitro* vs grafts vs fetal populations

**d** Differentially expressed genes excitatory lineage (transplanted cells)

**Extended Data Fig. 10 | Comparison of molecular phenotypes in iPSC-derived neural cells from transplants and *in vitro* with fetal DS cortex. a**, Mapping transcriptomes of all transplanted cells to fetal tissue populations (UMAP plot). **b**, Mapping transcriptomes of subsetted excitatory transplanted cells to PCW10-20 fetal tissue excitatory lineage populations (UMAP plot). **c**, Correlation of gene expression in neural cells *in vitro* with graft and fetal tissue populations; heatmap shows correlation of mean vst-normalised expression between each *in vitro* experiment (3-4 technical replicates) and each cluster pseudobulk from the transplants and fetal tissue analyses. **d**, Number of DEGs between DS vs CON in grafts; DESeq2 analysis for transplants with LRT test by cluster with correction for sequencing technology, threshold padj<0.10.

Vincenzo De Paola

# Reporting Summary

## Statistics

For all statistical analyses, confirm that the following items are present in the figure legend, table legend, main text, or Methods section.

| n/a | Confirmed | |
|---|---|---|
| ☐ | ☒ | The exact sample size (*n*) for each experimental group/condition, given as a discrete number and unit of measurement |
| ☐ | ☒ | A statement on whether measurements were taken from distinct samples or whether the same sample was measured repeatedly |
| ☐ | ☒ | The statistical test(s) used AND whether they are one- or two-sided *Only common tests should be described solely by name; describe more complex techniques in the Methods section.* |
| ☐ | ☒ | A description of all covariates tested |
| ☐ | ☒ | A description of any assumptions or corrections, such as tests of normality and adjustment for multiple comparisons |
| ☐ | ☒ | A full description of the statistical parameters including central tendency (e.g. means) or other basic estimates (e.g. regression coefficient) AND variation (e.g. standard deviation) or associated estimates of uncertainty (e.g. confidence intervals) |
| ☐ | ☒ | For null hypothesis testing, the test statistic (e.g. *F*, *t*, *r*) with confidence intervals, effect sizes, degrees of freedom and *P* value noted *Give P values as exact values whenever suitable.* |
| ☒ | ☐ | For Bayesian analysis, information on the choice of priors and Markov chain Monte Carlo settings |
| ☐ | ☒ | For hierarchical and complex designs, identification of the appropriate level for tests and full reporting of outcomes |
| ☒ | ☐ | Estimates of effect sizes (e.g. Cohen's *d*, Pearson's *r*), indicating how they were calculated |

*Our web collection on statistics for biologists contains articles on many of the points above.*

## Software and code

Policy information about availability of computer code

**Data collection**
Count matrices for snMultiome data were generated using cellranger-arc (v2.0.2; 10X Genomics). Count matrices for snRNA-seq data from Singleron were generated by Singleron using CeleScope scopeV3.0.1 (kit V2) technology. Bulk RNA-seq count matrices were generated with the pipelines nf-core/rnaseq (v.3.18.0; doi:10.5281/zenodo.1400710) or AccuraCode (v1.2.0; for Singleron data). Imaging data were acquired with the Leica Application Suite X, or the FV31S-SW Fluoview software.

**Data analysis**
Sequencing data were analysed in an R environment (R v4.3.3), using the Seurat single cell analysis package v5.1.0, with the Signac extension (v1.13.0) for basic analyses. sccomp (v1.7.15) and MiloR (v1.10.0) were used for compositional analyses. DESeq2 v1.42.1 and Nebula (v1.5.3) were used for differential gene expression analyses. limma (v3.58.1) was used for batch correction of expression matrices. clusterProfiler v4.10.1 with annotations from DOSE(v 3.28.2) and org.Hs.eg.db (v3.18.0) were used for functional enrichment analyses. BSgenome.Hsapiens.UCSC.hg38 (v1.4.5) and EnsDb.Hsapiens.v86 (v2.99.0) were used for ATAC-seq mapping and annotations. scMEGA v1.0.2 was used for gene regulatory network analyses, using binding motifs from the JASPAR2024 package, v0.99.6. Protein-protein-interactions from the BioGRID database (v4.4.233) were retrieved using the BioGRID API via the R packages jsonlite (v1.8.8) and httr (v1.4.7). tidyverse (v2.0.0) with ggplot (v3.5.1), pheatmap(v1.0.12) and the ggraph package (v2.2.1) were used for general data analyses and visualisations. FIJI/ImageJ v1.53q was used for processing and analysing immunofluorescence images, using R packages above for downstream quantifcation. The used analysis pipelines based on these software tools will be made available on https://github.com/lattkem1/Down_Syndrome_Multiome

For manuscripts utilizing custom algorithms or software that are central to the research but not yet described in published literature, software must be made available to editors and reviewers. We strongly encourage code deposition in a community repository (e.g. GitHub). See the Nature Portfolio guidelines for submitting code & software for further information.

# Data

Policy information about availability of data

All manuscripts must include a data availability statement. This statement should provide the following information, where applicable:

- Accession codes, unique identifiers, or web links for publicly available datasets
- A description of any restrictions on data availability
- For clinical datasets or third party data, please ensure that the statement adheres to our policy

> Raw and processed sequencing data generated for this study will be made available under GEO accessions GSEXXX. The fetal tissue dataset will also be available as interactive online resource on the DISCO platform (https://www.immunesinglecell.org/tool/integration/ res/Down_syndrome). Other data will be provided by the authors upon reasonable request.

# Research involving human participants, their data, or biological material

Policy information about studies with human participants or human data. See also policy information about sex, gender (identity/presentation), and sexual orientation and race, ethnicity and racism.

| | |
|---|---|
| Reporting on sex and gender | Samples from both sexes (based on Karyotyping provided by the HDBR) have been included in this study and are reported in Supplementary Table 1. Preliminary analyses did not identify major sex-specific differences, therefore no sex-specific analyses are reported in the manuscript. |
| Reporting on race, ethnicity, or other socially relevant groupings | As the material is from banked foetal tissue, no information such information is available for the analysed samples. |
| Population characteristics | Samples were derived from foetal tissue from post-conceptional weeks 10 to 20. The estimated stage is reported in Suppl. Table 1. Beside the sex (see above), no other population characteristics were available. |
| Recruitment | Samples were donated by individuals undergoing surgical terminations of pregnancy in UK clinics to the Human Developmental Biology Resource (https://www.hdbr.org/). |
| Ethics oversight | The HDBR provides tissue under an umbrella licence covered by ethics approvals (London/Newcastle; REC approval 18/LO/0822 and 18/NE/0290). The individual project whose results are reported here have been approved by the HDBR (HDBR Project 200585). |

Note that full information on the approval of the study protocol must also be provided in the manuscript.

# Field-specific reporting

Please select the one below that is the best fit for your research. If you are not sure, read the appropriate sections before making your selection.

☒ Life sciences  ☐ Behavioural & social sciences  ☐ Ecological, evolutionary & environmental sciences

For a reference copy of the document with all sections, see nature.com/documents/nr-reporting-summary-flat.pdf

# Life sciences study design

All studies must disclose on these points even when the disclosure is negative.

| | |
|---|---|
| Sample size | The study used 20 CON and 19 DS fetal brain samples. The study aimed to cover human foetal samples from post-conceptional weeks 10-20 with each 5 CON and DS samples for weeks 10-12, 13-16 and 17-20. However, Sample sizes for human foetal tissue were determined and limited by the availability of this precious donated tissue, mainly for weeks 12 and 13. Therefore we acknowledge that our study has limited power to assess changes before PCW11 and after PCW14.<br>For in vitro experiments derived from a trisomic iPSC line and a isogenic control line were used, with at least 3 technical replicates per group. This is not powered to comprehensively detect differences, but sufficient to validate a large proportion of differences detected in the fetal samples. To assess independence of detected changes from inter-experimental and genetic variability, the basic characterization was repeated with multiple different cell preparations and a second pair of isogenic trisomic and control iPSC lines as outlined in the manuscript. 8 CON and 4 DS graft samples derived from a trisomic iPSC line and a isogenic control line were obtained to assess the ability of these grafts to model human fetal development and DS phenotypes. While this design reduces variability due to genetic background variability, it cannot account for changes that may be influenced by the diverse genetic backgrounds of the human population. |
| Data exclusions | As the donated fetal tissue was of very variable quality, and various other technical and biological parameters affected data quality (e.g. the RNase inhibitor used affected RNA library quality, regional identity), we have excluded 5 Control and 4 DS samples from our analyses, as well as low quality libraries and cells as described in detail in the manuscript (Suppl. Table 1). We based exclusion criteria on histological assessments of tissue quality, cell and sample level QC measures established in the field (e.g. transcript counts/cell, enrichment of ATAC reads in transcriptional start sites, ...), mapping of data to regions of a developing brain reference atlas and on the ability of the combined datasets to distinguish cell subtypes expected to be present at these stages (Suppl. Table 1).<br>Similar inclusion/exclusion criteria were applied to the human graft analysis (Suppl. Table 6). We also excluded one tissue bulk RNA-seq library which showed a transcriptome globally very strongly diverging from all 11 other samples, identified by principal component analysis |

(Extended Data Fig. 6c). This sample also showed poorer QC measures and an apparent lack of expression of a subset of Chr21 genes (not shown), which were very consistently upregulated in the 5 other DS samples.

**Replication**

While a replication of the fetal sequencing study with other samples was not feasible, we replicated key analyses with different parameter settings and/or computational tools, and focused on findings that were reproducible between different analyses. For example, exclusion of lower quality samples retained the reduction of RORB/FOXP1 neurons and a large fraction of the DEGs. Using an alternative differential expression analysis approach (Nebula) or published data (Extended Data Fig. 5), or considering developmental stage as covariate (not shown), confirmed the detected DEGs, and detected many other DEGs, but these additional DEGs were less consistent between different analyses. We replicated many changes in the in vitro model, using multiple cell preparations and an additional pair of isogenic cell lines, confirming robustness of the models (Fig. 4, Extended Data Fig. 9).

**Randomization**

We matched sex and developmental stage between CON and DS syndrome groups, as far as sample availability allowed (see Suppl. Table 1). As mentioned above, gene expression changes were largely retained if sex and developmental stage were considered as covariates, suggesting the absence of major biases.
For in vitro and graft studies, cells derived from isogenic pairs of cell lines were used to avoid bias due to different genetic backgrounds.

**Blinding**

As the study is based mainly on systematic global computational analyses without strong initial hypotheses, no systematic blinding was required.

# Reporting for specific materials, systems and methods

We require information from authors about some types of materials, experimental systems and methods used in many studies. Here, indicate whether each material, system or method listed is relevant to your study. If you are not sure if a list item applies to your research, read the appropriate section before selecting a response.

## Materials & experimental systems

| n/a | Involved in the study |
|---|---|
| ☐ | ☒ Antibodies |
| ☐ | ☒ Eukaryotic cell lines |
| ☒ | ☐ Palaeontology and archaeology |
| ☐ | ☒ Animals and other organisms |
| ☒ | ☐ Clinical data |
| ☒ | ☐ Dual use research of concern |
| ☒ | ☐ Plants |

## Methods

| n/a | Involved in the study |
|---|---|
| ☒ | ☐ ChIP-seq |
| ☒ | ☐ Flow cytometry |
| ☒ | ☐ MRI-based neuroimaging |

## Antibodies

**Antibodies used**

IF: Primary antibodies used were mouse anti-SATB2 (Abcam, ab51502, fresh frozen sections), rabbit anti-PAX6 (Biolegend #901301), rat anti-CTIP2 (Abcam, ab18465), mouse anti-SATB2 (Santa Cruz Biotechnology, sc-81376, paraffin embedded sections), rabbit anti-FOXP1 (Abcam, ab16645). Western Blot: antibodies used were mouse anti-Beta-Actin (Santa Cruz Biotechnology, sc-47778, 1:100), mouse anti-BACH1 (Abcam, ab128486, 1:2000), rabbit anti-PKNOX1 (ThermoFisher, PA5-30244, 1:750), mouse anti-GABPA (ThermoFisher, MA5-15419, 1:1000), goat anti-Rabbit-HRP (ThermoFisher, 31460, 1:5000) and goat anti-Mouse-HRP (ThermoFisher, 31430, 1:5000).

**Validation**

Mouse anti-SATB2 (Abcam, ab51502; 1:100) has been KO-validated by the supplier for applications including ICC/IHC (note that the antibody cross-reacts with SATB1). Mouse anti-SATB2 (Santa Cruz Biotechnology, sc-81376; 1:400) according to the supplier website has been validated for IF. Rabbit anti-PAX6 (Biolegend #901301; 1:200) according to the supplier website has been validated for IHC and reported to be also suitable for ICC. Rat anti-CTIP2 (Abcam, ab18465; 1:500) according to the supplier website has been validated for IHC-P and ICC/IF. Rabbit anti-FOXP1 (Abcam, ab16645; 1:200) has been validated by the supplier for mouse Foxp1 staining in IHC-P and IHC-FoFr and is predicted to similarly bind the homologous human FOXP1. According to the manufacturer's website it has also been used in numerous publications for human tissue.
All antibodies showed staining consistent with their expected region-specific expression.

## Eukaryotic cell lines

Policy information about cell lines and Sex and Gender in Research

**Cell line source(s)**

The neural progenitors from the isogenic C9 (CON) and C13 (DS) iPSC lines have been provided by the laboratory that has generated and tested these lines (see Murray et al., 2015, Alic et al., 2021). The isogenic DS2U (CON) and DS1 (DS) iPSC lines have been purchased from WiCell.

**Authentication**

WiCell provides extensive batch-specific characterization of the DS2U and DS1 iPSC lines.
The ability of all cell lines to differentiate into neurons has been verified. The increased expression of Chr21 genes in DS iPSC-derived neural cells in our study (Fig. 4, Extended Data Fig. 9) further confirms the retained trisomy of the DS cells used in our study.

**Mycoplasma contamination**

The cells have not been systematically tested for mycoplasma.

| Commonly misidentified lines<br>(See ICLAC register) | NA |
|---|---|

## Animals and other research organisms

Policy information about studies involving animals; ARRIVE guidelines recommended for reporting animal research, and Sex and Gender in Research

| Laboratory animals | Immunodeficient mice (JAX™ NSG, Charles River) aged between 3-5 months |
|---|---|
| Wild animals | NA |
| Reporting on sex | Both male and female mice were used. As only human-derived cells were analysed, no mouse-sex-specific analyses were performed. |
| Field-collected samples | NA |
| Ethics oversight | All mouse experimental procedures were approved by the Animal Care and Use Committee (IACUC) at Duke-NUS, Singapore (Ref. nr. 2022/SHS/1766), and ethics approved by the Institutional Review Board (NUS-IRB-2022-149), as well as by the Ministry of Health (MOH Ref: RR-2023/01) under the Human Biomedical Research Act guidance. |

Note that full information on the approval of the study protocol must also be provided in the manuscript.

## Plants

| Seed stocks | *Report on the source of all seed stocks or other plant material used. If applicable, state the seed stock centre and catalogue number. If plant specimens were collected from the field, describe the collection location, date and sampling procedures.* |
|---|---|
| Novel plant genotypes | *Describe the methods by which all novel plant genotypes were produced. This includes those generated by transgenic approaches, gene editing, chemical/radiation-based mutagenesis and hybridization. For transgenic lines, describe the transformation method, the number of independent lines analyzed and the generation upon which experiments were performed. For gene-edited lines, describe the editor used, the endogenous sequence targeted for editing, the targeting guide RNA sequence (if applicable) and how the editor was applied.* |
| Authentication | *Describe any authentication procedures for each seed stock used or novel genotype generated. Describe any experiments used to assess the effect of a mutation and, where applicable, how potential secondary effects (e.g. second site T-DNA insertions, mosiacism, off-target gene editing) were examined.* |

