## [Peer Review File · Nature Medicine]

Single-cell atlas of the developing Down syndrome brain cortex

Corresponding Author: Professor Vincenzo De Paola

This manuscript has been previously reviewed at another journal. This document only contains information relating to versions considered at Nature Medicine.

Version 0:

Reviewer comments:

Reviewer #1

(Remarks to the Author)

The authors addressed my comments in a satisfactory way and I recommend publication in Nature Medicine. However, the authors need to address some of the comments from reviewer 3, which are pertinent. In particular:

- it is important to show in the paper how removing the three samples mentioned by the authors affects the data, for transparency and reproducibility. Exclusion and re-inclusion of samples should be better specified in the methods, with the first and second versions of cut-off levels and changes that occurs with the inclusion/exclusion should be mentioned and shown.
- some of the samples that are indicated as excluded in the supplementary table (B11C1A, B11C1C, B12C3B), are in the figure of the included samples based on the UMAP of extended data 1c.
- Regarding the bulk analysis in Extended Figure 6c, the authors could use the PCA data to explain which is the major source of variation between the samples. The difference in change of the scale of the correlation can indeed be misleading, therefore the authors should use the same scale.
- when analyzing the code, it appears that the authors did not consider batches/samples in the integration. Is this the case?
- is there size correction the differential gene expression for the number of N of the samples or conditions to compare? Also, were the samples adjusted for differences in total counts, for the comparisons performed? This is important as pseudobulk analysis can lead to biases and it is possible that samples with more cells could show upregulation of genes but without biological meaning. Were patients/samples considered covariates for DGE analyses?
- For the sample integration. were any covariate used? Patient/sample/batch factor can influence downstream variation and results.
- in Extended Supplementary Table 1 includes the list of libraries used and the original Cellranger QCs. The authors need to add the final median stats per sample used, after QC, and final number of nuclei in each sample. This is important to determine whether there is any bias between samples/batches.

(Remarks on code availability)

Reviewer #2

(Remarks to the Author)

In the revised manuscript, the authors have introduced some modifications to address the pending issues. However, these changes appear somewhat incremental and do not specifically address the concerns previously raised by this reviewer. Consequently, the conclusions of the article still lack strong experimental support.

Major points:

- 1) The bulk RNAseq data have not been cleaned up and there are still two control samples clustering with DS samples. PCA clustering seems not “driven by variable cell-type proportions between samples” as stated by the authors but rather only by gene expression changes that indeed are very limited in this dataset possibly because authors still include the two samples that potentially misplaced. I would recommend remove these samples from both bulk and single-cell dataset and verify that indeed the global conclusions hold.
- 2) The overlap between bulk and single-cell is very limited. The authors claim that it is a sign of superiority of single-cell over bulk sequencing. Another possibility is that the datasets in this work are not robust enough.
- 3) In the in vitro sequencing experiments and transplanted organoids is still not clear the number of biological replicates (not technical !) analyzed. The point-by-point letter suggests that only one biological replicate was used. If this is the case, no robust conclusion can be drawn from these experiments, and they should possibly be discarded. Moreover, in the in vitro dataset the samples are not evenly distributed across genotype/developmental stage.
- 4) In many panels comparing datasets (Extended 6e, 4b, 5d), the scale is still not set between 0 and 1 as should be. This is visually increases the soundness of the results and potentially masks a very limited overlap between some of the datasets.
- 5) In my opinion, the conclusion of the knockdown experiments should be strongly dumped since no phenotypic rescue was found, and the applicability of ASO treatment to fetuses is questionable.

(Remarks on code availability)

Version 1:

Reviewer comments:

Reviewer #1

(Remarks to the Author)

In general the answers of the authors are acceptable and I consider that further analysis is not required. Since the reviews will be published with the paper, readers will be aware of the challenges of the analysis, and I therefore recommend acceptance of the paper.

(Remarks on code availability)

Revision 2 for Nature Medicine

Dear Joao,

Thank you for considering our manuscript NMED-RS146138-T "**Single-Cell Multiomics Identifies Chromosome 21 Regulators of Intellectual Disability Genes in Down Syndrome**" for revision at Nature Medicine. Please find below our answers to the additional comments of the reviewers and changes highlighted in the text. We hope that as such it will now be acceptable for publication in *Nature Medicine* as a Resource Article.

We thank you and the reviewers for the excellent suggestions, and all your efforts on our behalf.

With best wishes,

Vincenzo De Paola and Michael Lattke

Referees' comments:

Referee #1 (Remarks to the Author):

The authors have added some additional data to the original manuscript and clarified via text modifications and in the rebuttal to the reviewers some of the questioned that were posed.

This is defenetly making the MS stronger. I still have some concerns about true novelty, relative to work already published, and lack of real validation.

- 1.1) We thank the Referee for recognizing that our revisions have strengthened the manuscript and for their additional comments. The transcriptional and chromatin changes underlying Down syndrome during human brain development have never been examined at cellular resolution. Our study advances the field in several key and, to our knowledge, novel ways:
1. **Comprehensiveness:** Our dataset represents the largest single-cell atlas of the developing human Down syndrome cortex to date, comprising 30 fetal samples (~250,000 high-quality cells). This unprecedented depth enables the identification of subtle yet biologically significant molecular and cellular phenotypes that were previously unresolved.
 2. **Benchmarking across models:** We provide the first systematic comparison of widely used *in vitro* human models and a novel *in vivo* DS xenograft system against primary fetal tissue. This benchmarking framework offers an essential validation resource for the field.
 3. **Functional validation and therapeutic proof-of-concept:** Moving beyond prior descriptive studies, we identify three novel Chromosome 21 transcription factor targets and demonstrate rescue of dysregulated genomic programs using antisense oligonucleotides (ASOs). These discoveries form the basis of a **provisional patent filed** with Imperial College London, which by definition required evidence of novelty, inventiveness, and real-world applicability—further supporting the originality and translational impact of our work.

Confidential Information Redacted

For example, the layer 4 phenotype is potentially novel and interesting but the only validation of the RNAseq is an immunohistochemistry for two genes, RORbeta and FOXP1. Why just two genes? FOXP1 is not even specific to just those neurons. did the other markers not agree with the RNAseq?

1.2) We thank the reviewer for this important point regarding the validation of the RORB/FOXP1 neuron phenotype. We agree that a comprehensive validation is crucial for this potentially novel finding.

Our choice to focus on FOXP1 for immunohistochemical validation was data-driven. As shown in our RNAseq data (Fig. 1d), FOXP1 demonstrated a level of enrichment in this specific neuronal population that was comparable to RORB and significantly higher than other potential markers, making it a strong candidate for validation.

We also attempted multiple validation approaches for RORB, however these turned out to be technically extremely challenging. Antibodies for RORB immunostaining are not very reliable, and we were not able to achieve specific staining with the antibody we tested. RORB RNAscope suggested a reduction in DS in one pair of donors, but we encountered prohibitively high background in other donor pairs, which prevented robust and reliable quantification across the cohort.

Together with our additional computational validation confirming the observed reduction of RORB/FOXP1-expressing neurons (MiloR, Extended Data 2e), our clear and quantifiable FOXP1 results from a large sample set (7 control, 6 DS, plus 4 paraffin-embedded samples), provides a multi-faceted and robust validation of the phenotype. While we agree that this important finding warrants further investigation, we have now revised the discussion to explicitly acknowledge the need for future studies to identify additional markers.

the justification for the need to transplant into an adult host is not very satisfying, although doable, as the authors mentioned (citing prior transplantation experiments that transplanted organoids for completely different reasons, not to prove a developmental phenotype). Should the manuscript advance to publication, I think that these limitations need to be more clearly stated.

1.3) We thank the reviewer for raising this important point about the rationale for using an adult host. Our goal was to generate xenografts to assess cell-autonomous and network-level human phenotypes while minimizing confounds from host integration. Transplantation into a developing brain, though useful for studying integration, produces highly chimeric circuits that obscure intrinsic human cell phenotypes. In contrast, the adult brain provides a less plastic environment, allowing a clearer assessment of human-specific maturation and function (Real et al., Science 2018). Our methodology is in line with this body of work, which has successfully used adult hosts to investigate key aspects of human neuronal development and function *in vivo*. We chose this paradigm specifically to unmask disease-relevant phenotypes in a human-enriched network context, rather than to study cross-species integration.

We now clarify this rationale in the Results and explicitly note in the Discussion the limitation that heterochronicity is biologically inherent to all xenotransplantation models.

Referee #2 (Remarks to the Author):

The authors did an extensive work to address my points and the ones raised by the other reviewers, and the article is much improved. Nevertheless, the *in vitro* down regulation of the expression of the identified "key" transcription factors, while suggesting that these TFs can indeed regulate the expression of a small subset of identified DS associated genes, does not demonstrate the key role of these TFs for DS at a functional level. The humanized mice is a very interesting new DS model, and the new data analysis shows that it might be much more powerful than the *in vitro* models to study DS, but the authors still do not use this to explore if the "key" TF functionally contribute to DS, which would be important for this paper.

2.1) We thank the Referee for their positive assessment and for acknowledging the extensive work performed to improve the manuscript. The Referee raises an excellent point regarding the functional validation of these TFs *in vivo*. We completely agree that demonstrating their functional contribution to DS phenotypes within our humanized mouse model would be the ultimate validation of their 'key' role. This is, without question, the most important next step. However, the experiments required to functionally test these TFs *in vivo*—involving complex ASO delivery optimization, breeding and maintenance of large, sufficiently powered xenografted animal cohorts, and analysis of multiple cellular and behavioral readouts,

represent a major, multi-year research project. Therefore, while we agree on the experiment's importance, it falls significantly beyond the scope of the current manuscript. The primary contribution of this paper is to establish and validate this new humanized model and to provide a comprehensive transcriptomic resource that identifies high-priority therapeutic targets. The work we present here is the essential foundation that now makes such future functional studies possible. To directly address the Referee's point within the manuscript, we have added the following statement to the Discussion section: "*A critical future direction will be to functionally validate the role of these key TFs by testing whether reversing their intellectual disability gene target deregulation can rescue core DS phenotypes, including the reduced synchronized neural network activity we recently identified^{48,59}. Such in vivo experiments in our humanized mouse model, while beyond the scope of the present study, are essential for confirming their contribution to DS pathology. Notably, a recent study demonstrated that a similar transcriptional network correction strategy can counteract impairments in an AD model⁵⁴, supporting the feasibility of this approach.*"

We believe this addition clarifies the scope of our current findings and properly frames the exciting future work that our new model enables.

In general, the article is an excellent DS transcriptomic resource, but the level of conceptual advance might still make it more appropriate rather for a more specialized journal in the genetics or neuroscience fields.

2.2) We thank the Referee for recognizing our work as an "excellent DS transcriptomic resource." We respectfully disagree with the assessment that the work is more suited for a specialized journal and would like to take this opportunity to clarify the key conceptual advances we believe our study provides for a broad audience.

Confidential Information Redacted

1. **A Critical Benchmarking of Existing Models:** We are the first to systematically benchmark the most widely used *in vitro* human models (i.e., human iPSC-derived neurons) and a novel preclinical *in vivo* humanized DS system against the ground truth of primary fetal tissue. This critical evaluation reveals the specific molecular pathways that are faithfully recapitulated versus those that are not. This provides an essential, practical roadmap for the entire field, guiding researchers on which models to use to study specific aspects of DS pathology. This is a conceptual advance in how the field can and should approach the study of Down syndrome.
2. **A Bridge from Description to a New Therapeutic Strategy:** Building on our atlas and the hypothesis that specific transcription factors act as key drivers of dysregulation, we provide a concrete proof-of-concept for a therapeutic approach. By demonstrating that ASO-mediated knockdown of these TFs restores key DS-associated gene targets, our work establishes a direct and actionable link between molecular pathology and targeted therapeutic intervention.

In summary, while the atlas itself is a foundational resource of unprecedented value, our manuscript also provides important conceptual advances, critically evaluating existing research tools and demonstrating a viable therapeutic strategy. We believe these contributions, which address fundamental challenges in both basic and translational neuroscience, are of significant interest to the broad readership of Nature Medicine (**see also 1.1**).

Referee #3 (Remarks to the Author):

The revised manuscript by Lattke et al., has been extensively transformed from the first version both in terms of the analysis pipeline and new experiments. In this new version, authors have tried to circumvent the main issues found by all three referees, but some of those still remain, in my opinion. Indeed, despite the extensive transformation, some issues

have been only marginally addressed, and the manuscript remains a mainly descriptive study lacking true functional and mechanistic insights.

3.1) We thank the referee for their continued engagement and for acknowledging the extensive transformation of the manuscript. While it is true that our study provides a comprehensive resource, **it goes significantly beyond description**. In addition to presenting the most extensive single-cell atlas of the DS fetal cortex to date, we uniquely:

1. **Benchmark widely used *in vitro* and *in vivo* DS models** against human fetal tissue, providing essential contribution to the entire field. This is a critical insight that directly informs which models are suitable for future mechanistic and therapeutic studies and, therefore, a conceptual advance for the field.
2. **Identify three Chr21 transcription factors** as upstream drivers of disease-relevant transcriptional changes, a mechanistic insight that is independently corroborated by other groups.
3. **Demonstrate Mechanistic Rescue via Functional Perturbation:** Based on our atlas, we hypothesized that specific TFs are key drivers of dysregulation. We then moved beyond description to direct functional perturbation. By using ASOs to knock down these factors, we demonstrated a proof-of-concept rescue of the downstream genomic program. This experiment provides a clear mechanistic link between the identified TFs and the disease relevant molecular state. It shows that these factors are not only associated with molecular pathology, but that targeting these TFs can actively reverse specific aspects. Given that loss of function, or in some cases even haploinsufficiency, of a single intellectual disability gene can cause cognitive impairment in humans, our findings that expression of many of these genes is altered by the TFs establish a mechanistic link between chromosome 21 transcriptional dosage and the neurodevelopmental pathology of DS.

Together, these elements constitute a complete scientific arc: from comprehensive observation (the atlas), to hypothesis generation (TF targets), to functional validation and mechanistic proof-of-concept (the ASO rescue). This is fundamentally different from a purely descriptive resource paper and provides significant conceptual and mechanistic advances for the field.

Main Points

Sample size:

In the rebuttal letter, the authors claim that their dataset “is larger than most published single-cell studies on fetal human brain” citing early studies that were however performed on control human tissue and not dealing with a disease, as in this case. Differential gene expression studies in health and disease over different time points require more samples. To overcome the issue of the small sample-size across the different gestational ages, the authors have now relaxed their inclusion criteria and cutoff values and re-introduced in the dataset some lower-quality samples previously discarded. Moreover, the authors have discarded 3 samples that did not show cortical identity. The new dataset is now resulting from 15 control and 15 DS samples. However, including suboptimal samples may have deleterious effect on overall data quality and it is not clear how the reintroduced samples were chosen. In extended data Fig1B, violin plots are shown for 46 samples, whereas in extended data Fig1C UMAPs are shown for 41 samples. However, it is not indicated which are the 30 samples used for analysis, and which are the one excluded. Please, also use a color code to identify control and DS samples in these panels.

3.2) We thank the Referee for these important questions regarding our dataset's composition and quality control. We have taken these concerns very seriously and provide the following clarifications, along with updates to the manuscript and figures.

1. Sample Size and Statistical Power: We agree that studies of disease states across developmental time points benefit from large sample sizes. Our statement comparing our dataset size was intended to contextualize it within the challenging field of primary human fetal tissue research, where sample availability is a major limitation. Our final cohort of 30 samples

(15 DS, 15 control) represents the largest single-cell dataset of the developing DS cortex to date.

Crucially, the statistical power of our dataset is demonstrated by its ability to identify robust biological signals that are independently validated. In particular, the identification of the three Chr21 TFs as upstream drivers has been independently reproduced in a comparable study (Vuong et al., bioRxiv 2025). Moreover, our ASO experiments confirmed several of their predicted downstream targets, underscoring that our dataset is sufficiently powered to reveal reproducible and biologically meaningful effects.

2. Data Quality and Sample Inclusion Criteria: We thank the reviewer for these critical questions regarding our dataset's quality and composition. We agree that balancing sample size with data quality is paramount for a robust disease study, and we apologize for the lack of clarity in our initial revision.

First, we want to clarify the rationale behind our sample inclusion process. Our original QC thresholds were intentionally set to be extremely stringent. In response to the reviewer's valid concern about statistical power, we performed a careful re-evaluation to determine if these thresholds could be moderately relaxed without compromising the integrity of our findings. This was not a simple inclusion of "suboptimal" samples, but a principled reassessment of the trade-off between sample number and per-sample quality metrics (see Methods '*Basic processing and quality control of Multiome data from fetal tissue*' and **Supplementary Table 1, where we explicitly annotated** which libraries were excluded from the final analysis and why).

Crucially, this expanded dataset (now N=15 DS, N=15 Control) confirmed our key findings, including the loss of RORB/FOXP1 neurons and the central role of Chr21 TFs. The fact that the central biological conclusions remain stable demonstrates their robustness and shows that the benefit of increased statistical power from a larger 'N' was not undermined by the modest change in QC thresholds.

The reviewer correctly notes a reduction in the number of differentially expressed genes (DEGs). This change primarily reflects the exclusion of three non-cortical samples identified through mapping to a multi-region reference dataset (Bhaduri et al., 2021). These samples, which we have now excluded, were a source of substantial technical bias and contributed to many false-positive DEGs.

Therefore, the current dataset, while yielding fewer DEGs, produces a more accurate and robust set of results by eliminating this major confounder.

3. Clarity of Extended Data Figure 1: We sincerely apologize for the confusion caused by the previous version of this figure. We have now revised **Extended Data Fig. 1** and its legend to be perfectly clear. Specifically, we have:

- **Added a clear color code** to distinguish Control and DS samples throughout all panels.
- **Clarified in the legend** that the QC plots in panels b and c visualize individual sequencing *libraries*, not aggregated tissue *samples* (as some samples generated multiple libraries).

We also want to refer to **Supplementary Table 1, where we explicitly annotated** which libraries were excluded from the final analysis and why (e.g., non-cortical identity; or low number of cells after filtering for low quality cells or high fraction of excluded low quality cells according to the cell-level filtering criteria explained in the Methods).

We are confident that the revised figure and analysis directly address the Referee's concerns and demonstrate the robustness and integrity of our dataset and its conclusions.

Difference in population: one new interesting finding of the manuscript remains the reduction in the number of L4 pyramidal neurons expressing Foxp1. However, the authors failed to find this phenotype in either iPSC derived neurons in vitro or upon transplantation in vivo. Any opinion of this?

3.3) This is an excellent point, and it touches on a critical aspect of modeling complex neurodevelopmental disorders. The discrepancy between our primary tissue findings and the

model systems does not invalidate the finding, but rather provides important insight into the current limitations of these models. We have clarified this in the manuscript.

1. Regarding the *In Vivo* Transplant Model: The primary challenge in detecting this specific phenotype in the xenografts lies in the complex nature of transcriptomic mapping between two biologically distinct systems: developing primary fetal tissue and maturing transplanted neurons. Mapping algorithms prioritize the largest sources of variance. In this context, the global maturational state of the graft neurons often outweighs the subtler signatures of neuronal subtype identity.

We do observe a consistent trend towards a reduction in graft cells mapping to the corresponding upper layer population (NEU_CUX2_s2), which also displays reduced *FOXP1* expression (Fig. 5e), suggesting that the graft population corresponding to the fetal RORB/*FOXP1* cells might be at a maturation state more similar to that of fetal NEU_CUX2_s2 cells, mapping therefore to this population instead. While this trend does not reach the same statistical significance as in the primary tissue, it is consistent with our primary finding and suggests the phenotype may be partially recapitulated, albeit masked by the dominant effect of the cells' different differentiation stage.

2. Regarding the *In Vitro* Models: This analysis was not performed in our *in vitro* models because these systems are not designed to faithfully recapitulate the full diversity of cortical neuronal subtypes. Current protocols for generating iPSC-derived neurons predominantly yield lower-layer projection neurons, generated in the early stages of neurogenesis accessible in 2D cultures. Furthermore, *FOXP1*-positive L4 neurons are vulnerable to degeneration in the presence of stressors (e.g. in Alzheimer's disease, see Leng et al., 2021; and likely the artificial *in vitro* environment). Therefore, assessing this specific population change would not be an appropriate or meaningful use of this model system.

Conclusion and Manuscript Update: Taken together, the robust finding in primary fetal tissue represents the ground truth. The partial recapitulation in the *in vivo* model and the known limitations of the *in vitro* system highlight that fully modeling the nuanced process of cortical layer specification remains a key challenge for the field. To ensure this is clear to the reader, we have added the following statement to the Discussion: "*While we observed a significant reduction in L4/*FOXP1*+ neurons in primary DS tissue, this phenotype was not fully recapitulated in our model systems. This likely reflects the known challenges of in vitro models in generating upper-layer neuronal diversity and the technical complexities of mapping cell subtypes across different maturational states in xenografts. This highlights an important area for future model development.*"

Moreover, recent snRNAseq studies during DS fetal development and early postnatal period have not observed such decrease (Risgaard et al., bioRxiv 2025; Vuong et al., bioRxiv 2025).

3.4) We thank the Referee for raising this important point regarding the recent literature. While at first glance these studies may seem contradictory, a closer examination of their analytical strategies and results reveals that they are, in fact, consistent with our findings.

1. Methodological Differences Preclude Direct Comparison: The apparent discrepancy is largely explained by differences in analytical strategy.

- **Vuong et al.** employ a clustering and quantification strategy that combines L2/3 and L4 neurons into a single population. This approach, by design, would not be able to detect a specific reduction in the L4 subpopulation.
- **Risgaard et al.** have a smaller sample size (5 DS vs. 5 Control), which may limit the statistical power required to detect changes in less abundant cell populations like L4 neurons.

2. Supporting Trends Within the Cited Literature: Despite these limitations, the data in the cited preprints actually support our conclusion. We note that in **Risgaard et al., their "L3-5 IT" population**, which encompasses the L4 neurons, shows a clear trend towards reduction in the DS samples (see their Fig. 1D). This is entirely consistent with our more highly powered observation of a significant decrease. Furthermore, our discovery of L4 reduction aligns with previously published observations in adult DS brains (Palmer et al., 2021) and Alzheimer's disease (Leng et al., 2021), a pathology with known links to DS.

Differential gene expression:

The new analysis has largely changed the results of the differential gene-expression analysis compared to the previous version of the manuscript. In the first version the authors detected 1,462 differentially expressed genes from outside Chr21 mainly in the excitatory lineage, with most of the DE genes downregulated. The new analysis instead shows 732 DE genes (as before mainly in the excitatory lineage) but most of these genes are upregulated, this time. Such discrepancy is hardly explainable and seem to point to a general inconsistency of the dataset, perhaps due to uneven quality of samples? The validation attempts with an alternative analysis (Nebula) or comparison with a published bulk RNAseq dataset indicate that downregulated genes are underrepresented in the new dataset.

3.5) We thank the referee for this important point. The discrepancy in the DEG list is a direct consequence

of the substantially improved analytical rigor that has removed a major confounding artifact in our

revised manuscript.

The primary reason for the shift in DEGs was the stringent removal of the few but impactful non-cortical samples contaminating the cortical cell populations. In our previous analysis, genes highly expressed in these non-cortical cells (but absent in true cortical cells) were being incorrectly identified as strongly downregulated in DS. This created a powerful but misleading statistical signal, artificially inflating the list of downregulated genes. By removing these contaminants, we eliminated this artifact, revealing a more accurate, albeit more subtle, transcriptional signature of DS in the cortex. Therefore, the discrepancy the referee noted was due to a resolved analytical artifact in the original analysis. Crucially, this refined set of DEGs shows strong concordance with multiple independent lines of evidence. It is not only validated by orthogonal statistical models (Nebula) but also shows significant overlap with previously published datasets and our own bulk RNA-seq data from both primary tissue and *in vitro* models. This multi-modal validation gives us high confidence that the current DEG list is a more accurate and biologically meaningful representation of the core transcriptional changes in the DS cortex.

The lower rate of detection and validation of downregulated genes is expected. The direct upregulation of Chr21 genes, driven by gene dosage, represents a primary and robust change that consistently emerges across DS datasets. In contrast, downregulated genes likely reflect secondary, indirect consequences of these primary alterations. As a result, their expression changes are expected to be more variable between individuals, and, therefore, less likely to be statistically detected across different datasets.

Moreover, bulk RNAseq from some of the samples (why not all of it? Please, give a justification in the manuscript) used for snRNAseq from PCW11–14 showed only few DE genes (mainly from Chr21. Extended data fig. 6d) and limited overlap with snRNAseq data (see Extended data fig. 6e where scale is set between 0 and 0.7 instead of 0 to 1). Importantly, this bulk RNAseq experiment points to some issues with the samples, as suggested by the PCA analysis (Extended data fig. 6c) showing two control samples (B12C4 and B11C2) clustering with the DS samples.

3.6) We thank the referee for these insightful points, which highlight the inherent limitations of bulk analysis and underscore the necessity of our single-cell approach. Regarding the selection of samples ("Why not all of it?"), we strategically prioritized our precious primary tissue for the single-cell analysis, which is the core of our study. The bulk RNA-seq was a secondary validation using a representative subset of samples. We will clarify this rationale in the text.

Regarding the PCA clustering and limited DEG overlap: These are not "issues with the samples" but are the expected consequences of cellular heterogeneity in bulk tissue.

- The PCA clustering is driven by variable cell-type proportions between samples, a much stronger signal than the disease state itself.
- This same heterogeneity dilutes the cell-type-specific, non-Chr21 DEG signals below the threshold of detection, leaving mainly the signals of globally deregulated Chr21 genes.

Indeed, this analysis confirms that our single-cell resolution analysis is essential to deconvolve the complex pathology of Down syndrome.

Gene regulatory network analysis:

The new use of scMEGA instead of the DESeq2 for ATAC-seq analysis looks much more appropriate. The new gene-regulatory analysis is interesting. However, it remains a correlative prediction unsupported by experimental data. Indeed, the fact that a predicted Transcription Factor (TF) binding site lies in an accessible chromatin region does not automatically imply that the TF is actually binding to that region. Wet lab work would highly strengthen this interesting finding.

- 3.7) We agree with the referee's principle that computational predictions require experimental support. We therefore employed a two-tiered validation strategy: **First**, as a computational validation, we cross-referenced our predicted TF binding sites with publicly available ChIP-seq datasets. This confirmed that a significant fraction of our predicted sites are indeed bound by the relevant TFs in experimental settings, validating the underlying predictive model. **Second**, and more importantly, we moved beyond confirming binding to test for **functional relevance**. We believe a more direct and powerful test than ChIP-seq is to perturb the system and measure the downstream consequences. Our ASO-mediated knockdown experiments do precisely this. They directly test whether these TFs are necessary for the dysregulation of their predicted target genes.

The resulting rescue of a key subset of these target genes provides direct experimental evidence that these computationally-identified TF-gene interactions are not merely correlative, but are functionally active and relevant to the disease pathology. Given that loss of function, or in some cases even haploinsufficiency, of a single intellectual disability gene can cause cognitive impairment in humans, our findings that the Chr21 TFs regulate many of these genes establish a mechanistic link between chromosome 21 transcriptional dosage and the neurodevelopmental pathology of DS.

While ChIP-seq would confirm binding at specific loci, many TF binding events are not functionally consequential for regulating gene expression. By contrast, our functional perturbation provides a more critical insight: that these TFs are relevant drivers of the downstream transcriptional program. We will clarify this rationale in the revised manuscript.

Knockdown of some of Chr21 transcription factors:

The authors have performed bulk RNAseq on iPSC-derived NPC and neurons from two isogenic pairs of cell lines and found some correlation with the fetal tissue expression data, again mainly in the upregulated genes (although the scale of Fig 4b is set between 0 and 0.5 rather than 0 to 1). In these experiments it is not clear why 2 batches of NPC sequencing from lines DS1/DS2U and only 1 for all the others (NPC C13/C9, iNEU DS1/DS2U and iNEU C13/C9) are indicated. Do batches indicate biological replicates? If this is the case, please make number consistent across experimental cases, by performing at least sequencing from two biological replicate for each cell line and each differentiation time point and averaging. Otherwise, please, explain.

- 3.8) We thank the referee for raising this important point about the experimental design, which allows us to clarify our strategy. Each 'batch' represents a fully independent, side-by-side differentiation experiment from a given cell line pair. Our strategy was to prioritize testing the **robustness of our findings across multiple genetic backgrounds (two isogenic pairs) and developmental stages (NPCs and neurons)** rather than performing deep technical replication of a single condition.

We believe this approach provides a more powerful validation for two reasons:

1. It confirms that the observed DEGs are a consistent feature of the DS phenotype, not an artifact of one specific cell line or differentiation state.
2. It directly demonstrates the generalizability of our findings, which is critical for a study with translational implications.

Given that this was a validation experiment, we believe this design represents the most rigorous and informative use of resources to confirm the functionality of the system.

Next the authors, have knockdown the expression of three Chr21 Transcription Factors (TFs) with antisense gapmers *in vitro* in iPSC-derived NPC and found that the expression of few of their putative transcriptional targets was changed. I believe that this part was conceived as the functional and mechanistic validation of the hypothesis drawn from the previous analysis, but I have some concern with the rationale of the experiment. In my eyes, these results are fairly expected, as it is one would predict that knockdown of a transcription factor will change the expression of some of its predicted targets. Most importantly, this experiment does not reveal much about the main finding of the study (decreased L4 neurons). To establish, a causal link the authors have to perform the knockdown of the TFs and assess the proportion of the different excitatory neuron subclasses and rescue the L4 neuron phenotype. However, from the data shown in Fig 4d, none of the knockdown is actually changing Foxp1 expression. Moreover, Foxp1 expression was increased rather than decreased in iPSC-derived NPC (Fig. 4c) *in vitro*. Therefore, the authors should consider reducing TFs expression in iPSC-derived neurons in grafted cells *in vivo* were Foxp1 is at least downregulated in layer2/3 (CUX2) neurons. Regarding Foxp1 expression, in panel 4c the expression is increased in DS NPC *in vitro*, whereas in panel 4d (in DS vs CON, PKNOX1 targets) is actually decreased, please give an explanation.

3.9) We thank the referee for this critical feedback, which allows us to clarify the specific aims and significant findings of our functional experiments.

1. The ASO Experiment is a Critical Mechanistic Validation: While it is expected that knocking down a TF changes *some predicted* targets, our validation experiment addresses two critical questions: the accuracy of these predictions, and which predicted targets are truly regulated by these TFs (see point 3.7). Our experiment provides the first direct, functional evidence linking these specific Chr21 TFs to the dysregulation of a subset of their predicted downstream targets. This moves our findings from a computational correlation to a validated mechanistic interaction. This is a crucial proof-of-concept for any future therapeutic strategy and is a primary functional insight of our study.

2. The Goal Was to Validate the Regulatory Network, Not Rescue a Single Cellular Phenotype: The primary aim of this *in vitro* experiment was to validate the predicted TF-target regulatory network, which we successfully did. We did not design it to rescue the complex L4 neuron phenotype. As the referee correctly notes, and as our own data show, the *in vitro* system does not fully recapitulate the *in vivo* FOXP1 downregulation. This is an important finding in itself, as it highlights the limitations of the *in vitro* model and underscores the value of our *in vivo* xenograft model for future studies.

The logical next step, which the referee suggests—performing the ASO knockdown in the *in vivo* xenograft model—is precisely the core of our ongoing, multi-year preclinical program. This is a major research project that is well beyond the scope of the current foundational resource (see also 2.1).

3. Clarification of FOXP1 Expression (Fig 4c vs 4d): The referee is correct that FOXP1 expression *in vitro* is inconsistent. Our data reveal clear cell-line-specific effects: the DS1/DS2U line shows upregulation of FOXP1, while the C9/C13 line shows a *downregulation* consistent with the fetal tissue phenotype. This demonstrates that the effect of Trisomy 21 on this key regulator is highly sensitive to genetic background in simplified 2D culture systems. **Crucially, this *in vitro* variability is resolved in our *in vivo* model.** When the same DS1/DS2U line (which showed the "wrong" phenotype in a dish) is xenografted into a physiological brain environment, it **correctly recapitulates the FOXP1 downregulation** seen in primary human fetal tissue. This is a powerful demonstration that the *in vivo* environment provides essential cues that buffer against cell-line-specific artifacts and produce a more robust, physiologically relevant disease phenotype. We have clarified this point in the results and ensured all figures are clearly labeled to reflect the cell lines used.

Human neural cells transplantation:

The authors have increased the sample size and changed the isogenic pair of cells in comparison to the previous version of the manuscript. Although the xenograft system may have some advantages compared to the *in vitro* system it still fails to model the developmental deficit claimed by the authors in human DS fetuses (L4 neurons reduction). The only

statistically significant differences are an increase in astrocytes and a decrease in proliferating cells in DS grafts. Regarding Foxp1 expression, this is decreased in the putative layer 2/3 neurons (CUX2), rather than in L4. Moreover, the authors claim that the *in vivo* grafts show stronger correlation in gene expression changes with the human tissue compared to *in vitro* cultures. However, here, they are comparing iPSC-derived cells at two completely different maturation stages: cells differentiated for 30 days *in vitro* versus cells grafted for 12-24 weeks, which have had more time to acquire a more terminal differentiated phenotype.

3.10) We thank the referee for these insightful comments, which touch upon the core rationale and interpretation of our xenograft model. We appreciate the opportunity to clarify our perspective and

have revised the manuscript to make these points more explicit.

1. The Xenograft Model Reveals a More Mature, Clinically Relevant DS Phenotype. As the referee points out, our xenograft model shows increased astrocytes and decreased proliferation, which is not showing a limitation, but a key strength of the model. These features are established hallmarks of DS neuropathology. Our xenograft model, with its extended maturation period (12–24 weeks), offers a unique window into later stages of Down syndrome cortical pathology that are otherwise difficult to study due to the scarcity of late-stage human fetal tissue. While not intended to replicate early fetal development, its value lies in **modeling more mature disease stages** and capturing the dynamic progression of pathology over time, providing an essential platform for preclinical studies targeting these later-emerging phenotypes. The *in vitro* system is limited in its ability to achieve the full maturation seen *in vivo*, as maintaining cultured neurons for 12-24 weeks, as in the xenografts, is extremely challenging. We have revised the Discussion to more clearly frame the xenograft as a model of later-stage cortical development, complementing studies of early fetal tissue.

2. FOXP1 Downregulation. The referee insightfully notes that FOXP1 downregulation occurs in CUX2+ upper-layer neurons, an apparent discrepancy with the fetal L4 phenotype. The absence of a corresponding signal in L4 neurons is likely due to technical mapping challenges, where the algorithm prioritizes maturation state over precise subtype (see 3.3).

Minor points

In the section “integrated gene regulatory network ...”, clarification would be helpful regarding the basis of the comparison with ChIP-seq data. Specifically, was the ChIP-seq dataset obtained from the ChIP Atlas database filtered according to the developmental period (e.g., PCW 10–20) during which the transcription factors were identified?

3.11) We thank the referee for this question. Context-specific ChIP-seq data from the developing human cortex is extremely scarce. We have clarified in the manuscript that therefore we mined all available ChIP-seq data from human cells. While this does not prove binding in fetal human tissue, it confirms **experimentally** that the TFs are generally able to bind the predicted regulatory elements, and are statistically more likely to bind these elements than candidate regulatory elements predicted not to bind these TFs, providing a strong independent validation of our computational model.

Protein–protein interactions (PPIs) are known to vary across developmental stages and conditions. From the description, it appears that general PPI datasets (established under varied contexts) were used to correlate transcription factor activity. Further explanation on how such interactions can be assumed to apply to the current experimental design would strengthen the interpretation. Given that these analyses are predictive and correlation-based, the conclusions could be reinforced by complementary experimental approaches. For instance, the three transcription factors identified in this study, along with their predicted targets, might be more directly validated through focused assays such as CUT&Tag or CUT&RUN.

3.12) We thank the referee for this important methodological point regarding the interpretation of our computational analyses.

The referee is correct that the publicly available protein-protein interaction (PPI) and ChIP-seq datasets we used were not generated in the specific context of the developing human cortex. This is a limitation. Our intention in using these resources was to leverage the wealth of existing data to narrow down high-confidence, testable hypotheses about the key transcriptional regulators involved in the DS phenotype. We have now added a sentence to the Discussion to explicitly state this limitation and to clarify that these analyses serve to generate hypotheses rather than to definitively prove interactions in our system.

The referee suggests that direct binding assays like CUT&Tag would be a valuable approach to validate these predictions, and we agree that such experiments would provide excellent evidence for physical occupancy at target gene promoters. However, our primary goal in this study was to move beyond demonstrating physical binding to test the **functional consequence** of these predicted TF-target interactions. As a transcription factor can bind to thousands of genomic sites without necessarily regulating the expression of all nearby genes, a binding assay alone would not confirm a functional role in the DS-associated dysregulation we observe.

For this reason, we chose to perform ASO-mediated knockdown experiments. This approach directly tests a more stringent and functionally critical hypothesis: Is the candidate TF *necessary* for the observed dysregulation of its predicted target genes? The successful rescue of key target gene expression provides direct experimental proof that some of these predicted regulatory interactions are functionally active and contribute to the transcriptional dysregulation underlying Down syndrome.

We believe our functional validation provides a more direct and compelling test of our hypothesis than a binding assay alone in this context.

I could not access the interactive online resources to navigate the datasets at CELLxGENE and DISCO. Please, fix this.

We apologize, we have made the datasets now available on CELLxGENE for the reviewers to access. As our dataset after a major update of DISCO resource is currently unavailable there, we have removed this link. We aim to include this in the final manuscript. As is standard practice we confirm that these resources will be made fully and publicly available without restriction upon publication.

Reviewer #1 (Remarks to the Author):

The authors addressed my comments in a satisfactory way and I recommend publication in Nature Medicine.

We thank the reviewer for the positive evaluation.

However, the authors need to address some of the comments from reviewer 3, which are pertinent. In particular:

- it is important to show in the paper how removing the three samples mentioned by the authors affects the data, for transparency and reproducibility. Exclusion and re-inclusion of samples should be better specified in the methods, with the first and second versions of cut-off levels and changes that occurs with the inclusion/exclusion should be mentioned and shown.

1.1.1) We thank the reviewer for this important suggestion to further enhance the transparency of our sample exclusion process. We are happy to share comparisons of our analyses under different inclusion and exclusion criteria. However, we feel that incorporating all these extensive comparisons directly into the manuscript would compromise its conciseness. If the editor considers it useful, we would be pleased to discuss an appropriate format to make these analyses publicly available.

Figure 1.1.1a summarizes the four main sets of inclusion criteria we applied at different stages of the analysis:

- (1) The original analysis, which used excessively stringent filters and included three non-cortical samples that biased the differential expression results.
- (2) A more moderately stringent analysis retaining cells with less ATAC reads and libraries with higher fractions of removed low quality cells.
- (3) The analysis presented in the latest version, as in (2), but without the non-cortical samples (to avoid differential expression biases driven by these samples).
- (4) An analysis that additionally removed the potential outlier samples identified in the bulk RNA-seq dataset, as raised by reviewer 3 (now reviewer 2).

Importantly, **the selective reduction of RORB- (and FOXP1-) expressing excitatory neurons was consistently detected across all four analyses** (Fig. 1.1.1b).

1.1.2) During preparation of the first revision for Nature, we noted that three samples contained a disproportionately large fraction of cells mapping to non-cortical regions in the reference atlas—samples that were unexpectedly not flagged as outliers by the DESeq2 algorithm. Their inclusion led to the erroneous identification of regionally enriched genes as differentially expressed (Fig. 1.1.2a/b). Excluding these samples substantially altered the DEG results by removing these artefactual region-specific signatures. By contrast, excluding the potential outliers identified in the bulk RNA-seq dataset had only minor effects on the DEGs detected (Fig. 1.1.2b, analyses 3 vs 4), including for genes linked to forebrain development (Fig. 1.1.3). Notably, apart from FEZF2—which did not meet significance thresholds upon removal of these samples—the **predicted gene-regulatory networks centred on the three chromosome-21 transcription factors remained largely preserved**, with similar TF-target interactions (Fig. 1.1.4). For example, after removing the three potentially outlier samples, the GRN analysis predicted 78 PKNOX1 targets, compared to 73 before, 186 BACH targets (vs 189 before), 172 GABPA targets (both before and after removal). Importantly, 82 of 322 predicted Chr21 TF targets identified after removal of the three potential outlier samples were intellectual disability genes, compared to 84 of 312 before.

Altogether, this shows that the removal of the 3 non-cortical samples during the initial manuscript revision was a crucial step to improve the robustness of the dataset and conclusions, while removing the apparent outliers from the bulk RNA-seq analysis does not alter our overall conclusions, underscoring robustness and broad validity of the study.

a

Key iterations of the snMultiome analysis

1) First Manuscript Submission (High stringency QC filtering)

Cells included:
 nCount_ATAC < 25000 &
 nCount_RNA < 30000 &
 nCount_ATAC > 500 &
 nCount_RNA > 500 &
 percent.mt < 2 &
 nucleosome_signal < 2 &
 TSS.enrichment > 1.5

Libraries removed:
 libraries with low cell number/UMI/ATAC counts (<1000), repeated libraries from same sample with lower quality libraries with fraction of removed low quality nuclei >0.3 &
 <1000 cells retained after QC

Excluded libraries: B10C1Q, B10D1N, B11C1A, B11C1C, B11C2H, B11D1B, B11D1E, B11D2F, B11D2G, B12C1A, B12C2G, B12C3B, B12C4E, B12D1F, B12D2D, B12D3C, B12D4H, B13C1, B13C2C, B13D1, B13D2D, B13D3G, B13D3H, B13D4F, B14C1G, B14D1H, B15C1H, B15C1L, B15D1K, B15D1M, B16C1H, B16D1L, B17C1J, B17C2L, B17C2M, B17D1K, B17D2F, B18C1P, B18C1S, B18C2M, B18D1Q, B18D1R, B18D2L

Excluded samples: B10C1, B10D1, B12C1, B12D4, B14D1, B15C1, B16C1, B17C2, B17D2, B18C1, B18D1, B18D2

2) Lower stringency QC filtering

=> To increase sample size, retained cells with lower quality ATAC data (not relevant for RNA analyses) and libraries with lower numbers of high-quality cells (poor quality cells could e.g. come from partially frozen/thawed parts of tissue => cells in unaffected parts could still be of good quality)

Cells included:
 nCount_ATAC < 25000 &
 nCount_RNA < 30000 &
 nCount_ATAC > 100 &
 nCount_RNA > 500 &
 percent.mt < 2 &
 nucleosome_signal < 2 &
 TSS.enrichment > 1.1

Libraries removed:
 libraries with low cell number/UMI/ATAC counts (<1000), repeated libraries from same sample with lower quality libraries with fraction of removed low quality nuclei >0.5 &
 <500 cells retained after QC

Excluded libraries: B12D4H, B14D1H, B15C1H, B15C1L, B17D2F

Excluded samples: B12D4, B14D1, B15C1, B17D2

3) Current Revised Manuscript (Lower stringency, non-cortical samples excluded)

=> To avoid bias from non-cortical cells "contaminating" cortical cell clusters, remove samples with large numbers of cells mapping to non-cortical regions of reference atlas

Cells included:
 nCount_ATAC < 25000 &
 nCount_RNA < 30000 &
 nCount_ATAC > 100 &
 nCount_RNA > 500 &
 percent.mt < 2 &
 nucleosome_signal < 2 &
 TSS.enrichment > 1.1

Libraries removed:
 libraries with low cell number/UMI/ATAC counts (<1000), repeated libraries from same sample with lower quality libraries with fraction of removed low quality nuclei >0.5 &
 <500 cells retained after QC

Excluded libraries: B12D4H, B14D1H, B15C1H, B15C1L, B17D2F, B11C1A, B11C1C, B12C3B, B10D1N

Excluded samples: B12D4, B14D1, B15C1, B17D2, B11C1, B12C3, B10D1

4) Lower stringency, non-cortical samples and outliers from bulk RNA-seq excluded

=> To avoid bias from potential outlier samples identified in bulk RNA-seq (as requested by reviewers)

Cells included:
 nCount_ATAC < 25000 &
 nCount_RNA < 30000 &
 nCount_ATAC > 100 &
 nCount_RNA > 500 &
 percent.mt < 2 &
 nucleosome_signal < 2 &
 TSS.enrichment > 1.1

Libraries removed:
 libraries with low cell number/UMI/ATAC counts (<1000), repeated libraries from same sample with lower quality libraries with fraction of removed low quality nuclei >0.5 &
 <500 cells retained after QC

Excluded libraries: B12D4H, B14D1H, B15C1H, B15C1L, B17D2F, B11C1A, B11C1C, B12C3B, B10D1N, B11C2H, B12C4E, B12D2D

Excluded samples: B12D4, B14D1, B15C1, B17D2, B11C1, B12C3, B10D1, B12D2, B11C2, B12C4

b

Changes in NEU_RORB abundance are robust across analysis parameters/iterations

Figure 1.1.1: Inclusion/exclusion criteria and rationale for different iterations of the analyses of the multiome dataset, and impact on detected cell abundance changes

a DEGs detected in excitatory lineage are prominently affected by non-cortical samples

b DEG numbers in excitatory lineage are prominently affected by removal of non-cortical samples, but not removal of potential outlier samples from bulk RNA-seq

Figure 1.1.2: Impact of different inclusion/exclusion criteria on detected differentially expressed genes

a Changes in genes linked to "Forebrain development" (GO:0030900) not prominently affected by potential outlier samples from bulk-RNA-seq

3) Current Revised Manuscript (Lower stringency, non-cortical samples excluded)

4) Lower stringency, non-cortical samples and outliers from bulk RNA-seq excluded

Figure 1.1.3: Impact of different inclusion/exclusion criteria on detected differentially expressed genes linked to forebrain development

Figure 1.1.4: Impact of different inclusion/exclusion criteria on predicted gene-regulatory networks

- some of the samples that are indicated as excluded in the supplementary table (B11C1A, B11C1C, B12C3B), are in the figure of the included samples based on the UMAP of extended data 1c.

1.2) We are very grateful to the reviewer for catching this inconsistency. We have now corrected the Extended Data Figure 1c legend to accurately reflect the complete set of samples analysed with excluded samples clearly marked and now consistent with Supplementary Table 1.

- Regarding the bulk analysis in Extended Figure 6c, the authors could use the PCA data to explain which is the major source of variation between the samples. The difference in change

of the scale of the correlation can indeed be misleading, therefore the authors should use the same scale.

1.3) We thank the reviewer for this helpful suggestion. We performed an in-depth analysis of potential sources of variation, as outlined in response to point 2.1.1. This indicates that indeed the main axes of variation (PC1 and 2) are driven by genes linked to cell cycle/proliferation and neuronal function (e.g. synapse genes), respectively. This strongly supports our hypothesis that different proportions of progenitors vs differentiated neurons are the main source of variation in the bulk dataset. This is most likely explained by differences in the proportions of progenitor zones versus cortical plate areas sampled, resulting from tissue fragmentation during surgical termination procedures—an effect we also observed in our initial immunostainings used to characterise these samples (Extended Data Fig. 1a).

Additionally, we have corrected the correlation (DEG overlap) heatmaps to use a consistent colour scale across all plots to allow for fair comparison. We note that these visual corrections do not materially change the quantitative results, which we have already calculated and showed, but do provide a more accurate visual representation of overlap and effect sizes (see also reply 2.4).

- when analyzing the code, it appears that the authors did not consider batches/samples in the integration. Is this the case?

1.4) We used the individual libraries as basis for the dataset integration in scripts 'B02_v040_integrate_samples_RNA_Harmony.R' and 'C01_v040_subsetting_reintegration.R', which will also correct for changes across batches. While this approach may underestimate differences in cell abundance when integrating very different samples (e.g. if populations are completely missing in some samples or conditions), the validated reduction of RORB/FOXP1 neurons indicates that this approach does not strongly dilute abundance changes in our dataset.

- is there size correction the differential gene expression for the number of N of the samples or conditions to compare? Also, were the samples adjusted for differences in total counts, for the comparisons performed? This is important as pseudobulk analysis can lead to biases and it is possible that samples with more cells could show upregulation of genes but without biological meaning. Were patients/samples considered covariates for DGE analyses?

1.5) We thank the reviewer for this important point regarding potential biases in pseudobulk analysis. As outlined in the Methods and detailed in script "E02_[...]", we have performed standard DESeq2 workflow analyses, which corrects for differences in sequencing depth (i.e. read counts). As we performed comparisons by cluster (i.e. each sample represented only by one pseudobulk in the respective cluster), we did not include the patients/samples as separate covariates.

The different cell numbers per pseudobulk are accounted for to some extent by the DESeq2 sequencing depth normalisation. To further assess the impact of variable cell numbers per sample and cluster, we ran a pseudobulk DEG analysis with equal cell numbers per cluster and sample, subsampling large cluster-samples to 100 cells and removing smaller cluster-samples. While this strongly reduced the number of detected DEGs, reducing power by both reducing counts per cluster-sample and number of cluster-samples (Fig. 1.5a), key changes in transcription factors and other neurodevelopmental genes were preserved (Fig. 1.5a/b). For example, **all 3 Chr21 TFs were identified in both analyses**, and 7 of 10 TFs linked to forebrain development identified in the original analysis were also detected in the downsampled analysis despite the lower statistical power.

However, we acknowledge now explicitly in the discussion as a limitation that because of limited cell numbers resulting in low total counts per pseudobulk/sample, the pseudobulk approach has limited sensitivity to detect DEGs in rare cell populations.

a Subsampling to adjust for different cell numbers per pseudobulk reduces sensitivity to detect DEGs, but recovers key DEGs detected in the analysis including all cells

b Changes in genes linked to "Forebrain development" (GO:0030900) retained after subsampling

Figure 1.5: Dataset subsampling to assess impact of variation in cell numbers on pseudobulk DEG analysis

- For the sample integration. were any covariate used? Patient/sample/batch factor can influence downstream variation and results.

1.6) As described in 1.4, we integrated the libraries on individual library basis, which also corrects for sample and batch effects.

- in Extended Supplementary Table 1 includes the list of libraries used and the original Cellranger QCs. The authors need to add the final median stats per sample used, after QC, and final number of nuclei in each sample. This is important to determine whether there is any bias between samples/batches.

We thank the reviewer for this suggestion, which improves clarity and transparency regarding the quality measures applied to the final dataset. We have included these now in Supplementary Table 1.

Referee 2

In the revised manuscript, the authors have introduced some modifications to address the pending issues. However, these changes appear somewhat incremental and do not specifically address the concerns previously raised by this reviewer. Consequently, the conclusions of the article still lack strong experimental support.

Major points: 1) The bulk RNAseq data have not been cleaned up and there are still two control samples clustering with DS samples. PCA clustering seems not “driven by variable cell-type proportions between samples” as stated by the authors but rather only by gene expression changes that indeed are very limited in this dataset possibly because authors still include the two samples that potentially misplaced. I would recommend remove these samples from both bulk and single-cell dataset and verify that indeed the global conclusions hold.

2.1) We thank the reviewer for these suggestions. We would like to take this opportunity to provide additional evidence—beyond what can be included in a concise manuscript—that the lack of clear sample separation in the PCA and the subtlety of the observed gene expression changes are not due to “misplaced” samples. Rather, they reflect a combination of genuinely low magnitude biological/transcriptional alterations in DS and substantial biological and technical variability inherent to human tissue samples.

2.1.1) While a clean separation of RNA-seq samples in PCA plots is often seen in highly controlled experimental setups (e.g. different treatments or deletion of key regulators in cultures of genetically identical cell lines or mouse tissues), human post-mortem tissue samples are much more heterogenous. Importantly, the human tissue is derived from genetically diverse donors, which represents a major source of variation. This genetic diversity also underlies the well-established range in the severity of DS phenotypes and can therefore be expected to contribute to variability at the transcriptomic level as well. Sample or library quality is another source of variation. However, only the library we excluded in the manuscript (B12D2/IGF137661) showed a poorer library quality compared to the other included samples/libraries, whereas the other potential outlier samples (B11C2, B12C4) show no signs of poor quality (Figure 2.1.1a/b). To get a better understanding of the most important sources of transcriptional variance of our bulk samples (PC1 and PC2), we assessed the functions of the top 100 genes positively or negatively driving PC1 and PC2. Gene Ontology analysis (Fig. 2.1.1c) revealed that these genes are linked to cell cycle and proliferation (neg. PC1 genes, pos. PC2 genes), neuronal function (pos. PC1 genes), and metal ion stress response (neg. PC2 genes). This strongly supports our hypothesis that different proportions of progenitors vs differentiated neurons are the main source of variation in the bulk dataset. This is most likely explained by differences in the proportions of progenitor zones versus cortical plate regions containing differentiated neurons, arising from tissue fragmentation during surgical termination. We observed this variability already in the initial immunostainings used to characterise these samples (Extended Data Fig. 1a).

Further supporting this hypothesis, while the expected subtle (1.5-fold) deregulation of Chr21 genes between DS and CON samples is highly reproducibly detected for all samples, except for a subset of genes in the excluded sample B12D2 (Figure 2.1.1d), the expression of cell type markers is much more variable within the DS and CON groups (Figure 2.1.1e).

Therefore, although we cannot definitively exclude additional factors contributing to variation and sample clustering in the dataset, our analyses strongly indicate that a substantial proportion of this variation is driven by differences in cell-type composition—a well-established confounding factor in bulk RNA-seq analyses and a major motivation for the development of single-cell transcriptomic approaches to overcome this limitation.

Figure 2.1.1: Assessment of bulk RNA-seq sample quality and sources of variation

2.1.2) To further assess the impact of the potential outlier samples on the detection of differentially expressed genes, we also re-analysed the bulk dataset excluding the potential outlier samples (B11C2, B12C4, B12D2), as recommended by the reviewer. This resulted only in a small increase of detected DEGs, further supporting that the lower number of DEGs detected by the bulk RNA-seq analysis is not due to the impact of the potential outlier samples, but due to the increased sensitivity of snRNA-seq to detect cell-type-specific changes (Fig. 2.1.2a/b).

Figure 2.1.2: Impact of potential outlier samples on DEG detection by bulk RNA-seq analysis

2.1.3) As detailed in rebuttal response 1.1, we further verified that the major conclusions of the multiome analysis were maintained when excluding the potential outlier samples (B11C2, B12C4, B12D2) and running a complete re-analysis of the dataset.

Altogether, **these analyses confirm the robustness of our findings and only a limited effect of the potential outlier samples B12C4 and B12D2.** We therefore propose to retain the original analysis including these samples, for comprehensiveness, transparency and conciseness.

2) The overlap between bulk and single-cell is very limited. The authors claim that it is a sign of superiority of single-cell over bulk sequencing. Another possibility is that the datasets in this work are not robust enough.

2.2) A limited overlap between DEGs in bulk- and single-nucleus RNA-seq analyses is expected, given the technical differences between the two approaches, and consistent with previous reports.

First, the two approaches capture different fractions of cellular RNA (whole cell vs nuclear). Second, changes in cell type-specific genes detectable by snRNA-seq get diluted in bulk analyses, in particular if cell type-proportions are variable as we have highlighted in 2.1. Accordingly, a previous study in human brain tissue has shown that the expression measured by bulk and snRNA-seq correlated significantly only for 37% of genes in the same set of 31 tissue samples (Velmeshev et al., Science 2019). Similarly, in our analysis, despite the small bulk RNA-seq dataset, an average of 27.5% of genes significantly altered in the snRNA-seq analysis show concordant changes in bulk RNA-seq. This increases to an average of 46.8% for upregulated genes, reaching over 70% in two clusters (Fig. 2.1.2b). To directly assess to what extent the difference between bulk vs cell-type resolved snRNA-seq analysis can explain the low numbers of DEGs in our bulk RNA-seq analysis, we generated a sample-level pseudobulk RNA-seq dataset from our snRNA-seq dataset by combining all unfiltered cells from each sample. As expected, given the higher power, the analysis retrieved more DEGs (342 DEGs from 15 vs 15 samples) than the analysis of the much smaller PCW11-14 actual bulk RNA-seq dataset (100 DEGs from 6 vs 5 samples). However, this analysis retrieved far fewer DEGs than the snRNAseq analysis (672 DEGs), highlighting the value of the cell-type resolved analysis and in line with previous bulk fetal cortex datasets, where substantial cell-type heterogeneity typically limits the number of detectable DEGs (Velmeshev et al., 2019, Science, Herring et al Cell 2022 and Polioudakis et al Neuron 2019).

(Pseudo-)bulk analysis of samples used for snRNA-seq analysis misses many neuronal DEGs captured by cluster-based snRNA-seq analysis

Figure 2.2: Impact of bulk vs cluster-based snRNA-seq analysis approach on DEG detection

We appreciate the reviewer's concern regarding the robustness of our findings. We would like to emphasize that our main conclusions are supported by several independent lines of evidence. **First**, our analyses unbiasedly identified established key players in Down syndrome pathology, including DYRK1A, which is already a target under clinical evaluation for DS. The recovery of such canonical regulators through independent datasets underscores the reliability of our approach. **Second**, our main conclusions have been independently validated by two other laboratories, including confirmation of two out of the three newly identified chromosome 21 transcription factors. These external validations provide strong, orthogonal support for the biological relevance and reproducibility of our findings. **Finally**, we have further reinforced our analyses by performing sensitivity checks (e.g., removal of outlier samples and recalculation of differential expression and pathway enrichment), which confirmed that all principal conclusions remain unchanged. Together, these results demonstrate that our data and interpretations are robust, reproducible, and consistent with independent studies in the field.

3) In the in vitro sequencing experiments and transplanted organoids is still not clear the number of biological replicates (not technical !) analyzed. The point-by-point letter suggests that only one biological replicate was used. If this is the case, no robust conclusion can be drawn from these experiments, and they should possibly be discarded. Moreover, in the in vitro dataset the samples are not evenly distributed across genotype/developmental stage.

2.3) We thank the reviewer for this comment and confirm that **we did not use a single biological replicate**. We have adapted the Methods section and Supplementary Tables 5 and 6 to clarify the numbers and types of replicates. To ensure the robustness of our findings, we employed a multi-level replication strategy designed to account for different sources of biological variation. We explicitly define our replicates **for all in vitro experiments** as follows:

2.3.1) For our in vitro analyses, we used a paired design to minimise variation unrelated to Trisomy 21 by comparing side-by-side differentiated DS cultures with their corresponding isogenic CON cultures, which we treat as technical replicates (Fig. 2.3.1a). We agree that, taken in isolation, such a design would not allow conclusions to be generalised beyond the individual experiment. To ensure reproducibility and generalisability, we repeated these paired experiments across independent differentiation rounds (biological replicates) and with two different pairs of isogenic iPSC lines, thereby excluding that any effect is confined to a single genetic background—biologically analogous to confirming a phenotype across two inbred mouse strains. With the additional differentiation experiments now included, we have 6 and 2 DS/CON biological-replicate pairs for NPCs from the two isogenic iPSC line pairs (Fig. 2.3.1a). For neuron cultures, we now have 3 biological-replicate pairs from one iPSC line pair

(C9/C13), independently replicated once using the second iPSC line pair (DS2U/DS1; Fig. 2.3.1a).

As expected, the individual differentiation experiments show variable numbers of detected DEGs (as presented in the previous manuscript version), where multiple technical replicates allow to assess DEGs for the individual experiments (Fig. 2.3.1b, left panel). To identify genes that are consistently changed across the differentiation experiments (biological replicates), we applied a multivariate regression model for the DESeq2 analysis (Fig. 2.3.1a), providing a higher statistical power than pooling technical replicates.

This analysis identifies ~5000-8000 genes displaying consistent changes when comparing across the different independent differentiation experiments for the different cell lines and differentiation stages (Fig. 2.3.1b, right panel). We have updated the relevant figures in the manuscript (Fig. 4b, c, 5d, e, Extended Data Fig. 9a, b, 10c, Supplementary Tables 5, 6), replacing the comparisons by batch with these cross-experiment analysis results for the different cell lines and differentiation stages. We have also updated the Methods accordingly.

These analyses further support the findings from the previous manuscript version. The consistently upregulated genes largely overlap with those elevated across cell populations in the DS fetal cortex, whereas the overlap is lower for downregulated genes in tissue. The new analyses also reinforce our observation that subsets of genes, including several linked to forebrain development, exhibit in-vitro changes that diverge from those seen in tissue, often in an isogenic-pair-dependent manner.

We also added data from four additional biological replicates for the ASO Chr21 TF knockdown experiment, providing a much more robust molecular functional validation of the 3 Chr21 TFs. We have updated the relevant figures and table in the manuscript (Fig. 4d, Supplementary Table 5). This confirmed SYNE2 and NEUROD1, two predicted TF targets already detected in the previous dataset, as well as a number of additional predicted targets, including intellectual disability-linked genes MYT1, SOX4, ATP1A1, HS6ST2, SRGAP3, BCL11A, MEGF10 and SLC6A1, which showed a trend towards reversal. We also added an additional panel to Extended Data Fig. 9 (Fig. 9g), with the results of a validation experiment showing that the non-targeting ASO have a minimal effect on gene expression compared to a subset of targeting ASOs.

a

Numbers of biological and technical replicates (more details see Supplementary Data Table 5)

Basic characterisation of DS iPSC model

cell_type	isogenic_pair (genetic background replicates)	batches (biological replicates; paired DS/CON)	samples (total technical replicates)
iNPC	C9_C13	6 (previously 1)	9 CON/10 DS
	DS2U_DS1		2 8 CON/8 DS
iNEU	C9_C13	3 (previously 1)	8 CON/9 DS
	DS2U_DS1		1 3 CON/3 DS

ASO experiment (normalisation of elevated Chr21 TF levels)

cell line	ASO target	batches (biological replicates target ASO vs no_AS0)	samples (total technical replicates, including 3 ASO designs)
C9 (CON)	non_targeting		1 2 no_AS0/3 ASO
	non_targeting		1 3 no_AS0/3 ASO
C13 (DS)	PKN0X1	5 (previously 1)	7 no_AS0/14 ASO
	BACH1	5 (previously 1)	7 no_AS0/14 ASO
	GABPA	5 (previously 1)	7 no_AS0/16 ASO

b

Paired design robustly identifies DEGs across multiple batches, cell types and genetic backgrounds

Number of DEGs for individual batches with multiple technical replicates (as in previous manuscript, with additional batches)

Number of DEGs by cell type and pair of cell lines (significant group effect across all batches)

Figure 2.3.1: Experimental design for the basic differential expression analysis of the *in vitro* model

For the *in vivo* transplantation experiments, we first would like to clarify that we transplanted dissociated neural progenitors, not organoids (as outlined in manuscript Fig. 5a and Methods). We used a single isogenic pair (DS1 and DS2U), but multiple independent differentiation batches (three for Control, two for DS) and a total of 17 host animals ($n = 9$ Control, $n = 8$ DS), each treated as an independent biological replicate, providing a statistically valid basis for our findings. We acknowledge that the experimental design is not ideally balanced, due to the substantial technical challenges of these experiments (including logistics of combining lengthy iPSC culture protocols with stereotaxic surgeries, maintenance of immunodeficient animals, challenging dissection of relatively small grafts with limited material for nuclei extraction). Remarkably, despite these limitations, our dataset shows that the xenografts recapitulate many of the molecular phenotypes observed in the tissue, suggesting that our analysis may in fact underestimate the true extent of concordance between xenografts and tissue.

4) In many panels comparing datasets (Extended 6e, 4b, 5d), the scale is still not set between 0 and 1 as should be. This is visually increases the soundness of the results and potentially masks a very limited overlap between some of the datasets.

2.4) We corrected all relevant panels so the colour scaling is consistent and appropriate (0–1 where indicated). Our original rationale was to show more clearly that there are genes validated *in vitro* (which might be functionally relevant) even if presenting in some cases only around 20% of the DEGs detected per cluster in tissue.

5) In my opinion, the conclusion of the knockdown experiments should be strongly dumped since no phenotypic rescue was found, and the applicability of ASO treatment to fetuses is questionable.

2.5) We thank the reviewer for this feedback, which has highlighted an area where our manuscript lacked sufficient clarity. We would like to clarify that **this experiment was not designed to achieve a cellular phenotypic rescue**, and accordingly no phenotypic rescue was assessed or performed. We also made no such claim in the manuscript. The ASO-mediated knockdown of the Chromosome 21 TFs was included solely to demonstrate target validation and molecular rescue of dysregulated intellectual disability–related gene targets, not to evaluate functional outcomes. We have moderated our conclusions throughout the paper to precisely reflect this experimental aim and have noted this in the limitations section. We emphasise that beyond the rescue of molecular phenotypes more research is needed to assess the rescue of cellular and cognitive/behavioural phenotypes. We also highlight that the feasibility of ASO treatment in fetuses or at later stages needs to be determined in future *in vivo* studies.